# Theoretical Analysis of Contrastive Learning in Vision-Language Model Pre-Training: The Role of Synthetic Text Captions for Feature Alignment

## Abstract

Vision-language models (VLMs) pre-trained on web-sourced image-text pairs have achieved remarkable success in multimodal tasks. Incorporating synthetic text captions during pre-training has been shown to enhance image-text alignment, significantly improving model performance. Despite these empirical advances, the theoretical understanding of how VLMs align modalities, extract features, and achieve zero-shot capabilities remains limited. This paper provides the first theoretical analysis of VLM training dynamics with nonlinear activation functions and offers the first theoretical support for synthetic text captions in enhancing pre-training performance. Specifically, we analyze the impact of misaligned image-text pairs, showing that neurons trained on noisy data learn mixtures of true and spurious features, degrading generalization. In contrast, text generated by image-grounded text decoders reduces spurious correlations and improves model accuracy, enabling success in zero-shot multi-class classification where models trained on raw text fail. While our analysis uses simplified models for theoretical tractability, our findings are validated through experiments on state-of-the-art VLMs, such as BLIP.

## 1. Introduction

Vision-language models (VLMs) have recently demonstrated remarkable progress across various multimodal tasks including vision-language understanding and generation. State-of-the-art methods, such as CLIP (Radford et al., 2021) and SimVLM (Wang et al., 2021), leverage contrastive learning to pre-train on large-scale image-text pairs sourced from the web. These models achieve outstanding performance on downstream tasks, particularly in zero-shot settings, where predictions are made on entirely unseen data or classes without task-specific training.

Web-sourced image-text pairs, however, often contain noisy and low-quality text. In many cases, the text includes spurious or irrelevant information that does not directly correspond to the image. Training on such low-quality data can lead to a misalignment of image and text features, undermining the model's generalization capability on downstream tasks. For instance, Nguyen et al. (2024) highlight an example of a blue Mercedes-Benz car in a parking lot, accompanied by a raw text caption: "2003 Mercedes-Benz C240 sedan, Leather, MUST BE SEEN - $6199." The price information in this caption is only superficially correlated with the image and does not contribute meaningfully to understanding the image context.

To address this issue, text generation methods (Nguyen et al., 2024; Wang et al., 2022b; An et al., 2022; Rotstein et al., 2024; Hu et al., 2022; Wang et al., 2022a) are widely adopted during VLM training to produce high-quality synthetic text captions that are more faithful to their respective images compared to raw text. For instance, BLIP (Li et al., 2022a) has demonstrated that incorporating synthetic text captions can greatly improve the quality and diversity of the training data, resulting in significantly enhanced model performance. To further validate the quality of synthetically generated text, Nguyen et al. (2024) show that the image-text cosine similarity between the synthetic text captions generated by BLIP2 (Li et al., 2023a) and the original image is higher than that for raw text paired with the image.

Despite the impressive success of large VLMs and the practical advancements driven by synthetic text captions, their theoretical foundations remain relatively underdeveloped. Several critical questions remain mostly open:

- How do VLMs align different modalities during pre-training, extract feature representations, and achieve zero-shot capabilities from noisy image-text pairs?
- How do synthetic text captions generalization provably enhance generalization performance?

Addressing these theoretical challenges could provide deeper insights into pre-trained VLMs, offering principled explanations and guidelines to advance their development.

Notably, even the theoretical understanding of vanilla multimodal contrastive learning is yet to be fully developed. For instance, built on the spectral contrastive loss (HaoChen et al., 2021), Zhang et al. (2023a) connects its multimodal counterpart to matrix factorization, demonstrating the existence of good solutions and characterizing their general-

Table 1: Comparison with existing theoretical works on contrastive learning

| Theoretical Works | Training Dynamics | Nonlinear Activation | Zero-shot Generalization | Synthetic Text Captions Enhancement | Multi-modal |
|---|---|---|---|---|---|
| Wen & Li (2021) | ✓ | ✓ | | | |
| Nakada et al. (2023) | ✓ | | | | ✓ |
| Chen et al. (2024) | ✓ | | ✓ | | ✓ |
| Lee et al. (2021) | | ✓ | | | ✓ |
| Zhang et al. (2023a) | | | | | ✓ |
| This paper | ✓ | ✓ | ✓ | ✓ | ✓ |

ization gaps. Furthermore, Huang et al. (2021), Lee et al. (2021), and Zadeh et al. (2020) show that, under certain conditions, multimodal models outperform single-modal models with better representation learning. However, these studies assume convergence to the optimal training loss, lacking analysis of the training dynamics that yield models with good generalization and thus overlooking challenges posed by the loss function's non-convexity. Additionally, they do not address the zero-shot capabilities of the learned VLMs. Chen et al. (2024) characterize CLIP's zero-shot performance by proving its ability to learn shared features while disregarding self-standing features, however, their framework does not account for the practical challenges posed by misaligned features. Additionally, Nakada et al. (2023) introduce a new contrastive loss leveraging unpaired datasets to detect ground-truth pairs and enhance performance, but their results are limited to linear networks without non-linear activations. No existing works theoretically analyze the impact of synthetic data on VLM performance.

**Contributions:** To the best of our knowledge, this work provides the first theoretical support for the enhanced zero-shot generalization performance of synthetically generated data in pre-training VLMs, especially when web-sourced raw data contains spurious or irrelevant information. We provide the training dynamics analysis of multimodal contrastive learning with generalization guarantees using stochastic gradient descent (SGD) in the presence of spurious alignment in the data model. Our analysis also moves from linear models to non-linear models, by analyzing the one-hidden-layer neural network model, which remains widely used in theoretical analysis of uni-modal contrastive learning (Wen & Li, 2021) and supervised learning (Li et al., 2023b; Allen-Zhu & Li, 2022; Zhang et al., 2023b). Furthermore, all theoretical insights are validated on practical VLMs, such as the BLIP model. Table 1 compares our work with existing theoretical works on contrastive learning. Our specific contributions include:

**1. Theoretical training dynamics analysis of contrastive learning in nonlinear VLMs.** We provide a theoretical demonstration of how neurons learn correlated image and text features using contrastive loss. Prior training dynamics analyses of multi-modal contrastive learning Chen et al.

(2024); Nakada et al. (2023) are limited to linear neural networks, while our analysis applies to nonlinear neural networks with ReLU activation functions.

**2. Theoretical characterization of the impact of mis-aligned image-text pairs on pre-training performance.** We analyze a data model where a fraction of text contains features spuriously related to image features. Our results show that neurons in the resulting model learn a mixture of true features and spurious features together, preventing the VLM from accurately distinguishing these features and leading to degraded generalization performance.

**3. Theoretical justification of enhanced out-of-domain generalization through pre-training with synthetic text captions.** We demonstrate that a properly trained image-grounded text decoder can generate text containing fewer spurious features than raw text data. Then the model trained with synthetic text captions can align image and text features properly, resulting in accurate results in a zero-shot multi-class image classification problem, whereas the model trained on the original data is guaranteed to fail.

### 1.1. Related Works

**Vision-language Models:** VLMs (Yu et al., 2022; Wang et al., 2023a; Radford et al., 2021; Jia et al., 2021; Li et al., 2020; 2021) rely on contrastive learning with large-scale image-text pairs from the web. Building on the CLIP model, a series of studies (Li et al., 2022b; Alayrac et al., 2022; Yao et al., 2022) have emerged with the primary goal of further improving its zero-shot performance. However, spurious alignments in noisy and web-curated datasets significantly limit their performance. Data selection has emerged as a critical challenge in pre-training VLMs, driving the development of recent filtering methods (Li et al., 2023a; Wang et al., 2024; Kim et al., 2025; Li et al., 2024) to overcome this issue. For instance, BLIP (Li et al., 2022a) effectively uses synthetic text captions to filter noisy image-text pairs, showcasing enhanced robustness and reliability in zero-shot performance.

**Theory of Contrastive Learning:** Many studies (Saunshi et al., 2022; Tian et al., 2020; Saunshi et al., 2019) have provided theoretical guarantees for contrastive learning in

unimodal settings. Tian et al. (2021); Wang et al. (2023b) provides a comprehensive theoretical understanding of the dynamics of self-supervised learning. (Wen & Li, 2021) demonstrated the training dynamics of contrastive learning and explained how neurons learn specific features through proper data augmentation, even in the presence of noise. However, these studies overlook multimodal scenarios and do not address the misalignment arising from noisy web-sourced image-text pairs, impacting vision-language alignment and performance.

## 2. Problem Formulation and Algorithm

**Preliminary on Pre-training Vision-Language Models (VLMs).** In general, VLMs leverage large-scale web-based datasets containing paired visual and textual data to pre-train two separate encoders: an image encoder $f$ and a text encoder $h$, parameterized by weights $\mathbf{W}$ and $\mathbf{V}$, respectively. Contrastive learning serves as the core framework, ensuring the learned embeddings of matching pairs are closer while separating mismatching pairs.

Specifically, let $S$ be the indices of the image-text pairs, e.g., $(x_p, y_p)$ with $p \in S$. $(x_p, y_p)$ is referred to as a positive pair, while $(x_p, y_n)$ with $p \neq n$ is referred to as a negative pair. We minimize the following spectral loss function:

$$\min_{\mathbf{W},\mathbf{V}} L(f,h), \text{ where } L(f,h) = \sum_{p \in S} \Bigg[ -\langle f(x_p), h(y_p)\rangle +$$
$$\sum_{n \in S\setminus\{p\}} \frac{(\langle f(x_n), h(y_p)\rangle)^2}{2\tau} + \sum_{n \in S\setminus\{p\}} \frac{(\langle f(x_p), h(y_n)\rangle)^2}{2\tau} \Bigg] \tag{1}$$

where the hyper-parameter $\tau > 0$ is referred as the temperature. The spectral contrastive loss $L$ in (1) has been extensively utilized in recent theoretical works (HaoChen et al., 2021; Shen et al., 2022; Zhang et al., 2023a). Although it differs from the commonly used SimCLR (Chen et al., 2020) in practice, the spectral contrastive loss closely resembles SimCLR numerically (HaoChen et al. (2021)).

### 2.1. Training Framework

Let $S = S_h \cup S_w$ include human-annotated high-quality image-text pairs with indices in $S_h$ and noisy web low-quality dataset with indices in $S_w$. Due to the inherently noisy nature of web data, the learned embeddings from (1) may be suboptimal. Instead, practical training frameworks, such as BLIP (Li et al., 2022a), incorporate synthetic text captions to enhance the quality and diversity of image-text pairs. The algorithm can be divided into four stages:

1. **Image-text contrastive pre-training (ITCP) on raw data:** The image encoder $f$ and text encoder $h$ are trained using the image-text pairs $\{(x_p, y_p)\}_{p \in S}$ by minimizing the contrastive loss as in (1). Let $\overline{\mathbf{W}}$ and $\overline{\mathbf{V}}$ denote the learned weights in $f$ and $h$. We then estimate the image and

text embeddings of $(x_p, y_p)$ by $z'_{x_p} = f_{\overline{\mathbf{W}}}(x_p)$ and $z'_{y_p} = h_{\overline{\mathbf{V}}}(y_p)$. Due to the low-quality data in $S_w$ when training the encoders, these estimations might not be accurate.

2. **Generating text captions:** We use the high-quality data pairs in $S_h$ to train an image-grounded text decoder $G$, which maps an image $x_p$ to text through $G(x_p)$. Then, the learned $G$ is applied to every image-text pair $(x_p, y_p)$ in $S_w$ to generate a synthetic caption $\hat{y}_p = G(x_p)$. Next, the estimated text embedding of $\hat{y}_p$ is computed as

$$\hat{z}_{y_p} = h_{\overline{\mathbf{V}}}(\hat{y}_p) = h_{\overline{\mathbf{V}}}(G(x_p)), \tag{2}$$

where $\overline{\mathbf{V}}$ represents the weights of $h$ learned from stage 1.

3. **Filtering:** For every $p$ in $S_w$, we compute the cosine similarity between the estimated image embedding $z'_{x_p}$ and the text embeddings $z'_{y_p}$, $\hat{z}_{y_p}$, respectively. If the pair $(z'_{x_p}, \hat{z}_{y_p})$ is closer than the pair $(z'_{x_p}, z'_{y_p})$, $(x_p, y_p)$ is replaced with $(x_p, \hat{y}_p)$. Let $\tilde{S}_w$ denote the index set of the resulting data pairs. By replacing noisy captions in $S_w$ with synthetic captions that better align with image embeddings, $\tilde{S}_w$ becomes a cleaner dataset.

4. **ITCP on synthetic data:** We use the data in $\tilde{S} = S_h \cup \tilde{S}_w$ to solve (3), where the data set $S$ is replaced with $\tilde{S}$, and the resulting contrastive loss is denoted by $\tilde{L}(f,h)$,

$$\min_{\mathbf{W},\mathbf{V}} \tilde{L}(f,h), \text{ where } \tilde{L}(f,h) = \sum_{p \in \tilde{S}} \Bigg[ -\langle f(x_p), h(y_p)\rangle +$$
$$\sum_{n \in \tilde{S}\setminus\{p\}} \frac{(\langle f(x_n), h(y_p)\rangle)^2}{2\tau} + \sum_{n \in \tilde{S}\setminus\{p\}} \frac{(\langle f(x_p), h(y_n)\rangle)^2}{2\tau} \Bigg] \tag{3}$$

Let $\widetilde{\mathbf{W}}$ and $\widetilde{\mathbf{V}}$ denote the resulting learned weights. $f_{\widetilde{\mathbf{W}}}$ and $g_{\widetilde{\mathbf{W}}}$ can produce improved embeddings compared with $f_{\overline{\mathbf{W}}}$ and $g_{\overline{\mathbf{V}}}$.

(1) and (3) are nonconvex. We apply the vanilla SGD with step size $\eta$ and later provide the training dynamics analysis and convergence guarantee in Section 4.

### 2.2. Downstream Tasks

The pre-trained model $(f_{\widetilde{\mathbf{W}}}, g_{\widetilde{\mathbf{V}}})$ is evaluated on a downstream image classification task in a zero-shot setting. Unlike the downstream regression and binary classification tasks used for uni-modal contrastive learning in (Wen & Li, 2021), we consider a $K$-classification problem for any constant $K \geq 2$. Each class label is associated with a given text prompt $y_k$, where $k \in [K]$. For any image $x$ with its ground-truth label $l_x \in [K]$, the zero-shot predicted label by the pre-trained models $(f_{\widetilde{\mathbf{W}}}, g_{\widetilde{\mathbf{V}}})$ is computed as:

$$\arg\max_{k \in [K]} \langle f_{\widetilde{\mathbf{W}}}(x), g_{\widetilde{\mathbf{V}}}(y_k)\rangle. \tag{4}$$

# 3. Technical Assumptions and Setups

We introduce a set of assumptions that are either derived conceptually from the real data distribution or follow existing approaches in contrastive learning theory.

## 3.1. Backbone of the Encoders

We use a one-hidden-layer neural network with ReLU activation functions as the backbone for both our image and text encoder networks, which are formally defined as

**Definition 3.1.** The image encoder $f_{\mathbf{W}} : \mathbb{R}^{d_1} \to \mathbb{R}^m$ and text encoder $h_{\mathbf{V}} : \mathbb{R}^{d_1} \to \mathbb{R}^m$ is expressed as

$$
\begin{aligned}
f(x) &= (f_1(x), \ldots, f_m(x))^\top \in \mathbb{R}^m, \\
f_i(x) &= \sigma\left(\langle w_i, x \rangle - b_i\right) - \sigma\left(-\langle w_i, x \rangle - b_i\right)
\end{aligned}
\tag{5}
$$

$$
\begin{aligned}
h(y) &= (h_1(y), \ldots, h_m(y))^\top \in \mathbb{R}^m, \\
h_i(y) &= \sigma\left(\langle v_i, y \rangle - b_i\right) - \sigma\left(-\langle v_i, y \rangle - b_i\right)
\end{aligned}
\tag{6}
$$

where $\sigma$ denotes the ReLU function, and $\mathbf{W} = [w_1, w_2, \ldots, w_m]^\top$, $\mathbf{V} = [v_1, v_2, \ldots, v_m]^\top \in \mathbb{R}^{m \times d_1}$.

The one-hidden-layer neural network model, though simple, is challenging for training dynamics analysis due to its non-linearity and is considered SOTA in supervised learning (Allen-Zhu & Li, 2022; Zhang et al., 2023b). Prior studies mainly focus on simpler settings, like one-hidden-layer uni-modal networks (Wen & Li, 2021) or linear multi-modal encoders (Nakada et al., 2023; Chen et al., 2024).

## 3.2. Data Formulation for ITCP

Our data model defined in Definition 3.2 extends the sparse coding in (Allen-Zhu & Li, 2022; Wen & Li, 2021) for uni-modal images to multi-modal image and text pairs contrastive learning. This sparse coding model is both favored in theoretical analyses (Arora et al., 2014; Barak et al., 2015; Gregor & LeCun, 2010) and effective in modeling both NLP (Arora et al., 2016; 2018; Prokhorov et al., 2021; Deng et al., 2023) and image data (Yang et al., 2011; Xiao et al., 2025; Yang et al., 2009).

**Definition 3.2** (Image and text pairs). For an image-text pairs $(x_p, y_p)$ where $p \in S$, the image $x_p \in \mathbb{R}^{d_1}$ and the text $y_p \in \mathbb{R}^{d_1}$ are generated i.i.d. from the following sparse coding form:

$$
x_p = \mathbf{M} z_{x_p} + \xi_{x_p}, \quad y_p = \mathbf{H} z_{y_p} + \xi_{y_p}.
\tag{7}
$$

where $z_{x_p}$ and $z_{y_p} \in \mathbb{R}^d$. We refer to $z_{x_p}$ and $z_{y_p}$ as the sparse signal and $\xi$ as the noise. We assume $d_1 = \text{poly}(d)$ for simplicity.

We assume the following on $\mathbf{M}, \mathbf{H}, z, \xi$, respectively:

**Assumption 3.3.** $\mathbf{M}, \mathbf{H}, z, \xi$ satisfy the following

(a) The image dictionary matrix $\mathbf{M} = [\mathbf{M}_1, \ldots, \mathbf{M}_d] \in \mathbb{R}^{d_1 \times d}$ is a column-orthonormal matrix, and satisfies $\|\mathbf{M}_j\|_\infty \le \widetilde{O}\left(\frac{1}{\sqrt{d_1}}\right)$ for all $j \in [d]$.

(b) The text dictionary matrix $\mathbf{H} = [\mathbf{H}_1, \ldots, \mathbf{H}_d] \in \mathbb{R}^{d_1 \times d}$ is a column-orthonormal matrix, and satisfies $\|\mathbf{H}_j\|_\infty \le \widetilde{O}\left(\frac{1}{\sqrt{d_1}}\right)$ for all $j \in [d]$.

(c) We call $\mathbf{M}^\perp = [\mathbf{M}_j]_{j \in [d_1] \setminus [d]}$ (the orthogonal complement of $\mathbf{M}$) the noisy features, which are the undesired features. $\mathbf{H}^\perp$ is defined similarly.

(d) The noise $\xi_{x_p}, \xi_{y_p} \sim \mathcal{N}(0, \sigma_\xi^2 \mathbf{I}_{d_1})$, and the variance satisfies $\omega(\frac{1}{d_1}) \le \sigma_\xi^2 \le O(\frac{1}{d})^1$.

(e) The coordinates of $z_{x_p} = (z_{x_p}^1, \ldots, z_{x_p}^d)^T$ take values from $\{1, 0, -1\}$ independently and are symmetric around zero, following

$$
\left| z_{x_p}^j \right| \sim \text{Bernoulli}\left(C_z/d\right), \quad \text{for } i = 1, 2, \ldots, d,
\tag{8}
$$

where $C_z$ is a constant.

Since $\omega(\frac{1}{d_1}) < \sigma_\xi^2 \le O\left(\sqrt{\frac{\log d}{d^{1+c_0}}}\right)$ where $c_0 \in (0, 1)$, the $\ell_2$-norm of $\xi$ becomes $\|\xi\|_2^2 \gg \Theta(1) \gg \|\mathbf{M}z\|_2$. However, whenever there is one $z_j \ne 0$, we have $|\langle \mathbf{M}z, \mathbf{M}_j \rangle| = \Theta(1)$ while $|\langle \xi, \mathbf{M}_j \rangle| \le O\left(\frac{1}{\sqrt{d}}\right)$ with high probability.

To capture the characteristics of high-quality and low-quality data in $S$, we introduce Assumptions 3.4 and 3.5.

**Assumption 3.4** (High-quality image-text pairs). Every high-quality image-text pair $(x_p, y_p)$ with $p \in S_h$ has a perfect alignment, i.e., $z_{y_p} = z_{x_p}$. Moreover, $|S_h| = \Theta(d^2)$.

To model the misaligned features in low-quality pairs in $S_w$, where spurious correlations occurs at a non-negligible level, we assume

**Assumption 3.5** (Low-quality image-text pairs). There exists a constant $C_s \in (\omega(1/\log d), 1/2)$ such that every image feature $\mathbf{M}_j$ ($j \in [d]$) is spuriously correlated with exactly one text feature $\mathbf{H}_{j'}$ ($j' \ne j$) in a low-quality pair $(x_p, y_p)$ with $p \in S_w$, with

$$
\begin{aligned}
\Pr\left(z_{y_p}^{j'} = z_{x_p}^j \mid |z_{x_p}^j| = 1\right) &= C_s \\
\Pr\left(z_{y_p}^{j'} = 0 \mid |z_{x_p}^j| = 1\right) &= 1 - C_s
\end{aligned}
\tag{9}
$$

Similarly, the image feature $\mathbf{M}_{j'}$ is spuriously correlated with exactly the text feature $\mathbf{H}_j$. Moreover, $|S_w| = \text{poly}(d) \gg \omega(d^2)$.

For example, consider the blue Mercedes-Benz car discussed by Nguyen et al. (2024) and referenced in the Introduction. Here, $\mathbf{M}_j$ corresponds to the image feature of the car, $\mathbf{H}_j$ represents the text feature describing the car,

---

[1]We follow the conventional notations that $f = O(x), o(x), \Omega(x), \omega(x), \Theta(x)$ describes the growth rate of $f$ relative to $x$, indicating whether $f$ grows at most, strictly slower, at least, strictly faster, or exactly at the order of $x$, respectively. $\widetilde{O}$, $\widetilde{\Omega}$, and $\widetilde{\Theta}$ notations to hide poly-logarithmic factors.

and $\mathbf{H}_{j'}$ corresponds to the price information, which is spuriously correlated with $\mathbf{M}_j$ in this image-text pair. (9) indicates that such spurious correlations exist with a constant probability. Although we assume pair-wise spurious correlations among features to simplify the analysis, our theoretical results are validated on a more general case where an arbitrary number of spurious features may exist for any feature $i \in [d]$, as demonstrated in Section 5.1. Finally, it is worth noting that the number of low-quality pairs is order-of-magnitude larger than the number of high-quality pairs, which may significantly degrade the model's performance.

### 3.3. Image-Grounded Text Decoder $G$ in Stage 2

Recall that $G$ is introduced in Stage 2 to generate synthetic text captions. In practice, the core idea behind the widely adopted approaches (Li et al., 2022a; Yu et al., 2022; Wang et al., 2023a) is to train the encoder-decoder model $G$ on high-quality image-text pairs $S_h$. In this paper, we consider a simplified form of $G$, given by:

$$G(x_p) = \mathbf{V}^T \sigma(\mathbf{W} x_p), \qquad (10)$$

where $\sigma$ denotes the ReLU function. The parameters $\mathbf{W}$ and $\mathbf{V}$ are learned by solving

$$\min_{\mathbf{W}, \mathbf{V}} L_C = \sum_{p \in S_h} \frac{1}{2} \left\| \mathbf{V}^T \sigma(\mathbf{W} x_p) - y_p \right\|_2^2, \qquad (11)$$

initialized at $\overline{\mathbf{W}}$ and $\overline{\mathbf{V}}$, using SGD with step size $\eta$. Although $G$ in (10) is a conceptual simplification, where $\sigma(\mathbf{W} x_p)$ acts as the encoder and $\mathbf{V}^T$ as the decoder, it serves as a realistic abstraction to illustrate the underlying advantages of synthetic text caption generation.

### 3.4. Zero-Shot Generalization on Image Classification

To formally quantify the generalization performance of the $K$-classification problem described in Section 2.2, we consider the following testing data setup:

**Out-of-domain image data**. Each testing image $x$ has a sparse coding representation similar to the training images,

$$x = \mathbf{M} z_x + \xi_x, \qquad (12)$$

where the noise $\xi_x$ follows the same distribution as the training data (as stated in Assumption 3.3(d)). We consider the more general out-of-domain setting where the distribution of $z_x$ differs from the training data. Specifically, $|z_x^j| \sim \text{Bernoulli}(C_z'/d)$ for all $j \in [d]$, where $C_z'$ may not equal $C_z$ from (8).

**Text prompts**. The text prompt $y_k$ for every class $k$ also has a sparse representation:

$$y_k = \mathbf{H} z_{y_k} + \xi_{y_k}, \, k \in [K], \qquad (13)$$

where each $z_{y_k}$ is a binary vector and $|z_{y_k}^j| \sim \text{Bernoulli}(\Theta(1/d))$. Moreover, if $x$ belongs to class $k$, i.e.,

$l_x = k$, then among all these $K$ binary vectors, $z_x$ is the most aligned with $z_{y_k}$,

$$\langle z_x, z_{y_k} \rangle > \langle z_x, z_{y_{k'}} \rangle, \forall k' \neq k, \text{ when } l_x = k \qquad (14)$$

This aligns with the intuition that $x$ belongs to class $k$ if its sparse signal is the most similar to the sparse signal of class $k$'s text prompt.

## 4. Main Results

### 4.1. Intuition and Informal Insights

Before formally presenting our main results, we first provide an intuitive explanation for the success of the encoder-learner. For a properly trained pair of image encoder $f$ and text encoder $g$ to learn the latent representation $z$ from $(x, y)$, each feature pair $(\mathbf{M}_j, \mathbf{H}_j)$ needs to be exclusively learned by at least one neuron pair $(w_i, v_i)$, without interference from spurious features, ensuring purified representations. This is referred to as *feature alignment*. Under these conditions, we have $\langle w_i, x \rangle \approx z_x^j$ and $\langle v_i, y \rangle \approx z_y^j$, ensuring that the pair $f$ and $g$ successfully captures the full latent space $z$. However, in practical scenarios where the dataset contains limited high-quality pairs in $S_h$ but is dominated by low-quality pairs with misaligned features in $S_w$, training $f$ and $g$ to achieve the above desirable properties becomes challenging. Our main theoretical insights include:

**1. SGD provably solves the nonconvex training problems (1) and (3).** The existing training dynamics and convergence analyses are limited to either single-modal contrastive learning (Wen & Li, 2021) or linear networks (Chen et al. (2024); Nakada et al. (2023)). Theorem 4.1 provides a convergence analysis of SGD for solving the nonconvex ITCG problem when the network contains nonlinear activations.

**2. Failure of feature alignment due to spurious correlations.** Theorem 4.3 provides a negative result: if $f$ and $g$ are directly trained on the raw data $S = S_h \cup S_w$, the model inevitably learns $\mathbf{M}_j$ and $\mathbf{M}_{j'}$ together via some $w_i$, and $\mathbf{H}_j$ and $\mathbf{H}_{j'}$ together via some $v_i$. As a result, the model fails to distinguish between these spuriously correlated features.

**3. Successful feature alignment on synthetic data**. Theorem 4.5 states that training $f$ and $g$ on the synthetic data $\tilde{S} = S_h \cup \tilde{S}_w$ enables the resulting encoder pair to learn purified representations of $\mathbf{M}_j$ and $\mathbf{H}_j$ accurately, as if trained solely on high-quality data. However, since $|S_h| \ll |S_w|$, training solely on $S_h$ is insufficient to achieve the result in Theorem 4.5. This highlights the necessity of leveraging the synthetic data $\tilde{S}_w$.

**4. Enhanced zero-shot image classification accuracy due to synthetic text captions**. The advantage of using synthetic text captions is further validated in downstream tasks. As shown in Theorem 4.7, for a zero-shot out-of-domain multi-class image classification task, ITCP trained using $\tilde{S}$

achieves high accuracy, whereas ITCP directly using $S$ fails to generalize accurately.

## 4.2. Impact of Synthetic Data on Feature Alignment of Convergent Models

We first characterize the training dynamics and convergence of solving (1) and (3) using SGD. Let $L^*$ and $\tilde{L}^*$ denote the optimal values of (1) and (3), respectively.

**Theorem 4.1** (**Convergence of ITCP**). *Suppose Assumptions 3.3 to 3.5 hold. Let the model complexity be* $m = d^{1.01}$, *initialized at* $w_i^{(0)}, v_i^{(0)} \sim \mathcal{N}(0, \sigma_0^2 \mathbf{I}_{d_1})$, *where* $\sigma_0^2 = \Theta\left(\frac{1}{d_1 poly(d)}\right)$. *After* $T = \Theta\left(d^2 \log d\right)$ *SGD iterations with batch size* $\Omega(d)$ *and* $\eta = O(1)$, *the returned weights to solve (1), denoted by* $\overline{\mathbf{W}}$ *and* $\overline{\mathbf{V}}$, *achieves a loss that is sufficiently close to* $L^*$, *i.e.,*

$$(L(f_{\overline{\mathbf{W}}}, h_{\overline{\mathbf{V}}}) - L^*)/|L^*| \leq o(1). \quad (15)$$

*Similarly, the returned weights to solve (3), denoted by* $\widetilde{\mathbf{W}}$ *and* $\widetilde{\mathbf{V}}$, *achieves a loss that is sufficiently close to* $\tilde{L}^*$, *i.e.,*

$$(\tilde{L}(f_{\widetilde{\mathbf{W}}}, h_{\widetilde{\mathbf{V}}}) - \tilde{L}^*)/\left|\tilde{L}^*\right| \leq o(1). \quad (16)$$

*Remark* 4.2. Theorem 4.1 demonstrates that SGD iterations can converge to weights that achieve a near optimal loss of (1) and (3), respectively. This result is of independent interest, as existing training dynamics and convergence analyses for contrastive loss are limited to linear networks. Here, we extend such analysis to nonconvex optimization settings where the network contains nonlinear ReLU activations. Next we characterize the feature alignment property of the learned models.

**Theorem 4.3** (**Unsuccessful feature alignment of ITCP on raw data** $S$). *Each neuron pairs* $(\bar{w}_i, \bar{v}_i)$ *in* $(\overline{\mathbf{W}}, \overline{\mathbf{V}})$ *($i \in [m]$) mainly learn the features in the set* $N_i \subseteq [d]$ *with* $|N_i| \geq 2$, *while ignoring other features. Specifically,*

$$\bar{w}_i = \sum_{j \in N_i} \alpha_{i,j} \mathbf{M}_j + \sum_{j \in [d] \setminus N_i} \beta_{i,j} \mathbf{M}_j + \sum_{j \in [d_1] \setminus [d]} \gamma_{i,j} \mathbf{M}_j^\perp$$

$$\bar{v}_i = \sum_{j \in N_i} \alpha_{i,j} \mathbf{H}_j + \sum_{j \in [d] \setminus N_i} \beta_{i,j} \mathbf{H}_j + \sum_{j \in [d_1] \setminus [d]} \gamma_{i,j} \mathbf{H}_j^\perp$$

$$(17)$$

*where* $\alpha_{i,j}^2 = \Theta(1)(\|\bar{w}_i\|_2^2 + \|\bar{v}_i\|_2^2)$, $\frac{\beta_{i,j}}{\alpha_{i,j}} \leq O(\frac{1}{\sqrt{d}})$, *and* $\frac{\gamma_{i,j}}{\alpha_{i,j}} \leq O(\frac{1}{\sqrt{d_1}})$ *for all* $i$. *Moreover, for every spuriously correlated pair* $(j, j')$ *satisfying Assumption 3.5, there exists at least* $\Omega(1)$ *many* $i \in [m]$ *such that* $N_i = \{j, j'\}$.

*Remark* 4.4. Theorem 4.3 indicates that the model learned by ITCP on raw data only achieves some level of feature alignment. Specifically, a neuron pair $(\bar{w}_i, \bar{v}_i)$ learns a mixture of image features and text features in $N_i$, respectively. Due to spurious correlations in $(j, j')$, $\mathbf{M}_j$ and $\mathbf{M}_{j'}$ are always mixed together, as are $\mathbf{H}_j$ and $\mathbf{H}_{j'}$. Consequently, the model with weights $\overline{\mathbf{W}}$ and $\overline{\mathbf{V}}$ cannot yet achieve the desired feature alignment of $(\mathbf{M}_j, \mathbf{H}_j)$.

We next show the feature alignment is enhanced in the model by ITCP using synthetic data.

**Theorem 4.5** (**Successful feature alignment when pre–training on synthetic data** $\tilde{S}$). *Each neuron pair* $(\tilde{w}_i, \tilde{v}_i)$ *in* $(\widetilde{\mathbf{W}}, \widetilde{\mathbf{V}})$ *($i \in [m]$) mainly learn the features in the set* $\tilde{N}_i \subseteq [d]$, *while ignoring other features, i.e.,*

$$\tilde{w}_i = \sum_{j \in \tilde{N}_i} \tilde{\alpha}_{i,j} \mathbf{M}_j + \sum_{j \in [d] \setminus \tilde{N}_i} \tilde{\beta}_{i,j} \mathbf{M}_j + \sum_{j \in [d_1] \setminus [d]} \tilde{\gamma}_{i,j} \mathbf{M}_j^\perp$$

$$\tilde{v}_i = \sum_{j \in \tilde{N}_i} \tilde{\alpha}_{i,j} \mathbf{H}_j + \sum_{j \in [d] \setminus \tilde{N}_i} \tilde{\beta}_{i,j} \mathbf{H}_j + \sum_{j \in [d_1] \setminus [d]} \tilde{\gamma}_{i,j} \mathbf{H}_j^\perp$$

$$(18)$$

*where* $\tilde{\alpha}_{i,j}^2 = \Theta(1)(\|\tilde{w}_i\|_2^2 + \|\tilde{v}_i\|_2^2)$, $\frac{\tilde{\beta}_{i,j}}{\tilde{\alpha}_{i,j}} \leq O(\frac{1}{\sqrt{d}})$ *and* $\frac{\tilde{\gamma}_{i,j}}{\tilde{\alpha}_{i,j}} \leq O(\frac{1}{\sqrt{d_1}})$. *Moreover, for every* $j \in [d]$, *there exists at least* $\Omega(1)$ *many* $i \in [m]$ *such that* $\tilde{N}_i = \{j\}$.

*Remark* 4.6. Theorem 4.5 differs from Theorem 4.3 mainly in the last sentence. For every image and text feature pair $(\mathbf{M}_j, \mathbf{H}_j)$, there exists at least one neuron pair $(\tilde{w}_i, \tilde{v}_i)$ such that $(\tilde{w}_i, \tilde{v}_i)$ primarily learn $(\mathbf{M}_j, \mathbf{H}_j)$, while the contributions of other features are order-wise smaller. As a result, the model with weights $\widetilde{\mathbf{W}}$ and $\widetilde{\mathbf{V}}$ achieves the desired feature alignment.

## 4.3. Performance Comparison on Downstream Tasks

We next compare the performance of the models $(f_{\overline{\mathbf{W}}}, g_{\overline{\mathbf{V}}})$ and $(f_{\widetilde{\mathbf{W}}}, g_{\widetilde{\mathbf{V}}})$ on the zero-shot image classification problem with out-of-domain data described in Sections 2.2 and 3.4.

**Theorem 4.7** (Zero-Shot Image Classification). *For the* $K$-*class image classification problem with a setup in Section 3.4, the model* $(f_{\overline{\mathbf{W}}}, g_{\overline{\mathbf{V}}})$ *from ITCP using raw data has a constant failure probability:*

$$\Pr\left(\arg\max_{k \in [K]} \langle f_{\overline{\mathbf{W}}}(x), g_{\overline{\mathbf{V}}}(y_k) \rangle = l_x \right) = 1 - \Theta(1); . \quad (19)$$

*In contrast, the model* $(f_{\widetilde{\mathbf{W}}}, g_{\widetilde{\mathbf{V}}})$ *from ITCP using synthetic data succeeds with high probability:*

$$\Pr\left(\arg\max_{k \in [K]} \langle f_{\widetilde{\mathbf{W}}}(x), g_{\widetilde{\mathbf{V}}}(y_k) \rangle = l_x \right) = 1 - o(1). \quad (20)$$

*Remark* 4.8. Theorem 4.7 first demonstrates that the zero-shot performance of $(f_{\overline{\mathbf{W}}}, g_{\overline{\mathbf{V}}})$ is unsatisfactory. This is due to the lack of accurate feature alignment in $(f_{\overline{\mathbf{W}}}, g_{\overline{\mathbf{V}}})$, as established in Theorem 4.3. Theorem 4.7 further shows that $(f_{\widetilde{\mathbf{W}}}, g_{\widetilde{\mathbf{V}}})$ achieves accurate classification. This success is attributed to the precise feature alignment in $(f_{\widetilde{\mathbf{W}}}, g_{\widetilde{\mathbf{V}}})$, as described in Theorem 4.5. Note that Theorem 4.7 holds for image data with a distribution shift from the training data.

## 5. Experiment

### 5.1. Simulated Experiment

**Experiment setup.** We first evaluate our results through simulated experiments, applying the same training frame-

work in Section 2.1. We simulate a more general model for spurious correlation than Assumption 3.5, where every $\mathbf{M}_j$ can be spuriously correlated with any text feature $\mathbf{H}_{j'}$ for any $j' \neq j$, while keeping the total conditional probability of spurious correlation in the raw data as $C_s$ in (9).

We set $d_1 = 2500$, $d = 50$, $|S_w| = 5000$, and $|S_h| = 1000$. $\mathbf{M}$ and $\mathbf{H}$ are generated from a standard Gaussian distribution and subsequently processed using QR decomposition to ensure orthonormality. $z_x$ follows a Bernoulli distribution with parameter $p = 0.1$, and the noise variance is set to $\sigma_\xi^2 = 1/d$. The learner model employs $m = 80$ neurons. The batch size for SGD is 500, and the step size is 0.001. The learned encoders are evaluated on a downstream 5-class classification task,, where the downstream data is generated with $z_x$ following a Bernoulli distribution with parameter $p = 0.2$. The vectors $z_{y_k}$ are constructed by partitioning the feature space $d$ into $K$ disjoint segments. All results are averaged over 20 independent experiments. $(\overline{\mathbf{W}}, \overline{\mathbf{V}})$ and $(\widetilde{\mathbf{W}}, \widetilde{\mathbf{V}})$ denote the model weights learned using raw data and filtered data, respectively.

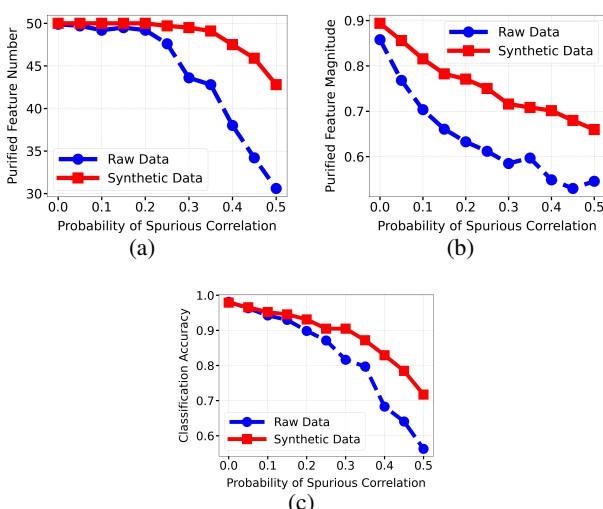

Figure 1: Performance comparison of ITCP on raw data and synthetic data when the probability of spurious correlation $C_s$ changes. (a) Number of features that have purified representation in the model (b) Average magnitude of purified presentations (c) Zero-shot out-of-domain classification accuracy.

**Improved feature representation using synthetic captions.** A weight $\bar{w}_i$ learns a *purified representation* of $\mathbf{M}_j$ if its projection along $\mathbf{M}_j$ achieves the largest magnitude and satisfies $|\langle \bar{w}_i, \mathbf{M}_j \rangle|/\|\bar{w}_i\| > 0.5$. The same applies to $(\widetilde{\mathbf{W}}, \widetilde{\mathbf{V}})$. Figure 1(a) shows the number of features $\mathbf{M}_j$ (out of $d = 50$ total features) for which at least one neuron in $\overline{\mathbf{W}}$ (or $\widetilde{\mathbf{W}}$, respectively) learns a purified representation. The results indicate that ITCP applied to synthetic data can learn purified representations for almost all features, even when the spurious correlation probability $C_s$ is as high as 0.3. In contrast, ITCP on raw data shows degraded performance,

with its ability to learn purified representations decreasing more rapidly as $C_s$ increases. This observation is consistent with Theorems 4.3 and 4.5, as well as Remark 4.6.

**Larger magnitude of purified representation using synthetic data.** Figure 1(b) presents the average of the largest magnitude of projections over neurons that learn purified representations. The largest projection magnitude achieved by $\widetilde{\mathbf{W}}$ (from ITCP on synthetic data) is higher than that achieved by $\overline{\mathbf{W}}$, further demonstrating that $\widetilde{\mathbf{W}}$ learns better-purified representations.

**Improved zero-shot out-of-domain performace using synthetic data.** Figure 1(c) compares the classification accuracy of both models on zero-shot out-of-domain data. The model trained on synthetic data consistently outperforms the one trained on raw data, with the performance gap widening as spurious correlations in the raw data increase.

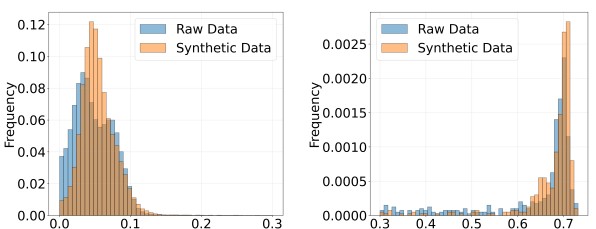

Figure 2: Histogram of $|\langle \bar{v}_i, \mathbf{H}_j \rangle|/\|\bar{v}_i\|$ for ITCP on raw data and $|\langle \tilde{v}_i, \mathbf{H}_j \rangle|/\|\tilde{v}_i\|$ for ITCP on synthetic data (split into two figures to highlight the significant differences in the value distributions).

**Neurons trained on synthetic data exhibit a more concentrated distribution.** Figure 2 visualizes the histograms of $|\langle \bar{v}_i, \mathbf{H}_j \rangle|/\|\bar{v}_i\|$ and $|\langle \tilde{v}_i, \mathbf{H}_j \rangle|/\|\tilde{v}_i\|$ for all $i \in [m]$ and $j \in [d]$. The values of $|\langle \tilde{v}_i, \mathbf{H}_j \rangle|/\|\tilde{v}_i\|$ are more concentrated, typically around 0.05 and 0.7. In contrast, the values for $|\langle \bar{v}_i, \mathbf{H}_j \rangle|/\|\bar{v}_i\|$ are less concentrated. This phenomenon is consistent with Theorem 4.5, which indicates that for every $\mathbf{H}_j$, certain neurons $\tilde{v}_i$ in $\widetilde{\mathbf{V}}$ predominately learns $\mathbf{H}_j$. In such cases, $|\langle \tilde{v}_i, \mathbf{H}_j \rangle|$ approaches 1, while $|\langle \tilde{v}_i, \mathbf{H}_{j'} \rangle|/\|\tilde{v}_i\|$ approaches 0 for $j' \neq j$. The concentrated values of 0.05 and 0.7 observed in Figure 2 are due to noise in the data. In contrast, feature alignment is less significant for $\overline{\mathbf{V}}$, leading to less concentration of the corresponding values. Similar results are obtained for $|\langle w_i, \mathbf{M}_j \rangle|$, deferred to Figure 6 in the appendix.

**Enhanced class separation of downstream tasks by ITCP with synthetic data**. Figure 3 visualizes the t-distributed stochastic neighbor embedding (t-SNE) of the feature embeddings generated by the two models, computed as $f_{\overline{\mathbf{W}}}(x_p)$ and $f_{\widetilde{\mathbf{W}}}(x_p)$ for each $x_p$, respectively. The t-SNE method projects the high-dimensional embeddings onto a two-dimensional map. One can see that the embeddings from different groups are more distinctly separated in the model trained using ITCP on synthetic data, indicating that this approach achieves better feature alignment.

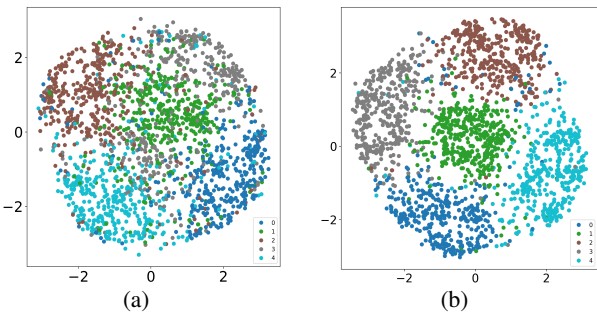

(a)                            (b)

Figure 3: t-SNE visualization of feature embeddings (probability of spurious correlation $C_s = 0.3$. (a) ITCP on raw data. (b) ITCP on synthetic data.

## 5.2. Experiments on Practical Data and Models

BLIP ViT-B/16, with 200M parameters (Li et al., 2022a), is a vision-language model pre-trained on 14M images, including a small amount of high-quality human-annotated datasets and a large amount of noisy web image-text pairs. It is an enhanced version of the earlier ALBEF model (Li et al., 2021), sharing the same model architecture and training data. The key difference is that ALBEF is trained using ITCP with raw data, whereas BLIP is trained using ITCP with synthetic caption data. Extensive experimental results in (Li et al., 2022a) demonstrate BLIP's improved generalization performance over ALBEF due to the use of synthetic text. Thus, we do not repeat those efforts here.

The BLIP/ALBEF model employs transformer-based architectures for both its image and text encoders, incorporating an image-text similarity loss. Both the image and text encoders consist of 12-layer transformers that output features in $\mathbb{R}^{768}$. These features are then passed through linear projection layers, mapping them into a shared low-dimensional embedding space in $\mathbb{R}^{256}$, respectively.

**Enhanced separation of images in the COCO dataset by BLIP**. We select 10 classes from the COCO dataset (Lin et al., 2014) and visualize the image feature embeddings in $\mathbb{R}^{256}$ produced by the BLIP and ALBEF models. Figure 4 compares the t-SNE visualizations of these embeddings. BLIP's embeddings exhibit much clearer separation among different image classes, whereas ALBEF's embeddings often show overlaps between classes, such as "sports ball" and "tennis racket." This observation further verifies our theoretical result that synthetic data improves feature alignment.

**Enhanced text feature representation learning by BLIP**. The final text projection layer in BLIP/ALBEF consists of 256 neurons and bears functional similarity to $\mathbf{V}$ in our simplified model in (6). Each neuron associated with a weight vector $v_i \in \mathbb{R}^{768}$. Figure 5 (a) presents a histogram of the normalized inner products $\langle v_j, v_{j'} \rangle / (\|v_j\| \|v_{j'}\|)$ for all $j, j' \in \{1, 2, \ldots, 256\}$ in BLIP and ALBEF. The values

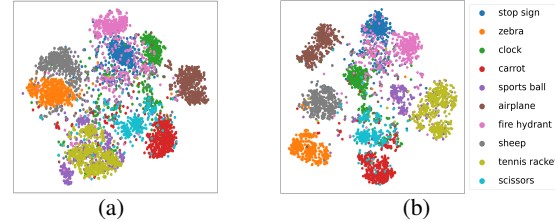

(a)                            (b)

Figure 4: t-SNE visualization of image feature embeddings of 10 classes in COCO: (a) ALBEF (b) BLIP

in BLIP are more concentrated around zero compared to ALBEF, indicating that the weight vectors exhibit a higher degree of orthogonality. This result aligns with Theorem 4.5, which suggests that training with synthetic data allows certain neurons to specialize in learning individual features from a set of orthogonal features, leading to a stronger concentration of values around zero. Similar results on image features are shown in Figure 7 in the Appendix.

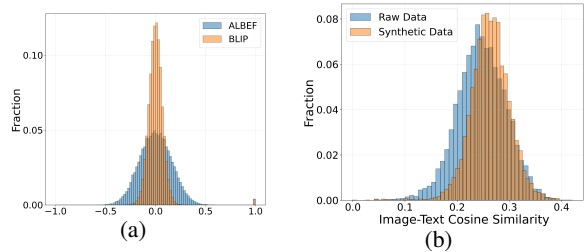

(a)                            (b)

Figure 5: (a) Histogram of $\langle v_j, v_{j'} \rangle / (\|v_j\| \|v_{j'}\|)$ of the text linear projection layer of ALBEF and BLIP (b) Histogram of image-text cosine of raw web data and BLIP generated synthetic data

**Enhanced image-text alignment on synthetic captions.** We randomly sample 1% of three raw web datasets: Conceptual Captions (Sharma et al., 2018), Conceptual 12M (Changpinyo et al., 2021), and SBU Captions (Ordonez et al., 2011), which are used for training both BLIP and ALBEF. We compute the cosine similarity between the image and its corresponding raw web caption, as well as the cosine similarity between the image and its BLIP-generated synthetic caption. Figure 5 (b) presents the histogram of cosine similarities for these two cases. BLIP-generated captions achieve significantly higher image-text cosine similarity, with a mean similarity of 0.26 compared to 0.24 for raw captions.

## 6. Conclusion

This paper presents a comprehensive theoretical analysis of the training dynamics of contrastive learning in VLMs with nonlinear functions. Our theoretical findings highlight the crucial role of synthetic captions in enhancing feature alignment and zero-shot performance, which we validate on practical models and datasets. Future directions including analzying contrastive learning on Transformer architectures and exploring more complex tasks, such as image-text retrieval and visual question answering.

## Impact Statement

This paper aims to theoretical analysis of VLM and synthetic text captions. No potential societal consequences are associated with our work.

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

## A. Preliminaries

We first restate some important notations used in the Appendix, which are summarized in Table 2.

Table 2: Summary of Notations

| Notations | Annotation |
|---|---|
| $\mathbf{M} \in \mathbb{R}^{d_1 \times d}, \mathbf{H} \in \mathbb{R}^{d_1 \times d}$ | $\mathbf{M}$ is the image dictionary matrix, $\mathbf{H}$ is the text dictionary matrix. |
| $\mathbf{W} \in \mathbb{R}^{m \times d_1}, \mathbf{V} \in \mathbb{R}^{m \times d_1}$ | $\mathbf{W}$ is the weight of image encoder, $\mathbf{V}$ is the weight of text encoder. |
| $x_p \in \mathbb{R}^{d_1}, y_p \in \mathbb{R}^{d_1}$ | $x_p$ and $y_p$ represent an image and a text data, respectively. |
| $z_{x_p}, z_{y_p} \in \mathbb{R}^d$ | $z_{x_p}$ and $z_{y_p}$ are the sparse signals of image and text, respectively. $z_{y_k}$ is the sparse signal for the text prompt $y_k$. |
| $z_{x_p}^j, z_{y_p}^j$ | $z_{x_p}^j$ is the $j$-th coordinate of $z_{x_p}$; $z_{y_p}^j$ is the $j$-th coordinate of $z_{y_p}$. |
| $L, L_C$ | $L$ is the loss for ITCP; $L_C$ is the loss for Image-grounded Text Decoding. |
| $S = S_h \cup S_w$ | $S_w$ is the noisy web low-quality dataset; $S_h$ is the human-annotated high-quality dataset. |
| $\tilde{S} = S_h \cup \tilde{S}_w$ | $\tilde{S}_w$ replaces noisy captions in $S_w$ with synthetic captions. |
| $T_1$ | Stage I of ITCP with $b_i^{(t)} = 0$. |
| $T_2$ | Stage II of ITCP with $b_i^{(t+1)} = (1 + \frac{\eta}{d})b_i^{(t)}$. |
| $T_3$ | Stage III of ITCP with $b_i^{(t+1)} = b_i^{(T_2)}$. |
| $T_C$ | Stage of training caption generators. |
| $\mathcal{S}_{j,\text{sure}}$ | The set of well-initialized neurons $(w_i, v_i)$ on features $(\mathbf{M}_j, \mathbf{H}_j)$. |

### A.1. Feature Coupling and Expected Values in $S_w$

**Assumption A.1** (High and low quality pairs). The high-quality image-text pairs in $S_h$ have size $|S_h| = \Theta(d^2)$. The low-quality image-text pairs in $S_w$ have size $|S_w| = \text{poly}(d)| \gg \omega(d^2)$

In $S_h$, for a positive pair $(x_p, y_p)$, we assume perfect alignment, meaning $z_{x_p} = z_{y_p}$. Consequently, the following holds:

$$\mathbb{E}\left[z_{x_p}^j z_{y_p}^j\right] = \frac{C_z}{d}, \quad \mathbb{E}\left[z_{x_p}^j z_{y_p}^{j'}\right] = \Theta\left(\frac{1}{d^2}\right), \quad j' \neq j \tag{21}$$

To model the misaligned features in low-quality pairs in $S_w$, where spurious misalignment occurs at a non-negligible level, we assume $[d]$ can be divided into $d/2$ disjoint sets, each containing exactly two entries. Let $(j, j') \subset [d]$ denote one such set, referred to as a "spuriously correlated set." The following assumptions capture the nature of spurious and true alignments:

$$\Pr(|z_{y_p}^{j'}| = 1 \mid |z_{x_p}^j| = 1) = \Theta(1) < \frac{1}{2},$$
$$\Pr(|z_{y_p}^{j'}| = 1 \mid |z_{x_p}^j| = 1) + \Pr(|z_{y_p}^j| = 1 \mid |z_{x_p}^j| = 1) = 1. \tag{22}$$

These assumptions imply that true alignment dominates, with $\Pr(|z_{y_p}^j| = 1 \mid |z_{x_p}^j| = 1) > \frac{1}{2}$, while spurious alignment exists at a constant percentage level, making it non-negligible. The intuition behind this assumption is that each feature $j$ is paired with exactly one spuriously correlated feature $j'$, ensuring that $j$ is not associated with any other feature $j'' \neq j'$. This design simplifies the analysis while effectively capturing the key challenges posed by low-quality data.

Then, for a positive pair $(x_p, y_p)$ with $p$ in $S_w$, we have:

$$\mathbb{E}\left[z_{x_p}^j z_{y_p}^j\right] + \mathbb{E}\left[z_{x_p}^j z_{y_p}^{j'}\right] = \frac{C_z}{d},$$
$$\mathbb{E}\left[z_{x_p}^j z_{y_p}^{j'}\right] = \Theta\left(\frac{1}{d}\right) < \frac{C_z}{2d}. \tag{23}$$

where $(j, j')$ is a spuriously correlated set.

For negative pairs $(x_p, y_q)$, where $p \neq q$, and $p, q \in S$, we have:

$$\mathbb{E}\left[z_{x_p}^j z_{y_q}^{j'}\right] = \Theta\left(\frac{1}{d^2}\right), \quad \forall j, j' \in [d]. \tag{24}$$

In $S_w$, mismatched text and image pairs are prevalent compared to $S_h$. For a postive pair $(x_p, y_p)$, we assume $\frac{\log(1/c_0)}{2\log d} <$ $\Pr(|z_{y_p}^{j'}| = 1 \mid |z_{x_p}^{j}| = 1) < \frac{1}{2}$. To model this, we assume that for each primary feature $j \in [d]$, there exists exactly one spurious feature $j'$ such that $j$ and $j'$ are uniquely coupled. This implies that $j$ cannot be associated with any other feature $j'' \neq j'$. Mathematically, the coupling is defined as:

$$\Pr(|z_{y_p}^{j'}| = 1 \mid |z_{x_p}^{j}| = 1) + \Pr(|z_{y_p}^{j}| = 1 \mid |z_{x_p}^{j}| = 1) = 1. \tag{25}$$

For a positive pair $(x_p, y_p)$ in $S_w$, the probabilities of spurious and aligned features are further constrained:

$$\frac{\log(1/c_0)}{2\log d} < \Pr(|z_{y_p}^{j'}| = 1 \mid |z_{x_p}^{j}| = 1) < \frac{1}{2}, \tag{26}$$

The lower bound is established in Lemma B.9.

and:

$$\Pr(|z_{y_p}^{j}| = 1 \mid |z_{x_p}^{j}| = 1) = 1 - \Pr(|z_{y_p}^{j'}| = 1 \mid |z_{x_p}^{j}| = 1). \tag{27}$$

Under these assumptions, the expected values for the aligned and spurious features are calculated as follows:

For the aligned feature $j$, we have:

$$\mathbb{E}\left[z_{x_p}^{j} z_{y_p}^{j}\right] = \Pr(|z_{y_p}^{j}| = 1, |z_{x_p}^{j}| = 1) = \Pr(|z_{y_p}^{j}| = 1 \mid |z_{x_p}^{j}| = 1) \cdot \Pr(|z_{x_p}^{j}| = 1) = \Pr(|z_{y_p}^{j}| = 1 \mid |z_{x_p}^{j}| = 1) \cdot \frac{C_z}{d}. \tag{28}$$

For the spurious feature $j'$, we have:

$$\mathbb{E}\left[z_{x_p}^{j} z_{y_p}^{j'}\right] = \Pr(|z_{y_p}^{j'}| = 1, |z_{x_p}^{j}| = 1) = \Pr(|z_{y_p}^{j'}| = 1 \mid |z_{x_p}^{j}| = 1) \cdot \Pr(|z_{x_p}^{j}| = 1) = \Pr(|z_{y_p}^{j'}| = 1 \mid |z_{x_p}^{j}| = 1) \cdot \frac{C_z}{d}. \tag{29}$$

The total expected value across both aligned and spurious features satisfies:

$$\mathbb{E}\left[z_{x_p}^{j} z_{y_p}^{j}\right] + \mathbb{E}\left[z_{x_p}^{j} z_{y_p}^{j'}\right] = \frac{C_z}{d} \tag{30}$$

Here, $j'$ denotes the spurious feature associated with $j$.

### A.2. Gradient

The contrastive loss in vision-language models (VLM) is defined as follows:

$$L(f^{(t)}, h^{(t)}) = \sum_{p \in S} \left[ -\langle f^{(t)}(x_p), h^{(t)}(y_p) \rangle + \sum_{x_n \in \mathfrak{N}'} \frac{\left(\langle f^{(t)}(x_n), h^{(t)}(y_p) \rangle\right)^2}{2\tau} + \sum_{y_n \in \mathfrak{N}'} \frac{\left(\langle f^{(t)}(x_p), h^{(t)}(y_n) \rangle\right)^2}{2\tau} \right], \tag{31}$$

where $\tau > 0$ is a temperature parameter.

We perform stochastic gradient descent (SGD) on this contrastive loss. Let $f^{(t)}$ and $h^{(t)}$ be the image encoder and text encoder networks at iteration $t$, respectively. The network parameters are updated as follows:

$$w_i^{(t+1)} \leftarrow w_i^{(t)} - \eta \nabla_{w_i} L(f^{(t)}, h^{(t)}), \tag{32}$$

$$v_i^{(t+1)} \leftarrow v_i^{(t)} - \eta \nabla_{v_i} L(f^{(t)}, h^{(t)}), \tag{33}$$

where $b_i^{(t)}$, the bias term, is manually tuned during training and thus excluded from gradient updates.

The gradient of $L(f^{(t)}, h^{(t)})$ with respect to $w_i^{(t)}$ at iteration $t$ is given by:

$$
\begin{aligned}
\nabla_{w_i} L(f^{(t)}, h^{(t)}) = & - \langle v_i^{(t)}, y_p \rangle x_p \cdot \mathbf{1}_{\left| \langle w_i^{(t)}, x_p \rangle \right| \geq b_i^{(t)}} \cdot \mathbf{1}_{\left| \langle v_i^{(t)}, y_p \rangle \right| \geq b_i^{(t)}} \\
& + \sum_{x_n \in \mathfrak{N}} \frac{\langle f^{(t)}(x_n), h^{(t)}(y_p) \rangle \langle v_i^{(t)}, y_p \rangle x_n}{\tau} \cdot \mathbf{1}_{\left| \langle w_i^{(t)}, x_n \rangle \right| \geq b_i^{(t)}} \cdot \mathbf{1}_{\left| \langle v_i^{(t)}, y_p \rangle \right| \geq b_i^{(t)}} \\
& + \sum_{y_n \in \mathfrak{N}} \frac{\langle f^{(t)}(x_p), h^{(t)}(y_n) \rangle \langle v_i^{(t)}, y_n \rangle x_p}{\tau} \cdot \mathbf{1}_{\left| \langle w_i^{(t)}, x_p \rangle \right| \geq b_i^{(t)}} \cdot \mathbf{1}_{\left| \langle v_i^{(t)}, y_n \rangle \right| \geq b_i^{(t)}}.
\end{aligned}
\tag{34}
$$

Similarly, the empirical gradient of $L(f^{(t)}, h^{(t)})$ with respect to $v_i^{(t)}$ is:

$$
\begin{aligned}
\nabla_{v_i} L(f^{(t)}, h^{(t)}) = & - \langle w_i^{(t)}, x_p \rangle y_p \cdot \mathbf{1}_{\left| \langle w_i^{(t)}, x_p \rangle \right| \geq b_i^{(t)}} \cdot \mathbf{1}_{\left| \langle v_i^{(t)}, y_p \rangle \right| \geq b_i^{(t)}} \\
& + \sum_{x_n \in \mathfrak{N}} \frac{\langle f^{(t)}(x_n), h^{(t)}(y_p) \rangle \langle w_i^{(t)}, x_n \rangle y_p}{\tau} \cdot \mathbf{1}_{\left| \langle w_i^{(t)}, x_n \rangle \right| \geq b_i^{(t)}} \cdot \mathbf{1}_{\left| \langle v_i^{(t)}, y_p \rangle \right| \geq b_i^{(t)}} \\
& + \sum_{y_n \in \mathfrak{N}} \frac{\langle f^{(t)}(x_p), h^{(t)}(y_n) \rangle \langle w_i^{(t)}, x_p \rangle y_n}{\tau} \cdot \mathbf{1}_{\left| \langle w_i^{(t)}, x_p \rangle \right| \geq b_i^{(t)}} \cdot \mathbf{1}_{\left| \langle v_i^{(t)}, y_n \rangle \right| \geq b_i^{(t)}}.
\end{aligned}
\tag{35}
$$

### A.3. Alignment Updates

We analyze how each neuron $i \in [m]$ aligns with the feature $\mathbf{M}_j$ during each iteration of SGD. The alignment can be described by the following update rule:

$$
\begin{aligned}
\langle w_i^{(t+1)}, \mathbf{M}_j \rangle &= \langle w_i^{(t)}, \mathbf{M}_j \rangle - \langle \nabla_{w_i} L(f^{(t)}, h^{(t)}), \mathbf{M}_j \rangle \\
&= \langle w_i^{(t)}, \mathbf{M}_j \rangle + \eta z_x^j z_y^j \langle v_i^{(t)}, \mathbf{H}_j \rangle + \eta z_x^j z_y^{j'} \langle v_i^{(t)}, \mathbf{H}_{j'} \rangle \pm Err_t.
\end{aligned}
\tag{36}
$$

Similarly, for $\langle v_i^{(t+1)}, \mathbf{H}_j \rangle$, the update rule becomes:

$$
\begin{aligned}
\langle v_i^{(t+1)}, \mathbf{H}_j \rangle &= \langle v_i^{(t)}, \mathbf{H}_j \rangle - \langle \nabla_{v_i} L(f^{(t)}, h^{(t)}), \mathbf{H}_j \rangle \\
&= \langle v_i^{(t)}, \mathbf{H}_j \rangle + \eta z_x^j z_y^j \langle w_i^{(t)}, \mathbf{M}_j \rangle + \eta z_x^j z_y^{j'} \langle w_i^{(t)}, \mathbf{M}_{j'} \rangle \pm Err_t.
\end{aligned}
\tag{37}
$$

Using Lemma B.7, we know that with high probability, $\sum_{x_n \in \mathfrak{N}} \frac{\langle f^{(t)}(x_n), h^{(t)}(y_p) \rangle}{\tau} \leq O(\frac{1}{d})$, so in Eq (34) and Eq (34) the sum of second term and third term is always less than the first term, until $\langle f^{(t)}(x_n), h^{(t)}(y_p) \rangle = \Theta(d)$.

The updates for the components $\langle w_i^{(t+1)}, \mathbf{M}_j \rangle$, $\langle v_i^{(t+1)}, \mathbf{H}_j \rangle$, $\langle w_i^{(t+1)}, \mathbf{M}_{j'} \rangle$, and $\langle v_i^{(t+1)}, \mathbf{H}_{j'} \rangle$ (where $j'$ represents the spurious aligned feature corresponding to $j$) can be expressed concisely in matrix form as follows:

$$
\begin{bmatrix}
\langle w_i^{(t+1)}, \mathbf{M}_j \rangle \\
\langle v_i^{(t+1)}, \mathbf{H}_j \rangle \\
\langle w_i^{(t+1)}, \mathbf{M}_{j'} \rangle \\
\langle v_i^{(t+1)}, \mathbf{H}_{j'} \rangle
\end{bmatrix}
=
\begin{bmatrix}
a & b & 0 & c \\
b & a & c & 0 \\
0 & c & a & b \\
c & 0 & b & a
\end{bmatrix}
\begin{bmatrix}
\langle w_i^{(t)}, \mathbf{M}_j \rangle \\
\langle v_i^{(t)}, \mathbf{H}_j \rangle \\
\langle w_i^{(t)}, \mathbf{M}_{j'} \rangle \\
\langle v_i^{(t)}, \mathbf{H}_{j'} \rangle
\end{bmatrix}
\pm Err_t,
\tag{38}
$$

where the coefficients are defined as:

$$
a = 1, \quad b = z_x^j z_y^j \cdot \mathbf{1}_{\left| \langle w_i^{(t)}, x_p \rangle \right| \geq b_i^{(t)}} \cdot \mathbf{1}_{\left| \langle v_i^{(t)}, y_p \rangle \right| \geq b_i^{(t)}},
$$

$$
c = z_x^j z_y^{j'} \cdot \mathbf{1}_{\left| \langle w_i^{(t)}, x_p \rangle \right| \geq b_i^{(t)}} \cdot \mathbf{1}_{\left| \langle v_i^{(t)}, y_p \rangle \right| \geq b_i^{(t)}}.
$$

Therefore, we have

$$\langle w_i^{(t)}, \mathbf{M}_j \rangle = \langle v_i^{(t)}, \mathbf{H}_j \rangle = \frac{(a+b+c)^t + (a+b-c)^t}{4} \left( \langle w_i^{(0)}, \mathbf{M}_j \rangle + \langle v_i^{(0)}, \mathbf{H}_j \rangle \right)$$
$$+ \frac{(a+b+c)^t - (a+b-c)^t}{4} \left( \langle w_i^{(0)}, \mathbf{M}_{j'} \rangle + \langle v_i^{(0)}, \mathbf{H}_{j'} \rangle \right) \tag{39}$$

and

$$\langle w_i^{(t)}, \mathbf{M}_{j'} \rangle = \langle v_i^{(t)}, \mathbf{H}_{j'} \rangle = \frac{(a+b+c)^t + (a+b-c)^t}{4} \left( \langle w_i^{(0)}, \mathbf{M}_{j'} \rangle + \langle v_i^{(0)}, \mathbf{H}_{j'} \rangle \right)$$
$$+ \frac{(a+b+c)^t - (a+b-c)^t}{4} \left( \langle w_i^{(0)}, \mathbf{M}_j \rangle + \langle v_i^{(0)}, \mathbf{H}_j \rangle \right) \tag{40}$$

This matrix representation highlights the interactions between the alignment of true and spurious features during SGD updates. The diagonal elements $a$ dominate the contribution from existing alignments, while the off-diagonal terms $b, c$ capture the mutual influence between paired features and spurious alignments. Note that if $c$ is very small, it indicates that the spurious alignment ($j'$) has minimal influence, allowing $w_i$ to focus on learning purified features. Conversely, if $c$ is large, the spurious alignment could significantly interfere with the learning process, hindering the purification of features. The error term $Err_t$ accounts for higher-order noise or unmodeled effects in the update process.

## B. Technical Lemmas

**Definition B.1** (Neuron Characterization). Let us define a few notations to characterize each neuron $w_i^{(t)}$'s behavior. For every constant $c_0 \in (0, 1)$ and $\gamma \in (0, 0.1)$, by choosing $c_1 = 2 + 2(1 - \gamma)c_0$ and $c_2 = \gamma c_0$, we define:

1. Let $\mathcal{S}_{j,\text{sure}}^{(t)} \subseteq [m]$ be those neurons $i \in [m]$ satisfying

- $(\frac{1}{n} \sum_{i=1}^{n} \langle w_i^{(t)}, \mathbf{M}_j \rangle)^2 \geq \frac{(c_1 + c_2) \log d}{d} \|\mathbf{M}\mathbf{M}^\top w_i^{(t)}\|_2^2$
- $(\frac{1}{n} \sum_{i=1}^{n} \langle w_i^{(t)}, \mathbf{M}_{j'} \rangle)^2 < \frac{(c_1 - c_2) \log d}{d} \|\mathbf{M}\mathbf{M}^\top w_i^{(t)}\|_2^2$

2. Let $\mathcal{S}_{j,\text{pot}}^{(t)} \subseteq [m]$ be those neurons $i \in [m]$ satisfying

- $\langle w_i^{(t)}, \mathbf{M}_j \rangle^2 \geq \frac{(c_1 - c_2) \log d}{d} \|\mathbf{M}\mathbf{M}^\top w_i^{(t)}\|_2^2$

**Lemma B.2** (Geometry at initialization). *We initialize the parameters by $w_i^{(0)} \sim \mathcal{N}(0, \sigma_0^2 \mathbf{I}_{d_1})$, where $\sigma_0^2 = \Theta\left(\frac{1}{d_1 poly(d)}\right)$. We have with probability $\geq 1 - o(1/d^3)$ over the random initialization, for all $j \in [d]$:*

$$\left| \mathcal{S}_{j,sure}^{(0)} \right| = \Omega\left( d^{\frac{\gamma}{4} c_0} \right) =: \Xi_1$$
$$\left| \mathcal{S}_{j,pot}^{(0)} \right| \leq O\left( d^{2\gamma c_0} \right) =: \Xi_2$$

*Proof.* If $g$ is standard Gaussian, then for every $t > 0$,

$$\frac{1}{\sqrt{2\pi}} \frac{(t)}{t^2 + 1} e^{-t^2/2} < \Pr_{g \sim \mathcal{N}(0,1)}[g > t] < \frac{1}{\sqrt{2\pi}} \frac{1}{(t)} e^{-t^2/2}. \tag{41}$$

We initialize the parameters by $w_i^{(0)} \sim \mathcal{N}(0, \sigma_0^2 \mathbf{I}_{d_1})$, where $\sigma_0^2 = \Theta\left(\frac{1}{d_1 \text{poly}(d)}\right)$. We have $\frac{1}{n} \sum_{i=1}^{n} \langle w_i^{(0)}, \mathbf{M}_i \rangle \sim \mathcal{N}\left(0, \frac{\sigma_0^2}{n}\right)$.

Therefore, for every $i \in m$ and $j \in d$, we have

$$p_1 = \Pr\left[ (\frac{1}{n} \sum_{i=1}^{n} \langle w_i^{(0)}, \mathbf{M}_j \rangle)^2 \geq (c_1 + c_2) \frac{\sigma_0^2}{n} \log d \right] = \Theta\left(\frac{1}{\log d}\right) \cdot \frac{1}{d^{(c_1+c_2)/2}} = \Theta\left(\frac{1}{\sqrt{\log d}}\right) \cdot \frac{1}{d \cdot d^{(1-\gamma/2)c_0}} \tag{42}$$

$$p_2 = \Pr\left[ (\frac{1}{n} \sum_{i=1}^{n} \langle w_i^{(0)}, \mathbf{M}_{j'} \rangle)^2 \geq (c_1 - c_2) \frac{\sigma_0^2}{n} \log d \right] = \Theta\left(\frac{1}{\log d}\right) \cdot \frac{1}{d^{(c_1-c_2)/2}} = \Theta\left(\frac{1}{\sqrt{\log d}}\right) \cdot \frac{1}{d \cdot d^{(1-3\gamma/2)c_0}} \tag{43}$$

Let $\mathcal{S}^{(0)}_{j,\text{sure}} \subseteq [m]$ be those neurons $i \in [m]$ satisfying

- $(\frac{1}{n} \sum_{i=1}^{n} \langle w_i^{(0)}, \mathbf{M}_j \rangle)^2 \geq \frac{(c_1+c_2)\log d}{d} \|\mathbf{MM}^\top w_i^{(0)}\|_2^2$
- $(\frac{1}{n} \sum_{i=1}^{n} \langle w_i^{(0)}, \mathbf{M}_{j'} \rangle)^2 < \frac{(c_1-c_2)\log d}{d} \|\mathbf{MM}^\top w_i^{(0)}\|_2^2$

By concentration with respect to all $m$ choices of $i \in [m]$, we know with probability at least $1 - o\left(\frac{1}{d^3}\right)$ it satisfies $\left| \mathcal{S}^{(0)}_{j,\text{sure}} \right| = \Omega\left(d^{\frac{\gamma}{4}c_0}\right)$.

Let $\mathcal{S}^{(0)}_{j,\text{pot}} \subseteq [m]$ be those neurons $i \in [m]$ satisfying

- $\langle w_i^{(0)}, \mathbf{M}_j \rangle^2 \geq \frac{(c_1-c_2)\log d}{d} \|\mathbf{MM}^\top w_i^{(0)}\|_2^2$

By concentration with respect to all $m$ choices of $i \in [m]$, we know with probability at least $1 - o\left(\frac{1}{d^3}\right)$ it satisfies $\left| \mathcal{S}^{(0)}_{j,\text{pot}} \right| = O\left(d^{2\gamma c_0}\right)$.

More details of the proof can be found in Lemma B.2 of (Allen-Zhu & Li, 2022). □

**Lemma B.3.** *With high probability* $1 - \frac{1}{poly(d)}$*, for every* $i \in [m]$*, the following holds:*

$$\Pr\left[ (\frac{1}{2n} \sum_{i=1}^{n} \langle w_i^{(0)}, \mathbf{M}_j \rangle - \langle w_i^{(0)}, \mathbf{M}_{j'} \rangle)^2 \geq \frac{1}{d}\frac{\sigma_0^2}{2n} \log d \right] \geq 1 - O(\frac{1}{\sqrt{d}}) \tag{44}$$

**Lemma B.4.** *With high probability* $1 - \frac{1}{poly(d)}$*, for every* $i \in [m]$*, the following holds:*

$$\|\mathbf{MM}^\top w_i^{(0)}\|_2^2 + \|\mathbf{HH}^\top v_i^{(0)}\|_2^2 \in 2d\sigma_0^2 \left[ 1 - \tilde{O}\left(\frac{1}{\sqrt{d}}\right), 1 + \tilde{O}\left(\frac{1}{\sqrt{d}}\right) \right]. \tag{45}$$

*Proof.* Let $X \sim \chi_n^2$. By standard properties of the chi-squared distribution, we know that with probability at least $1 - \delta$,

$$|X - n| \leq 2\sqrt{n\log(1/\delta)}. \tag{46}$$

In our case, we consider $\frac{\|\mathbf{MM}^\top w_i^{(0)}\|_2^2 + \|\mathbf{HH}^\top v_i^{(0)}\|_2^2}{\sigma_0^2} \sim \chi_{2d}^2$. Setting $\delta = \frac{1}{poly(d)}$, we have $n = 2d$, and thus, with high probability $1 - \frac{1}{poly(d)}$, the following holds:

$$\left| \frac{\|\mathbf{MM}^\top w_i^{(0)}\|_2^2 + \|\mathbf{HH}^\top v_i^{(0)}\|_2^2}{\sigma_0^2} - 2d \right| \leq 2\sqrt{2d\log(poly(d))}. \tag{47}$$

Rearranging and incorporating the scaling factor $\sigma_0^2$, we get:

$$\|\mathbf{MM}^\top w_i^{(0)}\|_2^2 + \|\mathbf{HH}^\top v_i^{(0)}\|_2^2 \in 2d\sigma_0^2 \left[ 1 - \tilde{O}\left(\frac{1}{\sqrt{d}}\right), 1 + \tilde{O}\left(\frac{1}{\sqrt{d}}\right) \right]. \tag{48}$$

□

**Lemma B.5** (Noise Projection Bound). *For the spurious dense noise* $\xi_{x_p} \sim \mathcal{N}(0, \sigma_\xi^2 \mathbf{I}_{d_1})$*, where the variance satisfies* $\omega\left(\frac{1}{d_1}\right) \leq \sigma_\xi^2 \leq O\left(\frac{1}{d}\right)$*, the following holds with high probability* $1 - e^{-\Omega(d_1)}$*:*

$$|\langle w_i, \xi \rangle|^2 \leq O\left(\frac{\|w_i\|_2^2}{d^{1+c_0}}\right), \quad \forall i \in [m]. \tag{49}$$

*Proof.* For all $j \in [d_1]$, by the properties of the Gaussian distribution, we have:

$$\Pr_{\xi}\left[ \langle \mathbf{M}_j, \xi \rangle^2 \leq O\left(\frac{1}{d^{1+c_0}}\right) \right] \geq 1 - e^{-\Omega(d_1)}. \tag{50}$$

Now, consider the term $|\langle w_i, \xi \rangle|^2$. We decompose it as:

$$|\langle w_i, \xi \rangle|^2 = \sum_{j \in [d]} |\langle w_i, \mathbf{M}_j \rangle|^2 \cdot |\langle \mathbf{M}_j, \xi \rangle|^2 + \sum_{j \in [d_1] \setminus [d]} |\langle w_i, \mathbf{M}_j^\perp \rangle|^2 \cdot |\langle \mathbf{M}_j^\perp, \xi \rangle|^2. \tag{51}$$

For the first term, since $|\langle \mathbf{M}_j, \xi \rangle|^2 \leq O\left(\frac{1}{d^{1+c_0}}\right)$ with high probability, we have:

$$\sum_{j \in [d]} |\langle w_i, \mathbf{M}_j \rangle|^2 \cdot |\langle \mathbf{M}_j, \xi \rangle|^2 \leq \sum_{j \in [d]} O\left(\frac{|\langle w_i, \mathbf{M}_j \rangle|^2}{d^{1+c_0}}\right). \tag{52}$$

Similarly, for the second term:

$$\sum_{j \in [d_1] \setminus [d]} |\langle w_i, \mathbf{M}_j^\perp \rangle|^2 \cdot |\langle \mathbf{M}_j^\perp, \xi \rangle|^2 \leq \sum_{j \in [d_1] \setminus [d]} O\left(\frac{|\langle w_i, \mathbf{M}_j^\perp \rangle|^2}{d^{1+c_0}}\right). \tag{53}$$

Combining these, we have:

$$|\langle w_i, \xi \rangle|^2 \leq O\left(\frac{\|\mathbf{M}\mathbf{M}^\top w_i\|_2^2}{d^{1+c_0}} + \frac{\|\mathbf{M}^\perp \mathbf{M}^{\perp \top} w_i\|_2^2}{d^{1+c_0}}\right). \tag{54}$$

Since $\|\mathbf{M}\mathbf{M}^\top w_i\|_2^2 + \|\mathbf{M}^\perp \mathbf{M}^{\perp \top} w_i\|_2^2 = \|w_i\|_2^2$, we conclude:

$$|\langle w_i, \xi \rangle|^2 \leq O\left(\frac{\|w_i\|_2^2}{d^{1+c_0}}\right). \tag{55}$$

Thus, the lemma holds. $\qquad\square$

**Lemma B.6** (Tail Bound for Matrix Product). *Let $\mathbf{Q} \in \mathbb{R}^{n \times n}$ be a symmetric matrix, and let $w, v$ be independent zero-mean Gaussian random vectors with covariance matrix $\mathbf{I}_n$. Define*

$$Z := \sum_{i,j=1}^{n} Q_{ij} w_i v_j. \tag{56}$$

*Then, for any $\delta > 0$, the following tail bound holds:*

$$\Pr[|Z| \geq \delta] \leq 4 \exp\left(-\frac{\delta^2}{4\|\mathbf{Q}\|_F^2 + 4\delta\|\mathbf{Q}\|_{op}}\right). \tag{57}$$

**Lemma B.7** (Bound Inner Product). *Consider the inner product between the feature vectors at initialization:*

$$\langle f(x), h(y) \rangle = \langle \mathbf{W}x, \mathbf{V}y \rangle = \sum_{l=1}^{m} w_l^\top x y^\top v_l = \sum_{l=1}^{m} \sum_{i,j=1}^{d_1} (x_i^\top y_j) w_l^\top v_l. \tag{58}$$

*Here, using Lemma B.6, $\mathbf{Q} = xy^\top$, with $\|\mathbf{Q}\|_{op} = \Theta(1)$, $\|\mathbf{Q}\|_F = \Theta(1)$ and $\sigma_0^2 = \Theta\left(\frac{1}{d_1 poly(d)}\right)$. Then, at initialization $(t = 0)$, the following holds:*

$$\Pr[|\langle f^{(t)}(x), h^{(t)}(y) \rangle| \geq \Omega(1)] \leq e^{-poly(d)}, \tag{59}$$

**Lemma B.8** (Concentration bound for empirical loss and gradients). *There exist $N \geq poly(d)$ for some sufficiently large polynomial and all $\|w_i\|_2 \leq O(d)$, $i \in [m]$, it satisfies*

$$\left| \frac{1}{N} \sum_{p \in [N]} L(f^{(t)}, h^{(t)}; (x_p, y_p)) - \mathbb{E}_{(x_p, y_p) \in D}[L(f^{(t)}, h^{(t)}; (x_p, y_p))] \right| \leq O(\frac{1}{d}) \tag{60}$$

$$\left\| \frac{1}{N} \sum_{p \in [N]} \nabla_{w_i} L(f^{(t)}, h^{(t)}; (x_p, y_p)) - \mathbb{E}_{(x_p, y_p) \in D}[\nabla_{w_i} L(f^{(t)}, h^{(t)}; (x_p, y_p))] \right\|_2 \leq O(\frac{1}{d}) \tag{61}$$

*Proof.* The proof can be done by trivial VC dimension or Rademacher complexity arguments similarly to Lemma A.2. (Allen-Zhu & Li, 2022). $\qquad\square$

**Lemma B.9** (Misalignment Probability Bound). *The probability of spurious alignment satisfies:*

$$\frac{\log\left(\frac{1}{2\gamma c_0}\right)}{2\log\frac{d_1}{d}} < \Pr(|z_{y_p}^j| = 1 \mid |z_{x_p}^{j'}| = 1) < \frac{1}{2}. \tag{62}$$

*Proof.* By concentration over all $m$ choices of $i \in [m]$, we find that with probability at least $1 - o\left(\frac{1}{d^3}\right)$, the number of neurons satisfying:

$$\left(\frac{1}{n}\sum_{i=1}^n \langle w_i, \mathbf{M}_j\rangle\right)^2 < (c_1 + 4c_2)\frac{\sigma_0^2}{n}\log d \tag{63}$$

is $o(1)$.

In addition, for all neurons, we have:

$$\max\left(\langle w_i^{(T_1)}, \mathbf{M}_{j'}\rangle^2\right) \le \frac{c_1 + 3c_2}{2}\frac{\log d}{d} \cdot \frac{\|w_i^{(T_1)}\|_2^2 + \|v_i^{(T_1)}\|_2^2}{2}. \tag{64}$$

Define:

$$\Delta^{(T_1)} = \frac{(a+b-c)^{T_1}}{4}\left|\langle w_i^{(0)}, \mathbf{M}_j\rangle + \langle v_i^{(0)}, \mathbf{H}_j\rangle - \langle w_i^{(0)}, \mathbf{M}_{j'}\rangle - \langle v_i^{(0)}, \mathbf{H}_{j'}\rangle\right|. \tag{65}$$

Thus:

$$\langle w_i^{(T_1)}, \mathbf{M}_{j'}\rangle^2 = \left|\max\left(\langle w_i^{(T_1)}, \mathbf{M}_{j'}\rangle\right) - \Delta^{(T_1)}\right|^2 \ge \frac{c_1 - c_2}{2}\frac{\log d}{d} \cdot \frac{\|w_i^{(T_1)}\|_2^2 + \|v_i^{(T_1)}\|_2^2}{2}. \tag{66}$$

We begin by expressing $a + b - c$ and $a + b + c$ as functions of $P_1 = \Pr(|z_{y_p}^j| = 1 \mid |z_{x_p}^{j'}| = 1)$ and $P_2 = \Pr(|z_{y_p}^j| = 1 \mid |z_{x_p}^j| = 1)$, where $P_1 + P_2 = 1$:

$$a + b - c = 1 - \eta\lambda + \eta\frac{(P_1 - P_2)C_z\log\log d}{d}, \tag{67}$$

$$a + b + c = 1 - \eta\lambda + \eta\frac{(P_1 + P_2)C_z\log\log d}{d}. \tag{68}$$

Using Eq (66), Eq (39) and Eq (40), we derive:

$$\frac{(a+b-c)^{2T_1}}{(a+b+c)^{2T_1}} \le \left(\sqrt{\frac{c_1 + 3c_2}{2}} - \sqrt{\frac{c_1 - c_2}{2}}\right)^2 \le 2c_2^2. \tag{69}$$

Substituting back, we find:

$$\frac{\log\left(\frac{1}{2\gamma c_0}\right)}{2\log\frac{d_1}{d}} < P_1 < \frac{1}{2}. \tag{70}$$

For example, setting $c_0 = 0.1$, $\gamma = 0.005$, $d = 100$, and $d_1 = 10000$, we calculate:

$$\frac{1}{4} \le \Pr(|z_{y_p}^j| = 1 \mid |z_{x_p}^{j'}| = 1) < \frac{1}{2}. \tag{71}$$

This concludes the proof by bounding $\Pr(|z_{y_p}^j| = 1 \mid |z_{x_p}^{j'}| = 1)$ under the given conditions. $\qquad\square$

## C. ITCP on Raw Data I

In this section we analyze stage I of ITCP on Raw Data as the training iterations $t \leq T_1$, where $T_1 = \Theta\left(\frac{d \log d}{\eta}\right)$ is the iteration when all $\frac{\|w_i^{(T_1)}\|_2^2 + \|v_i^{(T_1)}\|_2^2}{2} \geq \|w_i^{(0)}\|_2^2 + \|v_i^{(0)}\|_2^2$. When $t \leq T_1$, we set $b_i^{(t)} = 0$. For every neuron $i \in [m]$, the weights $w_i$ and $v_i$ exhibit an increase in alignment along the direction of informative features $\mathbf{M}$ and $\mathbf{H}$, while showing negligible increase in alignment along the direction of noise features $\mathbf{M}^\perp$ and $\mathbf{H}^\perp$.

Based on subsection A.1, we have $\Pr(|z_{y_p}^j| = 1 \mid |z_{x_p}^{j'}| = 1) = \Theta(1)$, so $\mathbb{E}\left[z_x^j z_y^j\right]$ and $\mathbb{E}\left[z_x^j z_y^{j'}\right]$ both in $\Theta\left(\frac{1}{d}\right)$. In this case, $w_i^{(t+1)}$ is jointly influenced by $\mathbf{M}_j$ and $\mathbf{M}_{j'}$, with both features contributing comparably to the updates.

To simplify our analysis, we consider the worse case where $\Pr(|z_{y_p}^{j'}| = 1 \mid |z_{x_p}^j| = 1) = \Pr(|z_{y_p}^j| = 1 \mid |z_{x_p}^j| = 1) = \frac{1}{2}$ such that $\mathbb{E}\left[z_x^j z_y^j\right] = \mathbb{E}\left[z_x^j z_y^{j'}\right] = \frac{C_z}{2d}$, so using Eq (39), Eq (40) and $b_i^{(t)} = 0$ we have

$$\langle w_i^{(t)}, \mathbf{M}_j \rangle = \frac{(a+b+c)^t}{4}\left(\langle w_i^{(0)}, \mathbf{M}_j \rangle + \langle v_i^{(0)}, \mathbf{H}_j \rangle + \langle w_i^{(0)}, \mathbf{M}_{j'} \rangle + \langle v_i^{(0)}, \mathbf{H}_{j'} \rangle\right) \tag{72}$$

$$\langle w_i^{(t)}, \mathbf{M}_{j'} \rangle = \frac{(a+b+c)^t}{4}\left(\langle w_i^{(0)}, \mathbf{M}_j \rangle + \langle v_i^{(0)}, \mathbf{H}_j \rangle + \langle w_i^{(0)}, \mathbf{M}_{j'} \rangle + \langle v_i^{(0)}, \mathbf{H}_{j'} \rangle\right) \tag{73}$$

This represents the worst-case scenario as the contributions of the aligned feature $\mathbb{E}\left[z_x^j z_y^j\right]$ and the spurious feature $\mathbb{E}\left[z_x^j z_y^{j'}\right]$ are identical. Under real circumstances, we expect $\mathbb{E}\left[z_x^j z_y^j\right] < \mathbb{E}\left[z_x^j z_y^{j'}\right]$, which would result in $\langle w_i^{(t+1)}, \mathbf{M}_j \rangle > \langle w_i^{(t+1)}, \mathbf{M}_{j'} \rangle$. However, in this worst-case scenario, the equality of contributions prevents the network from prioritizing purified features, resulting in equal magnitudes for $\langle w_i^{(t+1)}, \mathbf{M}_j \rangle$ and $\langle w_i^{(t+1)}, \mathbf{M}_{j'} \rangle$, thereby hindering effective feature separation.

We first provide a lower bound for $\|\mathbf{M}\mathbf{M}^\top w_i^{(t)}\|_2^2$ for iterations $t \leq t_1$. From Eq (124) and Eq (73) we have:

$$\|\mathbf{M}\mathbf{M}^\top w_i^{(t)}\|_2^2 = \sum_{i=1}^d \left[\frac{(a+b+c)^t}{4}\left(\langle w_i^{(0)}, \mathbf{M}_j \rangle + \langle v_i^{(0)}, \mathbf{H}_j \rangle\right) + \frac{(a+b+c)^t}{4}\left(\langle w_i^{(0)}, \mathbf{M}_{j'} \rangle + \langle v_i^0, \mathbf{H}_{j'} \rangle\right)\right]^2$$
$$= \left(1 + \frac{\eta C_z}{d}\right)^{2t} \frac{\|\mathbf{M}\mathbf{M}^\top w_i^{(0)}\|_2^2 + \|\mathbf{H}\mathbf{H}^\top v_i^{(0)}\|_2^2}{8} \tag{74}$$

$$\|\mathbf{M}^\perp(\mathbf{M}^\perp)^\top w_i^{(t)}\|_2^2 \leq \left(1 + \frac{1}{\text{poly}(d)}\right)\|\mathbf{M}^\perp(\mathbf{M}^\perp)^\top w_i^{(0)}\|_2^2. \tag{75}$$

The detailed proof of Eq (75) can be found in Hypothesis C.4 of (Wen & Li, 2021).

A similar result holds for $\|\mathbf{H}\mathbf{H}^\top v_i^{(t)}\|_2^2$ and $\|\mathbf{H}^\perp(\mathbf{H}^\perp)^\top v_i^{(t)}\|_2^2$.

Eq (74) and Eq (75) shows that the image and text dictionary features $\mathbf{M}, \mathbf{H}$ can grow exponentially, while the noisy features $\mathbf{M}^\perp, \mathbf{H}^\perp$ remain almost unchanged when $t \leq T_1$.

For $\mathbf{M}_j^\perp$ where $j \in [d_1] \setminus [d]$, using Eq (75), we obtain:

$$|\langle w_i^{(t+1)}, \mathbf{M}_j^\perp \rangle|^2 \leq O\left(\frac{1}{d_1}\right)\|w_i^{(0)}\|_2^2 \leq O\left(\frac{1}{d_1}\right) \cdot \frac{\|w_i^{(T_1)}\|_2^2 + \|v_i^{(T_1)}\|_2^2}{2}. \tag{76}$$

This result demonstrates that the noisy features $\mathbf{M}_j^\perp$ experience nearly no increase during this phase, remaining insignificant in their contribution to the alignment of $w_i$.

### C.1. Lower Bound of Alignment for $i \in S_{j,\text{sure}}$

This section provides a analysis of the alignment growth for neurons $i \in S_{j,\text{sure}}$. Specifically, we demonstrate that for every $j \in [d]$, if $i \in S_{j,\text{sure}}$, the alignment $\langle \mathbf{M}_j, w_i^{(t)} \rangle^2$ and its spurious alignment $\langle \mathbf{M}_j', w_i^{(t)} \rangle^2$ increase exponentially when $t \leq T_1$.

We now prove the lower bound of $|\langle w_i^{(T_1)}, \mathbf{M}_j \rangle|^2$ for $i \in S_{j,\text{sure}}$:

$$
\begin{aligned}
|\langle w_i^{(T_1)}, \mathbf{M}_j \rangle|^2 &= \left( 1 + \eta \frac{C_z}{d} \right)^{2T_1} \left( \frac{\langle w_i^{(0)}, \mathbf{M}_j \rangle + \langle v_i^{(0)}, \mathbf{H}_{j'} \rangle + \langle w_i^{(0)}, \mathbf{M}_{\mathbf{j}'} \rangle + \langle v_i^{(0)}, \mathbf{H}_j \rangle}{4} \right)^2 \\
&\overset{\diamondsuit}{\geq} \left( 1 + \eta \frac{C_z}{d} \right)^{2T_1} \cdot \frac{(c_1 + c_2) \log d}{d} \cdot \frac{\|\mathbf{MM}^\top w_i^{(0)}\|_2^2 + \|\mathbf{HH}^\top v_i^{(0)}\|_2^2}{8} \\
&\overset{\heartsuit}{=} \frac{(c_1 + c_2) \log d}{d} \cdot \frac{\|\mathbf{MM}^\top w_i^{(T_1)}\|_2^2 + \|\mathbf{HH}^\top v_i^{(T_1)}\|_2^2}{2} \\
&\overset{\clubsuit}{\geq} \frac{(c_1 + c_2) \log d}{d} \cdot \frac{\|w_i^{(T_1)}\|_2^2 + \|v_i^{(T_1)}\|_2^2 - \|w_i^{(0)}\|_2^2 - \|v_i^{(0)}\|_2^2}{2} \\
&\overset{\spadesuit}{>} \frac{(1 + c_0 - \gamma c_0) \log d}{d} \cdot \frac{\|w_i^{(T_1)}\|_2^2 + \|v_i^{(T_1)}\|_2^2}{2}
\end{aligned}
\tag{77}
$$

In $\diamondsuit$ we use Definition B.1. In $\heartsuit$ we use Eq (74). In $\clubsuit$ we use $\frac{\|w_i^{(T_1)}\|_2^2 + \|v_i^{(T_1)}\|_2^2}{2} \geq \|w_i^{(0)}\|_2^2 + \|v_i^{(0)}\|_2^2$. In $\spadesuit$ we use $c_1 + c_2 > 2(1 + c_0 - \gamma c_0)$.

Similarly, $|\langle w_i^{(T_1)}, \mathbf{M}_{j'} \rangle|^2$ have the same lower bound.

### C.2. Upper Bound of Alignment for $i \notin S_{j,\text{pot}}$

In this subsection, we analyze the alignment of neuron $i \notin S_{j,\text{pot}}$ with the feature $\mathbf{M}_j$ and provide an upper bound for $|\langle w_i^{(T_1)}, \mathbf{M}_j \rangle|^2$. While neurons $i \notin S_{j,\text{pot}}$ still exhibit exponential growth in their alignment, their weaker initialization results in significantly smaller alignment compared to neurons in $S_{j,\text{sure}}$, limiting their contribution to learning the feature $\mathbf{M}_j$.

To establish the bound, we begin with the following expression:

$$
\begin{aligned}
|\langle w_i^{(T_1)}, \mathbf{M}_j \rangle|^2 &= \left( 1 + \eta \frac{C_z}{d} \right)^{2T_1} \left( \frac{\langle w_i^{(0)}, \mathbf{M}_j \rangle + \langle v_i^{(0)}, \mathbf{H}_{j'} \rangle + \langle w_i^{(0)}, \mathbf{M}_{\mathbf{j}'} \rangle + \langle v_i^{(0)}, \mathbf{H}_j \rangle}{4} \right)^2 \\
&\overset{\diamondsuit}{\leq} \left( 1 + \eta \frac{C_z}{d} \right)^{2T_1} \cdot \frac{(c_1 - c_2) \log d}{d} \cdot \frac{\|\mathbf{MM}^\top w_i^{(0)}\|_2^2 + \|\mathbf{HH}^\top v_i^{(0)}\|_2^2}{8} \\
&= \frac{(c_1 - c_2) \log d}{d} \cdot \frac{\|\mathbf{MM}^\top w_i^{(T_1)}\|_2^2 + \|\mathbf{HH}^\top v_i^{(T_1)}\|_2^2}{2}.
\end{aligned}
\tag{78}
$$

Here, in $\diamondsuit$, we use Lemma B.1, which captures the reduced alignment for neurons outside $S_{j,\text{pot}}$.

Similar to the analysis for $i \in S_{j,\text{sure}}$, the alignment strength for $i \notin S_{j,\text{pot}}$ is weaker, as $c_1 - c_2$ is less than $2(1 + c_0 - \gamma c_0)$, leading to:

$$
|\langle w_i^{(T_1)}, \mathbf{M}_j \rangle|^2 < \frac{(1 + c_0 - 3\gamma c_0) \log d}{d} \cdot \frac{\|w_i^{(T_1)}\|_2^2 + \|v_i^{(T_1)}\|_2^2}{2}.
\tag{79}
$$

This inequality highlights the slower alignment for neurons outside $S_{j,\text{pot}}$, distinguishing their behavior from neurons in $S_{j,\text{sure}}$. Consequently, $i \notin S_{j,\text{pot}}$ contributes less significantly to the alignment of $\mathbf{M}_j$, reinforcing the importance of initial affinity for effective alignment.

### C.3. Summary

At this stage ($t \leq T_1$), we do not consider the worst-case scenario where the probability bounds for feature coupling satisfy

$$
\frac{\log(1/c_0)}{2 \log d} < \Pr(|z_{y_p}^{j'}| = 1 \mid |z_{x_p}^j| = 1) < \frac{1}{2} < \Pr(|z_{y_p}^j| = 1 \mid |z_{x_p}^j| = 1) < 1
$$

(as assumed in Subsection A.1). Thus, we summarize the results when $t \leq T_1$ as follows:

1. For $i \in S_{j,\text{sure}}$, the alignment strength satisfies:

$$|\langle w_i^{(T_1)}, \mathbf{M}_j \rangle|^2 > |\langle w_i^{(T_1)}, \mathbf{M}_{j'} \rangle|^2 > \frac{(1 + c_0 - \gamma c_0) \log d}{d} \cdot \frac{\|w_i^{(T_1)}\|_2^2 + \|v_i^{(T_1)}\|_2^2}{2}, \tag{80}$$

where $j'$ represents the corresponding spurious alignment feature.

2. For $i \notin S_{j,\text{pot}}$, the alignment strength satisfies:

$$|\langle w_i^{(T_1)}, \mathbf{M}_j \rangle|^2 < \frac{(1 + c_0 - 3\gamma c_0) \log d}{d} \cdot \frac{\|w_i^{(T_1)}\|_2^2 + \|v_i^{(T_1)}\|_2^2}{2}. \tag{81}$$

3. For $\mathbf{M}_j^\perp$ where $j \in [d_1] \setminus [d]$, we have:

$$|\langle w_i^{(t+1)}, \mathbf{M}_j^\perp \rangle|^2 < O\left(\frac{1}{d_1}\right) \cdot \frac{\|w_i^{(T_1)}\|_2^2 + \|v_i^{(T_1)}\|_2^2}{2}. \tag{82}$$

These results demonstrate that when $t \leq T_1$, all features in $\mathbf{M}$ increase, but the alignment for $i \in S_{j,\text{sure}}$, including the corresponding spurious alignment, grows significantly larger due to favorable initialization. In contrast, noisy features $\mathbf{M}^\perp$ remain unchanged.

## D. ITCP on Raw Data II

The stage II of ITCP on Raw Data is defined as the training iterations $T_1 < t \leq T_2$, where $T_2 - T_1 = \Theta\left(\frac{d \log d}{\eta}\right)$.

At the beginning of this phase, we set the bias threshold as:

$$b_i^{(T_1)} = \sqrt{\frac{(1 + c_0 - 2\gamma c_0) \log d}{d} \cdot \frac{\|w_i^{(T_1)}\|_2^2 + \|v_i^{(T_1)}\|_2^2}{2}}. \tag{83}$$

During training, the bias threshold is iteratively updated as:

$$b_i^{(t+1)} = \left(1 + \frac{\eta}{d}\right) b_i^{(t)}, \tag{84}$$

until all neurons satisfy:

$$\|w_i^{(T_2)}\|_2^2 \geq \Omega(d) \|w_i^{(T_1)}\|_2^2. \tag{85}$$

In this phase, the dynamics of alignment vary depending on whether a neuron belongs to $S_{j,\text{sure}}$ or not:

- For $i \notin S_{j,\text{pot}}$: The weights $w_i$ and $v_i$ show negligible alignment growth with both the informative features $\mathbf{M}_j, \mathbf{H}_j$ and the noise features $\mathbf{M}^\perp, \mathbf{H}^\perp$. This is due to their weaker initialization, as shown in Stage I, and the effect of the indicator function when $t \geq T_1$ which prevents them from being activated. As a result, their capacity to learn meaningful alignments during this phase is significantly limited.
- For $i \in S_{j,\text{sure}}$: The weights $w_i$ and $v_i$ exhibit continued alignment growth with the informative features $\mathbf{M}_j, \mathbf{H}_j$. Additionally, their alignment with the corresponding spurious features $\mathbf{M}_{j'}, \mathbf{H}_{j'}$ also increases due to their strong initialization, as shown in Stage I, and the effect of the indicator function when $t \geq T_1$, which ensures they are always activated.

By the end of this stage ($t = T_2$), the weights $w_i, v_i$ will predominantly focus on the features $\mathbf{M}_j, \mathbf{H}_j$ if $i \in S_{j,\text{sure}}$, while largely ignoring the features $\mathbf{M}_j, \mathbf{H}_j$ if $i \notin S_{j,\text{pot}}$, as well as the noise features $\mathbf{M}^\perp, \mathbf{H}^\perp$. This separation lays the foundation for the stage II of ITCP on Raw Data, where spurious alignments are expected to further diminish due to the dominance of true feature alignments.

Similarly to the proof of $t \leq T_1$ To simplify our analysis, we still consider the worse case where $\Pr(|z_{y_p}^{j'}| = 1 \mid |z_{x_p}^j| = 1) = \Pr(|z_{y_p}^j| = 1 \mid |z_{x_p}^j| = 1) = \frac{1}{2}$ such that $\mathbb{E}\left[z_x^j z_y^j\right] = \mathbb{E}\left[z_x^j z_y^{j'}\right] = \frac{C_z}{2d}$.

**D.1. Alignment for $i \in S_{j,\text{sure}}$**

This section provides a analysis of the alignment growth for neurons $i \in S_{j,\text{sure}}$. Specifically, we demonstrate that for every $j \in [d]$, if $i \in S_{j,\text{sure}}$, the alignment $\langle \mathbf{M}_j, w_i^{(t)} \rangle^2$ and its spurious alignment $\langle \mathbf{M}'_j, w_i^{(t)} \rangle^2$ increase exponentially when $T_1 < t \leq T_2$.

For $i \in S_{j,\text{sure}}$, using Lemma B.5, the following holds with high probability $1 - e^{-\Omega(d_1)}$ when $T_1 < t \leq T_2$:

$$\left| \langle w_i^{(t)}, \xi \rangle \right|^2 \leq O \left( \frac{\left\| w_i^{(t)} \right\|_2^2}{d^{1+c_0}} \right) < b_i^{(t)} \tag{86}$$

Therefore, with high probability $1 - e^{-\Omega(d_1)}$, using Eq (80) and Eq (83) the indicator function satisfies the condition when $t = T_1$:

$$\mathbf{1}_{\left| \langle w_i^{(t)}, x_p \rangle \right| \geq b_i^{(t)}} \cdot \mathbf{1}_{\left| \langle v_i^{(t)}, y_p \rangle \right| \geq b_i^{(t)}} = 1, \tag{87}$$

we can ensure that:

$$\mathbb{E} \left[ z_x^j z_y^j \cdot \mathbf{1}_{\left| \langle w_i^{(t)}, x_p \rangle \right| \geq b_i^{(t)}} \cdot \mathbf{1}_{\left| \langle v_i^{(t)}, y_p \rangle \right| \geq b_i^{(t)}} \right] = \frac{C_z}{d}. \tag{88}$$

Using Eq (118) we know that $\left( 1 + \eta \frac{C_z}{2d} \right) > \left( 1 + \frac{\eta}{d} \right)$ and using Eq (38) we have

$$|\langle w_i^{(t+1)}, \mathbf{M}_j \rangle| > (1 + \frac{\eta}{d}) b_i^{(t)} = b_i^{(t+1)}. \tag{89}$$

This implies that when $t > T_1$, the alignment strength of informative features surpasses the updated bias threshold $b_i^{(t)}$. Consequently, the indicator functions become consistently activated $T_1 < t \leq T_2$ such that

$$\mathbf{1}_{\left| \langle w_i^{(t)}, x_p \rangle \right| \geq b_i^{(t)}} \cdot \mathbf{1}_{\left| \langle v_i^{(t)}, y_p \rangle \right| \geq b_i^{(t)}} = 1, \tag{90}$$

Using Eq (38), the weight dynamics for $|\langle w_i^{(t+1)}, \mathbf{M}_j \rangle|$ can be expressed as when $T_1 < t \leq T_2$:

$$|\langle w_i^{(t+1)}, \mathbf{M}_j \rangle| = \left( 1 + \eta \frac{C_z}{d} \right) \left( \frac{\langle w_i^{(t)}, \mathbf{M}_j \rangle + \langle v_i^{(t)}, \mathbf{H}_{j'} \rangle + \langle w_i^{(t)}, \mathbf{M}_{\mathbf{j}}^{\perp} \rangle + \langle v_i^{(t)}, \mathbf{H}_j \rangle}{4} \right). \tag{91}$$

Similarly, $|\langle w_i^{(T_1)}, \mathbf{M}_{j'} \rangle|^2$ have the same result.

**D.2. Alignment for $i \notin S_{j,\text{pot}}$**

In this section, we analyze the alignment behavior for neurons $i \notin S_{j,\text{pot}}$. Specifically, we demonstrate that for every $j \in [d]$, if $i \notin S_{j,\text{pot}}$, the alignment $\langle \mathbf{M}_j, w_i^{(t)} \rangle^2$ exhibits negligible growth during the interval $T_1 < t \leq T_2$.

For $i \notin S_{j,\text{pot}}$, using Eq (158), Eq (83) and Eq (80), we have with high probability $1 - e^{-\Omega(d_1)}$, similarly to the proof of $i \in S_{j,\text{sure}}$, the indicator function satisfies the condition when $t = T_1$:

$$\mathbf{1}_{\left| \langle w_i^{(t)}, x_p \rangle \right| \geq b_i^{(t)}} \cdot \mathbf{1}_{\left| \langle v_i^{(t)}, y_p \rangle \right| \geq b_i^{(t)}} = 0, \tag{92}$$

We can ensure that:

$$\mathbb{E} \left[ z_x^j z_y^j \cdot \mathbf{1}_{\left| \langle w_i^{(t)}, x_p \rangle \right| \geq b_i^{(t)}} \cdot \mathbf{1}_{\left| \langle v_i^{(t)}, y_p \rangle \right| \geq b_i^{(t)}} \right] \leq o \left( \frac{1}{d^2} \right). \tag{93}$$

Using Eq (118) we know that $\left( 1 + o(\frac{\eta}{d^2}) \right) < \left( 1 + \frac{\eta}{d} \right)$ and using Eq (38) we have

$$|\langle w_i^{(t+1)}, \mathbf{M}_j \rangle| < (1 + \frac{\eta}{d}) b_i^{(t)} = b_i^{(t+1)}. \tag{94}$$

This implies that when $t > T_1$, the alignment strength of informative features does not surpass the updated bias threshold $b_i^{(t)}$. Consequently, the indicator functions become consistently not activated $T_1 < t \leq T_2$ such that

$$\mathbf{1}_{\left|\langle w_i^{(t)}, x_p \rangle\right| \geq b_i^{(t)}} \cdot \mathbf{1}_{\left|\langle v_i^{(t)}, y_p \rangle\right| \geq b_i^{(t)}} = 0, \tag{95}$$

Using Eq (38), the weight dynamics for $|\langle w_i^{(t+1)}, \mathbf{M}_j \rangle|$ can be expressed as when $T_1 < t \leq T_2$:

$$|\langle w_i^{(t+1)}, \mathbf{M}_j \rangle| \leq \left(1 + o\left(\frac{\eta}{d^2}\right)\right)^t \left(\frac{\langle w_i^{(T_1)}, \mathbf{M}_j \rangle + \langle v_i^{(T_1)}, \mathbf{H}_{j'} \rangle + \langle w_i^{(T_1)}, \mathbf{M}_{\mathbf{j'}} \rangle + \langle v_i^{(T_1)}, \mathbf{H}_j \rangle}{4}\right) \tag{96}$$

Because $\left(1 + o\left(\frac{\eta}{d^2}\right)\right)^{T_2} \leq 1 + o\left(\frac{1}{d}\right)$, the growth in $|\langle w_i^{(T_2)}, \mathbf{M}_j \rangle|$ is negligible. Consequently, we have:

$$|\langle w_i^{(T_2)}, \mathbf{M}_j \rangle|^2 \leq \left(1 + o\left(\frac{1}{d}\right)\right) |\langle w_i^{(T_1)}, \mathbf{M}_j \rangle|^2. \tag{97}$$

### D.3. Summary

When $T_2 = \Theta\left(\frac{d \log d}{\eta}\right)$, we know $\left(1 + \eta \frac{C_z}{d}\right)^{T_2} = poly(d)$. Using Eq (80), we can ensure that when all neurons satisfy the following condition:

$$\|w_i^{(T_2)}\|_2 \geq \Omega(d)\|w_i^{(T_1)}\|_2, \tag{98}$$

we terminate the training process at $T_2 = \Theta\left(\frac{d \log d}{\eta}\right)$. This ensures that the alignment has sufficiently progressed for effective learning.

Thus, using Eq (97) and Eq (75) we have

$$|\langle w_i^{(T_2)}, \mathbf{M}_j \rangle|^2 + |\langle w_i^{(T_2)}, \mathbf{M}_{j'} \rangle|^2 = \|w_i^{(T_2)}\|_2^2 - \sum_{j \in [d], j \notin \mathcal{N}_i} \langle w_i^{(T_2)}, \mathbf{M}_j \rangle^2 - \sum_{j \in [d_1] \setminus [d]} \langle w_i^{(T_2)}, \mathbf{M}_j^\perp \rangle^2$$

$$\geq \|w_i^{(T_2)}\|_2^2 - (1 + o(\frac{1}{d}))(\|w_i^{(T_1)}\|_2^2 - |\langle w_i^{(T_1)}, \mathbf{M}_j \rangle|^2 - |\langle w_i^{(T_1)}, \mathbf{M}_{j'} \rangle|^2) \tag{99}$$

$$\geq \|w_i^{(T_2)}\|_2^2 - \|w_i^{(T_1)}\|_2^2 - o(\frac{\|w_i^{(T_1)}\|_2^2}{d})$$

Thus, at this stage ($T_1 < t \leq T_2$), we do not consider the worst-case scenario where the probability bounds for feature coupling satisfy

$$\frac{\log(1/c_0)}{2 \log d} < \Pr(|z_{y_p}^{j'}| = 1 \mid |z_{x_p}^j| = 1) < \frac{1}{2} < \Pr(|z_{y_p}^j| = 1 \mid |z_{x_p}^j| = 1) < 1$$

We summarize the results when $T_1 < t \leq T_2$ as follows:

1. For $i \in S_{j,\text{sure}}$, the alignment strength satisfies:

$$|\langle w_i^{(T_2)}, \mathbf{M}_j \rangle|^2 > |\langle w_i^{(T_2)}, \mathbf{M}_{j'} \rangle|^2 \geq \frac{1}{4} \frac{\|w_i^{(T_2)}\|_2^2 + \|v_i^{(T_2)}\|_2^2}{2} \tag{100}$$

where $j'$ represents the corresponding spurious alignment feature.

2. For $i \notin S_{j,\text{pot}}$, the alignment strength satisfies:

$$|\langle w_i^{(T_1)}, \mathbf{M}_j \rangle|^2 \leq O(\frac{1}{d}) \cdot \frac{\|w_i^{(T_2)}\|_2^2 + \|v_i^{(T_2)}\|_2^2}{2} \tag{101}$$

3. For $\mathbf{M}_j^\perp$ where $j \in [d_1] \setminus [d]$, we have:

$$|\langle w_i^{(t+1)}, \mathbf{M}_j^\perp \rangle|^2 < O\left(\frac{1}{d_1}\right) \cdot \frac{\|w_i^{(T_2)}\|_2^2 + \|v_i^{(T_2)}\|_2^2}{2}. \tag{102}$$

These results demonstrate that when $T_1 < t \leq T_2$, the alignment for $i \in S_{j,\text{sure}}$, including the corresponding spurious alignment, grows significantly larger. In contrast, the alignment strength for $i \notin S_{j,\text{pot}}$ and noisy features $\mathbf{M}^\perp$ remains unchanged. Similar results also hold for $v_i$.

# E. ITCP on Raw Data III Convergence

In the previous section, we demonstrated that for $t \leq T_2$, the neurons $(w_i, v_i)$ are sparsely activated and remain consistently activated for $i \in S_{j,\text{sure}}$. Building on this result, this section establishes the convergence of these neurons to sparse solutions, providing a detailed analysis of their behavior during Stage III of ITCP on Raw Data. The following theorem outlines the convergence guarantees under these conditions.

The Stage III of ITCP on Raw Data is defined as the training iterations $T_2 < t \leq T_3$, where $T_3 - T_2 = \Theta(d)$. At the beginning of this phase, we fix the bias threshold as $b_i^{(t)} = b_i^{T_2}$ for $T_2 < t \leq T_3$. Because $b_i^{(T_2)} = \left(1 + \frac{\eta}{d}\right)^{\Theta(d \log d/\eta)} b_i^{(T_1)}$, it is easy to know that for $t \geq T_2$, only when $(x_p, y_p)$ and $(x_n, y_n)$ contain the true feature $j$ and its corresponding spurious feature $j'$, the indicator functions remain consistently activated for $i \in S_{j,\text{sure}}$.

Consequently, using Eq (31), Eq (34), and Eq (35), the loss function $L$ becomes convex with respect to $w_i$ and $v_i$ independently when $(x_p, y_p)$ and $(x_n, y_n)$ contain the true feature $j$ and its corresponding spurious feature $j'$.

At the end of Stage II, using Eq (85), we know that $\|w_i^{(T_2)}\|_2 \geq \Omega(d)$. Consequently, we cannot only consider $-\langle f^{(t)}(x_p), h^{(t)}(y_p)\rangle$, and the error term $Err_t$ becomes non-negligible.

Specifically, based on Eq (31), it can be observed that the term $-\langle f^{(t)}(x_p), h^{(t)}(y_p)\rangle$ is convex and $l_{i,j,1} = \|x_p\|_2 \|y_p\|_2 = \Theta(1)$-smooth. This ensures that the true features contribute consistently to the optimization process.

Additionally, $L_{i,j,2} = \frac{\left(\langle f^{(t)}(x_n), h^{(t)}(y_p)\rangle\right)^2}{2\tau}$ is also convex, and we further establish its smoothness to provide a rigorous understanding of its behavior.

To analyze the $l_{i,j,2}$-smoothness, we aim to find an upper bound that satisfies:

$$\|\nabla_{w_i, v_i} L_2(w_{i,1}, v_{i,1}) - \nabla_{w_i, v_i} L_2(w_{i,2}, v_{i,2})\|_2 \leq l_{i,j,2} \|(w_{i,1} - w_{i,2}, v_{i,1} - v_{i,2})\|_2. \tag{103}$$

The gradient difference for $w_i$ is given by:

$$\begin{aligned}
\|\nabla_{w_i} L_{i,j,2}(w_{i,1}, v_{i,1}) - \nabla_{w_i} L_{i,j,2}(w_{i,2}, v_{i,2})\|_2 &= \left\| \left(x^\top W_1^\top V_1 y\right) x(v_{i,1} y)^\top - \left(x^\top W_2^\top V_2 y\right) x(v_{i,2} y)^\top \right\|_2 / (2\tau) \\
&\leq \frac{l_{w_i,1}}{2\tau} \|w_{i,1} - w_{i,2}\|_2 + \frac{l_{w_i,2}}{2\tau} \|v_{i,1} - v_{i,2}\|_2,
\end{aligned} \tag{104}$$

where $l_{w_i,1} = \|x_n\|_2^2 \|y_p\|_2^2 \|v_{i,1}\|_2 \|v_{i,2}\|_2 \leq O(d)$ and $l_{w_i,2} = \|x_n\|_2^2 \|y_p\|_2^2 \left(\|v_{i,1}\|_2 \|w_{i,2}\|_2 + \|w_{i,1}\|_2 \|v_{i,1}\|_2\right) \leq O(d)$.

Similarly, the gradient difference for $v_i$ is:

$$\|\nabla_{v_i} L_{i,j,2}(w_{i,1}, v_{i,1}) - \nabla_{v_i} L_{i,j,2}(w_{i,2}, v_{i,2})\|_2 \leq \frac{l_{v_i,1}}{2\tau} \|w_{i,1} - w_{i,2}\|_2 + \frac{l_{v_i,2}}{2\tau} \|v_{i,1} - v_{i,2}\|_2, \tag{105}$$

where $l_{v_i,1} \leq O(d)$ and $l_{v_i,2} \leq O(d)$.

Combining the results, we find:

$$l_{i,j,2} = \frac{\sqrt{l_{w_i,1}^2 + l_{w_i,2}^2 + l_{v_i,1}^2 + l_{v_i,2}^2}}{2\tau} \leq O(1). \tag{106}$$

Thus, the total smoothness constant is:

$$l_{i,j} = l_{i,j,1} + l_{i,j,2} = \Theta(1). \tag{107}$$

These results demonstrate that the loss function $L$ remains convex and $l_{i,j}$-smooth for neurons $(w_i, v_i)$ when $(x_p, y_p)$ and $(x_n, y_n)$ contain the true feature $j$ and its corresponding spurious feature $j'$ during Stage III of ITCP on Raw Data, ensuring their convergence to sparse solutions while maintaining consistency in their activation patterns.

We verify that the following inequality holds

$$L_j(w_i^{(t+1)}, v_i^{(t+1)}) \leq L_j(w_i^{(t)}, v_i^{(t)}) + \left\langle \nabla L_j(w_i^{(t)}, v_i^{(t)}), \left(w_i^{(t+1)} - w_i^{(t)}, v_i^{(t+1)} - v_i^{(t)}\right)\right\rangle + \frac{l_{i,j}}{2} \left\|\left(w_i^{(t+1)} - w_i^{(t)}, v_i^{(t+1)} - v_i^{(t)}\right)\right\|^2 \tag{108}$$

Let $L = \max_{i \in m}(l_{i,j}/(2\tau)) = \Theta(1)$ and $\eta = \frac{1}{L}$ to ensure a monotonic decrease, plug Eq (32) and Eq (33) into Eq (180), we have

$$L_j(w_i^{(t+1)}, v_i^{(t+1)}) \leq L_j(w_i^{(t)}, v_i^{(t)}) - \frac{\eta}{2}\|\nabla L_j(w_i^{(t)}, v_i^{(t)})\|^2. \tag{109}$$

Under our data assumptions for $S_w$ and conclusion in Eq (100), we define $w_i^* = \alpha_{i,j}^* \mathbf{M}_j + \alpha_{i,j'}^* \mathbf{M}_{j'}, v_i^* = \alpha_{i,j}^* \mathbf{H}_j + \alpha_{i,j'}^* \mathbf{H}_{j'}$. Thus, $L_j(w_i^*, v_i^*)$ captures both the alignment with the true feature $\mathbf{M}_j, \mathbf{H}_j$ and the spurious feature $\mathbf{M}_{j'}, \mathbf{H}_{j'}$, representing the minimal loss achievable under the influence of both true and spurious features in the optimization process. Using Eq (85), we know $w_i^{(T_2)} = \Theta(d)$, so $L_j(w_i^*, v_i^*) = -\Theta(d)$.

By the property of smoothness, we have

$$\|\nabla L_j(w_i^{(t)}, v_i^{(t)})\|_2^2 \geq \frac{2}{L}\left(L_j(w_i^{(t)}, v_i^{(t)}) - L_j(w_i^*, v_i^*)\right) \tag{110}$$

Take the telescope sum of from $T_2$ to $T_3$, we have

$$\frac{1}{T_3 - T_2}\sum_{t=T_2}^{T_3} L_j(w_i^{(t)}, v_i^{(t)}) \leq L_j(w_i^*, v_i^*) + \frac{L^2 \Delta_0}{T_3 - T_2}$$

$$\overset{\diamondsuit}{\leq} L_j(w_i^*, v_i^*) + \Theta(1) \tag{111}$$

where $\Delta_0 = L_j(w_i^{(T_1)}, v_i^{(T_1)}) - L_j(w_i^*, v_i^*) = \Theta(d)$. In $\diamondsuit$, we use $T_3 - T_2 = \Theta(d)$, and $L = \Theta(1)$.

Generalized to every $j \in d$, the same convergence holds for all $i \in S_{j,\text{sure}}$ when $(x_p, y_p)$ and $(x_n, y_n)$ contain feature $j, j'$. For all $(x_p, y_p)$ and $(x_n, y_n)$ in $S_w$, the following inequality holds:

$$\frac{1}{T_2}\sum_{t=0}^{T_2} L(f^{(T_2)}, h^{(T_2)}) \leq L(f^*, h^*) + \Theta(1). \tag{112}$$

As a result, the relative difference is bounded by:

$$\frac{L(f^{(T_2)}, h^{(T_2)}) - L(f^*, h^*)}{|L(f^*, h^*)|} \leq \Theta\left(\frac{1}{d}\right). \tag{113}$$

## F. Captioning

In this stage, the model fine-tunes the pre-trained encoder parameters $\mathbf{W}$ and $\mathbf{V}$ to obtain the updated parameters $\hat{\mathbf{W}}$ and $\hat{\mathbf{V}}$ through Image-Text Contrastive Pre-training (ITCP) on raw data.

Given an image-text pair $(x_p, y_p)$ in $S_w$, the decoder generates synthetic captions $\hat{y}_p = \hat{\mathbf{V}}^T \sigma(\hat{\mathbf{W}} x_p)$, where $\sigma(\cdot)$ denotes the activation function. The Image-Grounded Text Decoder, initialized with $\mathbf{W}$ and $\mathbf{V}$ from the pre-trained encoders, is fine-tuned on $S_h$ by minimizing the following loss function:

$$L_C = \mathbb{E}_{(x_p, y_p) \in S_h}\left[\frac{1}{2}\|\mathbf{V}^T \sigma(\mathbf{W} x_p) - y_p\|_2^2\right], \tag{114}$$

where $\|\cdot\|_2$ denotes the Euclidean norm. This fine-tuning process refines the model to generate captions that are more closely aligned with the target text data in $S_h$.

During the captioning, we sample a batch of image-text pairs $S_h^{(t)} = \{(x_p, y_p)\}_{p=1}^B \subseteq S_h$. We perform stochastic gradient descent on $L_C$. At each iteration, we update as

$$w_i^{(t+1)} \leftarrow w_i^{(t)} - \eta \nabla_{w_i} L_C^{(t)} \tag{115}$$

$$v_i^{(t+1)} \leftarrow v_i^{(t)} - \eta \nabla_{v_i} L_C^{(t)} \tag{116}$$

At the beginning of this phase, we set the bias threshold as:

$$b_i^{(0)} = \sqrt{\frac{\|w_i^{(T_2)}\|_2^2 - \|w_i^{(T_1)}\|_2^2}{2}} \tag{117}$$

During training, the bias threshold is iteratively updated as:

$$b_i^{(t+1)} = \left(1 + \frac{\eta}{d}\right) b_i^{(t)}, \tag{118}$$

The gradient of $L_C$ with respect to $w_i^{(t)}$, $v_i^{(t)}$, $\mathbf{W}$, and $\mathbf{V}$ at iteration $t$ is given by:

$$\nabla_{w_i^{(t)}} L_C = v_i^{(t)}(y_p - \mathbf{V}^T\mathbf{W}x_p)x_p^T \cdot \mathbf{1}_{\left|\left\langle w_i^{(t)}, x_p \right\rangle\right| \geq b_i^{(t)}} \tag{119}$$

$$\nabla_{v_i^{(t)}} L_C = w_i^{(t)} x_p(y_p - \mathbf{V}^T\mathbf{W}x_p)^T \cdot \mathbf{1}_{\left|\left\langle w_i^{(t)}, x_p \right\rangle\right| \geq b_i^{(t)}} \tag{120}$$

The alignment can be described by the following update rule:

$$
\begin{aligned}
\langle w_i^{(t+1)}, \mathbf{M}_j \rangle &= \langle w_i^{(t)}, \mathbf{M}_j \rangle - \langle \nabla_{w_i} L_C, \mathbf{M}_j \rangle \\
&= \langle w_i^{(t)}, \mathbf{M}_j \rangle + \eta \cdot \operatorname{tr}(v_i^{(t)}(y_p - \mathbf{V}^T\mathbf{W}x_p)x_p^T\mathbf{M}_j \cdot \mathbf{1}_{|\langle w_i, x_p \rangle| \geq b_i^{(t)}})
\end{aligned} \tag{121}
$$

$$
\begin{aligned}
\langle v_i^{(t+1)}, \mathbf{H}_j \rangle &= \langle v_i^{(t)}, \mathbf{H}_j \rangle - \langle \nabla_{v_i} L_C, \mathbf{H}_j \rangle \\
&= \langle v_i^{(t)}, \mathbf{H}_j \rangle + \eta \cdot \operatorname{tr}(w_i^{(t)} x_p(y_p - \mathbf{V}^T\mathbf{W}x_p)^T\mathbf{H}_j \cdot \mathbf{1}_{|\langle w_i, x_p \rangle| \geq b_i^{(t)}})
\end{aligned} \tag{122}
$$

### F.1. Alignment for $i \in S_{j,\text{sure}}$

This section analyzes the alignment growth for neurons $i \in S_{j,\text{sure}}$. Specifically, we show that when $t \leq T_C$, the alignment with the true feature $\mathbf{M}_j$ grows exponentially if $x_p$ contains the true feature $\mathbf{M}_j$. In contrast, the alignment with the spurious feature $\mathbf{M}_{j'}$ exhibits negligible growth, even for neurons $i \in S_{j,\text{sure}}$. Specially,

1. For the true feature $\mathbf{M}_j$, based on the result in Eq (100) and the bias threshold in Eq (117), the indicator functions are always activated. This ensures that the neuron can consistently increase its alignment in the direction of the true feature $\mathbf{M}_j$.

2. For the spurious feature $\mathbf{M}_{j'}$, based on the result in Eq (100) and the bias threshold in Eq (117), the indicator functions remain non-activated. This prevents the neuron from increasing its alignment in the direction of the spurious feature $\mathbf{M}_{j'}$.

The details of proof as follow:

Using Eq (99), we know

$$\|w_i^{(T_2)}\|_2^2 - \|w_i^{(T_1)}\|_2^2 \geq |\langle w_i^{(T_2)}, \mathbf{M}_j \rangle|^2 + |\langle w_i^{(T_2)}, \mathbf{M}_{j'} \rangle|^2 \geq \|w_i^{(T_2)}\|_2^2 - \|w_i^{(T_1)}\|_2^2 - o(\frac{\|w_i^{(T_1)}\|_2^2}{d}) \tag{123}$$

Using Eq (39) and Eq (40), we have

$$\langle w_i^{(t)}, \mathbf{M}_j \rangle - \langle w_i^{(t)}, \mathbf{M}_{j'} \rangle = \frac{(a+b-c)^t}{2}\left(\langle w_i^{(0)}, \mathbf{M}_j \rangle + \langle v_i^{(0)}, \mathbf{H}_j \rangle - \langle w_i^{(0)}, \mathbf{M}_{j'} \rangle - \langle v_i^{(0)}, \mathbf{H}_{j'} \rangle\right) + Err_t \tag{124}$$

Using Eq (44) and $(a+b-c)^{T_1+T_2} \geq \Omega(d^2)$, with high probability $1 - O(\frac{1}{\sqrt{d}})$ we have,

$$|\langle w_i^{(T_2)}, \mathbf{M}_j \rangle|^2 - |\langle w_i^{(T_2)}, \mathbf{M}_{j'} \rangle|^2 \geq \Omega(\frac{\|w_i^{(T_1)}\|_2^2}{d}) \tag{125}$$

Therefore, with high probability $1 - O(\frac{1}{\sqrt{d}})$ we have

$$|\langle w_i^{(T_2)}, \mathbf{M}_j \rangle|^2 > \frac{\|w_i^{(T_2)}\|_2^2 - \|w_i^{(T_1)}\|_2^2}{2} > |\langle w_i^{(T_2)}, \mathbf{M}_{j'} \rangle|^2 \tag{126}$$

We set $b_i^{(0)} = \sqrt{\frac{\|w_i^{(T_2)}\|_2^2 - \|w_i^{(T_1)}\|_2^2}{2}}$, and using Eq (126), so similarly to the proof of Eq (90) we can prove:

1. For $i \in S_{j,\text{sure}}$ and $x_p$ contain the true feature $\mathbf{M}_j$, with high probability $1 - O(\frac{1}{\sqrt{d}})$ the indicator functions become consistently activated $0 \le t \le T_C$ such that:

$$\mathbf{1}_{\left|\langle w_i^{(t)}, x_p \rangle\right| \ge b_i^{(t)}} = 1 \tag{127}$$

2. For $i \in S_{j,\text{sure}}$ and $x_p$ contain the corresponding spurious aligned feature $\mathbf{M}_{j'}$, with high probability $1 - O(\frac{1}{\sqrt{d}})$ the indicator functions become consistently activated $0 \le t \le T_C$ such that:

$$\mathbf{1}_{\left|\langle w_i^{(t)}, x_p \rangle\right| \ge b_i^{(t)}} = 0 \tag{128}$$

3. For $i \notin S_{j,\text{pot}}$ and $\mathbf{M}_j^{\perp}$ where $j \in [d_1] \setminus [d]$, we have:

$$\mathbf{1}_{\left|\langle w_i^{(t)}, x_p \rangle\right| \ge b_i^{(t)}} = 0 \tag{129}$$

For the residual loss in Eq (121) and Eq (122), we bound the difference if $\mathbf{1}_{\left|\langle w_i^{(t)}, x_p \rangle\right| \ge b_i^{(t)}} = 1$:

$$
\begin{aligned}
\mathbf{H}_j z_{x_p}^j z_{y_p}^j &\overset{\diamond}{\ge} (y_p - \mathbf{V}^T \mathbf{W} x_p) x_p^T \mathbf{M}_j \cdot \mathbf{1}_{|\langle w_i, x_p \rangle| \ge b_i^{(t)}} \\
&= \left( \mathbf{H}_j z_{x_p}^j z_{y_p}^j - \sum_{i=1}^{m} \langle v_i, \mathbf{H}_j \rangle \langle w_i, \mathbf{M}_j \rangle \mathbf{H}_j z_{x_p}^j z_{y_p}^j \right) \cdot \mathbf{1}_{\langle w_i, x_p \rangle \ge b} \\
&\overset{\heartsuit}{\ge} \mathbf{H}_j z_{x_p}^j z_{y_p}^j - O(d^{\gamma c_0}) \langle v_i, \mathbf{H}_j \rangle \langle w_i, \mathbf{M}_j \rangle \mathbf{H}_j z_{x_p}^j z_{y_p}^j
\end{aligned}
\tag{130}
$$

In $\diamond$, we employ the approximation $y_p x_p^\top \mathbf{M}_j \approx \mathbf{H}_j z_{x_p}^j z_{y_p}^j$, based on the observation that $z_{x_p}^j z_{y_p}^{j'} \ll z_{x_p}^j z_{y_p}^j$ when $j \ne j'$. In $\heartsuit$, we utilize Eq (42). There are at most $O(d^{\gamma c_0})$ neurons capable of learning $\mathbf{M}_j$, which satisfy the condition $\mathbf{1}_{\langle w_i, x_p \rangle \ge b}$.

For $i \in S_{j,\text{sure}}$ and for $x_p$ contain $\mathbf{M}_j$, using Eq (130), Eq (121) and Eq (128) we have:

$$
\begin{aligned}
\langle w_i^{(t+1)}, \mathbf{M}_j \rangle &\ge \langle w_i^{(t)}, \mathbf{M}_j \rangle + \eta \cdot \text{tr}\left( v_i^{(t)} \cdot (1 - \alpha_t^2) \mathbf{H}_j \mathbb{E}\left[ z_{x_p}^j z_{y_p}^j \right] \right) \\
&\ge \langle w_i^{(t)}, \mathbf{M}_j \rangle + \eta \frac{C_z (1 - \alpha_t^2)}{d} \langle v_i^{(t)}, \mathbf{H}_j \rangle,
\end{aligned}
\tag{131}
$$

Similar to Eq (39), we have

$$|\langle w_i^{(t)}, \mathbf{M}_j \rangle| \ge \left( 1 + \eta \frac{C_z \cdot (1 - \alpha_t^2)}{d} \right)^t \left( \frac{\langle w_i^{(0)}, \mathbf{M}_j \rangle + \langle v_i^{(0)}, \mathbf{H}_j \rangle}{2} \right) \tag{132}$$

Similarly, for $i \in S_{j,\text{sure}}$ and $x_p$ contain the corresponding spurious aligned feature $\mathbf{M}_{j'}$, because $\Pr[\mathbf{1}_{\left|\langle w_i^{(t)}, x_p \rangle\right| \ge b_i^{(t)}} = 0] \ge 1 - O(\frac{1}{\sqrt{d}})$, we have

$$\langle w_i^{(t+1)}, \mathbf{M}_{j'} \rangle \le \langle w_i^{(t)}, \mathbf{M}_{j'} \rangle + O(\frac{\eta}{d^{1.5}}) \langle v_i^{(t)}, \mathbf{H}_{j'} \rangle \tag{133}$$

and

$$|\langle w_i^{(t)}, \mathbf{M}_{j'} \rangle| \le \left( 1 + O(\frac{\eta}{d^{1.5}}) \right)^t \left( \frac{\langle w_i^{(T_2)}, \mathbf{M}_{j'} \rangle + \langle v_i^{(T_2)}, \mathbf{H}_{j'} \rangle}{2} \right) \tag{134}$$

At $T_C = \Theta\left(\frac{d\log(d)}{\eta}\right)$, we have:

$$\frac{|\langle w_i^{(T_C)}, \mathbf{M}_j\rangle|}{|\langle w_i^{(T_C)}, \mathbf{M}_{j'}\rangle|} > \frac{\left(1 + \eta\frac{C_z\cdot(1-\alpha_t^2)}{d}\right)^{T_C}}{\left(1 + O(\frac{\eta}{d^{1.5}})\right)^{T_C}} \geq \Omega(d) \tag{135}$$

Therefore, we summarize that when $t = T_C$, the alignment with the true feature $\mathbf{M}_j$ dominates, satisfying:

$$\frac{|\langle w_i^{(T_C)}, \mathbf{M}_j\rangle|}{|\langle w_i^{(T_C)}, \mathbf{M}_{j'}\rangle|} \geq \Omega(d), \tag{136}$$

highlighting the significant separation between the true feature $\mathbf{M}_j$ and the spurious feature $\mathbf{M}_{j'}$ for neurons $i \in S_{j,\text{sure}}$. A similar result holds for $v_i$, where the alignment with the true feature $\mathbf{H}_j$ similarly dominates over the spurious feature $\mathbf{H}_{j'}$.

### F.2. Convergence

For $i \in S_{j,\text{sure}}$, when $x_p, y_p$ contains the true feature $j$, the indicator functions remain consistently activated. Consequently, the loss function $L_C$ becomes convex with respect to $w_i$ and $v_i$ independently. We verify that the following inequality holds

$$L_{C,j}(w_i^{(t+1)}, v_i^{(t+1)}) \leq L_{C,j}(w_i^{(t)}, v_i^{(t)}) + \left\langle \nabla L_{C,j}(w_i^{(t)}, v_i^{(t)}), \left(w_i^{(t+1)} - w_i^{(t)}, v_i^{(t+1)} - v_i^{(t)}\right)\right\rangle + \frac{l_i}{2}\left\|\left(w_i^{(t+1)} - w_i^{(t)}, v_i^{(t+1)} - v_i^{(t)}\right)\right\|^2 \tag{137}$$

where $l_i = O(C_z d^{2\gamma c_0})(\left\|v_i^{(t)}\right\|_2^2 \|x_p\|_2^2 + \left\|v_i^{(t)}\right\|_2^2 \|x_p\|_2^2) = \Theta(1)$. This means $L_{C,j}(w_i^{(t)}, v_i^{(t)})$ is $l_i$-smooth for all $i \in S_{j,\text{sure}}$ when $x_p, y_p$ contains the true feature $j$. Let $L = \max_{i\in m}(l_i) = \Theta(1)$

Let $\eta = \frac{1}{L}$ to ensure a monotonic decrease, plug Eq (119) and Eq (120) into Eq (137), we have

$$L_{C,j}(w_i^{(t+1)}, v_i^{(t+1)}) \leq L_{C,j}(w_i^{(t)}, v_i^{(t)}) - \frac{\eta}{2}\|\nabla L_{C,j}(w_i^{(t)}, v_i^{(t)})\|^2. \tag{138}$$

By the property of smoothness, we have

$$\|\nabla L_{C,j}(w_i^{(t)}, v_i^{(t)})\|_2^2 \geq \frac{2}{L}\left(L_{C,j}(w_i^{(t)}, v_i^{(t)}) - L_{C,j}(w_i^*, v_i^*)\right). \tag{139}$$

Take the telescope sum of from 0 to $T_C$, we have

$$\begin{aligned}\frac{1}{T_C}\sum_{t=0}^{T_C} L_{C,j}(w_i^{(t)}, v_i^{(t)}) &\leq L_{C,j}(w_i^*, v_i^*) + \frac{L^2\Delta_0}{T_C}\\ &\overset{\diamondsuit}{\leq} L_{C,j}(w_i^*, v_i^*) + \Theta(\frac{1}{d})\\ &\overset{\heartsuit}{=} \Theta(\frac{1}{d})\end{aligned} \tag{140}$$

where $\Delta_0 = L_{C,j}(w_i^{(0)}, v_i^{(0)}) - L_{C,j}(w_i^*, v_i^*)$. In $\diamondsuit$, we use $T_C = \Theta(d)$, and $\|w_i^{(t)}\|_2^2 = \|v_i^{(t)}\|_2^2 = \Theta(1)$. In $\heartsuit$, we use $w_i^* = \alpha_{i,j}^*\mathbf{M}_j, V_i^* = \alpha_{i,j}^*\mathbf{H}_j$ and $L_{C,j}(w_i^*, v_i^*) = \Theta(\frac{1}{d})$ if $x_p$ contains the true feature $\mathbf{M}_j$.

Therefore, for all $j \in d$ and all $(x_p, y_p) \in S_h$, when $T_C = \Theta(d^2)$, we can ensure

$$L_C = \mathbb{E}_{(x_p, y_p)\in S_h}\left[\frac{1}{2}\left\|\mathbf{V}^T\sigma(\mathbf{W}x_p) - y_p\right\|_2^2\right] \leq \Theta(\frac{1}{d}) \tag{141}$$

### F.3. Summary

After $T_C$ iterations, the parameters $\mathbf{W}$ and $\mathbf{V}$ are updated to $\mathbf{W}^{T_C} = \hat{\mathbf{W}}$ and $\mathbf{V}^{T_C} = \hat{\mathbf{V}}$, respectively, using the dataset $S_h$. The generated caption is given by:

$$\hat{y}_p = \hat{\mathbf{V}}^T\sigma(\hat{\mathbf{W}}x_p), \tag{142}$$

where the expected loss satisfies:

$$\mathbb{E}\left[\frac{1}{2}\|\hat{y}_p - y_p\|_2^2\right] = L_C \leq \Theta\left(\frac{1}{d}\right). \tag{143}$$

1. For $i \in S_{j,\text{sure}}$, the alignment strength satisfies:

$$|\langle w_i^{(T_C)}, \mathbf{M}_j \rangle|^2 = \Theta(1)\left\|w_i^{(T_C)}\right\|_2^2 \tag{144}$$

and

$$|\langle w_i^{(T_C)}, \mathbf{M}_j' \rangle|^2 \leq O(\frac{1}{d})\left\|w_i^{(T_C)}\right\|_2^2 \tag{145}$$

where $j'$ represents the corresponding spurious alignment feature.

2. For $i \notin S_{j,\text{pot}}$, the alignment strength satisfies:

$$|\langle w_i^{(T_1)}, \mathbf{M}_j \rangle|^2 \leq O(\frac{1}{d})\left\|w_i^{(T_C)}\right\|_2^2 \tag{146}$$

3. For $\mathbf{M}_j^\perp$ where $j \in [d_1] \setminus [d]$, we have:

$$|\langle w_i^{(t+1)}, \mathbf{M}_j^\perp \rangle|^2 < O(\frac{1}{d_1})\left\|w_i^{(T_C)}\right\|_2^2 \tag{147}$$

## G. Filtering

During filtering, we sample the synthetic image-text pair $(x_p, \hat{y}_p)$ in $\hat{S}_w$ and the corresponding image-text pair $(x_p, y_p)$ in $S_w$. The image encoder $f$ and text encoder $h$ trained on raw data are employed to obtain the corresponding embeddings.

$$z'_{x_p} = f(x_p), \quad \hat{z}_{y_p} = h(\hat{y}_p), \quad z'_{y_p} = h(y_p) \tag{148}$$

Then, we calculate the cosine similarity of $\langle z'_{x_p}, \hat{z}_{y_p} \rangle$ and $\langle z'_{x_p}, z'_{y_p} \rangle$, and select the image-text pair with higher cosine similarity denoted as $(x, \tilde{y})$. In this way, we replace the noisy pairs in $S_w$ with synthetic pairs in $\hat{S}_w$. The resulting dataset is denoted as $\tilde{S} = \tilde{S}_w \cup S_h$.

The decoder generates synthetic captions $\hat{y}_p = \hat{\mathbf{V}}^T \sigma(\hat{\mathbf{W}} x_p)$. Using Eq (143), for each data pair $(x_p, y_p)$ which contain feature $(\mathbf{M}_j, \mathbf{H}_j)$ in $S_h$ we have

$$\mathbb{E}_{(x_p, y_p)}\left[\mathbb{E}_{j \in d}\left[\frac{1}{2}\left\|\mathbf{H}_j z_{\hat{y}_p}^j - \mathbf{H}_j z_{y_p}^j\right\|_2^2\right]\|z_{y_p}^j| = 1\right] \leq \mathbb{E}_{(x_p, y_p)}\left[\frac{1}{2}\|\hat{y}_p - y_p\|_2^2\|z_{y_p}^j| = 1\right] = L_C \leq \Theta(\frac{1}{d}) \tag{149}$$

Therefore, using $\|\mathbf{H}_j\|_2 = 1$ and $z_{x_p} = z_{y_p}$ in $S_h$, we have

$$\mathbb{E}_{x_p, j \in d}\left[z_{\hat{y}_p}^j z_{x_p}^j \| z_{x_p}^j| = 1\right] \geq 1 - \Theta(\frac{1}{d}) \tag{150}$$

Base on Assumption 8 $z_{x_p}^j \sim \text{Bernoulli}\left(\frac{C_z}{d}\right)$, we have

$$\Pr(z_{\hat{y}_p}^j = 1 \mid |z_{x_p}^j| = 1) \geq 1 - \Theta(\frac{1}{d}) \tag{151}$$

Using Eq (136) and Eq (151), we have

$$\Pr(z_{\hat{y}_p}^{j'} = 1 \mid |z_{x_p}^j| = 1) \leq \Theta(\frac{1}{d}) \tag{152}$$

Therefore, after replace all noisy text $y_p$ in $S_w$ by synthetic caption $\hat{y}_p$ in $\hat{S}_w$

1. for a positive pair $(x_p, y_p)$, we have

$$\mathbb{E}\left[z_{\tilde{x}_p}^j z_{\tilde{y}_p}^j\right] = \Theta(\frac{1}{d}), \quad \mathbb{E}\left[z_{\tilde{x}_p}^j z_{\tilde{y}_p}^{j'}\right] = \Theta\left(\frac{1}{d^2}\right), \quad \forall j' \neq j. \tag{153}$$

2. for negative pairs $(x_p, y_q)$, where $p \neq q$, we have:

$$\mathbb{E}\left[z_{x_p}^j z_{y_q}^{j'}\right] = \Theta\left(\frac{1}{d^2}\right), \quad \forall j, j' \in [d]. \tag{154}$$

## H. ITCP on Synthetic Data

During ITCP on Raw Data, we use a noisy dataset $S$. Based on Subsection A.1, we have $\mathbb{E}\left[z_x^j z_y^j\right]$ and $\mathbb{E}\left[z_x^j z_y^{j'}\right]$ both in $\Theta\left(\frac{1}{d}\right)$. In this scenario, for $i \in S_{j,\text{sure}}$, $w_i^{(t)}$ is jointly influenced by $\mathbf{M}_j$ and $\mathbf{M}_{j'}$, with both features contributing comparably to the updates. However, during ITCP on Data, we sample image-text pairs from the dataset $\tilde{S}$. Using Eq. (153), we find that $\mathbb{E}\left[z_{\tilde{x}_p}^j z_{\tilde{y}_p}^{j'}\right] = \Theta\left(\frac{1}{d^2}\right)$. In this case, for $i \in S_{j,\text{sure}}$, $w_i^{(t)}$ is influenced solely by $\mathbf{M}_j$, without interference from spurious features, ensuring purified representations.

The only difference between ITCP on Raw Data and Data lies in the $\mathbb{E}\left[z_{\tilde{x}_p}^j z_{\tilde{y}_p}^{j'}\right]$; all other training processes remain largely the same. Therefore, we simplify our proof accordingly.

### H.1. Stage I of ITCP on Data

The stage I of ITCP on Data is defined as the training iterations $t \leq T_1$, where $T_1 = \Theta\left(\frac{d \log d}{\eta}\right)$ is the iteration when all $\|w_i^{(T_2)}\|_2^2 = 2\|w_i^{(0)}\|_2^2$. Before $T_1$, we set $b_i^{(t)} = 0$. For every neuron $i \in [m]$, the weights $w_i$, $v_i$ will mostly ignore the noise features $\mathbf{M}^\perp$, $\mathbf{H}^\perp$ and learn to emphasize the features $\mathbf{M}$, $\mathbf{H}$.

If $\Pr(|z_{y_p}^j| = 1 \mid |z_{x_p}^{j'}| = 1) < 0.1$, we have $\mathbb{E}\left[z_x^j z_y^j\right] \gg \mathbb{E}\left[z_x^j z_y^{j'}\right]$ and $(a+b+c)^t \approx (a+b-c)^t$. In this case, $w_i^{(t+1)}$ is predominantly influenced by $\mathbf{M}_j$, with minimal contributions from $\mathbf{M}_{j'}$. The updates are thus primarily driven by the single feature $\mathbf{M}_j$, ensuring that spurious interactions from $\mathbf{M}_{j'}$ are negligible.

$$\|\mathbf{M}\mathbf{M}^\top w_i^{(t)}\|_2^2 = \sum_{i=1}^d \left[\frac{(a+b+c)^t}{2}\left(\langle w_i^{(t)}, \mathbf{M}_j\rangle + \langle v_i^{(t)}, \mathbf{H}_j\rangle\right)\right]^2$$
$$= \left(1 + \frac{\eta C_z}{d}\right)^{2t} \frac{\|\mathbf{M}\mathbf{M}^\top w_i^{(0)}\|_2^2 + \|\mathbf{H}\mathbf{H}^\top v_i^{(0)}\|_2^2}{4}. \tag{155}$$

$i \in S_{j,\text{sure}}$:

$$|\langle w_i^{(T_1)}, \mathbf{M}_j\rangle|^2 = \left(1 + \eta\frac{C_z}{d}\right)^{2T_1}\left(\frac{\langle w_i^{(0)}, \mathbf{M}_j\rangle + \langle v_i^{(0)}, \mathbf{H}_j\rangle}{2}\right)^2$$
$$\geq \left(1 + \eta\frac{C_z}{d}\right)^{2T_1} \cdot \frac{c_1 \log d}{d} \cdot \frac{\|\mathbf{M}\mathbf{M}^\top w_i^{(0)}\|_2^2 + \|\mathbf{H}\mathbf{H}^\top v_i^{(0)}\|_2^2}{4}$$
$$= \frac{c_1 \log d}{d} \cdot \frac{\|\mathbf{M}\mathbf{M}^\top w_i^{(T_1)}\|_2^2 + \|\mathbf{H}\mathbf{H}^\top v_i^{(T_1)}\|_2^2}{2} \tag{156}$$
$$\geq \frac{c_1 \log d}{d} \cdot \frac{\|w_i^{(T_1)}\|_2^2 + \|v_i^{(T_1)}\|_2^2 - \|w_i^{(0)}\|_2^2 - \|v_i^{(0)}\|_2^2}{2}$$
$$\geq \frac{(1+c_0) \log d}{d} \cdot \frac{\|w_i^{(T_1)}\|_2^2 + \|v_i^{(T_1)}\|_2^2}{2}$$

Because $\frac{\|w_i^{(T_1)}\|_2^2 + \|v_i^{(T_1)}\|_2^2}{2} = \|w_i^{(0)}\|_2^2 + \|v_i^{(0)}\|_2^2$ and $c_1 > 2(1+c_0)$

$i \notin S_{j,\text{sure}}$:

$$
\begin{aligned}
|\langle w_i^{(T_1)}, \mathbf{M}_j\rangle|^2 &= \left(1 + \eta\frac{C_z}{d}\right)^{2T_1}\left(\frac{\langle w_i^{(0)}, \mathbf{M}_j\rangle + \langle v_i^{(0)}, \mathbf{H}_j\rangle}{2}\right)^2 \\
&\leq \left(1 + \eta\frac{C_z}{d}\right)^{2T_1} \cdot \frac{c_2 \log d}{d} \cdot \frac{\|\mathbf{M}\mathbf{M}^\top w_i^{(0)}\|_2^2 + \|\mathbf{H}\mathbf{H}^\top v_i^{(0)}\|_2^2}{4} \\
&= \frac{c_2 \log d}{d} \cdot \frac{\|\mathbf{M}\mathbf{M}^\top w_i^{(T_1)}\|_2^2 + \|\mathbf{H}\mathbf{H}^\top v_i^{(T_1)}\|_2^2}{2} \\
&\leq \frac{\log d}{d} \cdot \frac{\|w_i^{(T_1)}\|_2^2 + \|v_i^{(T_1)}\|_2^2}{2}
\end{aligned}
\tag{157}
$$

$|\langle w_i^{(t+1)}, \mathbf{M_j^\perp}\rangle|^2 \leq O(\frac{1}{d_1})\frac{\|w_i^{(T_1)}\|_2^2 + \|v_i^{(T_1)}\|_2^2}{2}$

## H.2. Stage II:

The Stage II of ITCP on Data is defined as the training iterations $t \leq T_2$, where $T_2 - T_1 = \Theta\left(\frac{d \log d}{\eta}\right)$ is the iteration.

We set $b_i^{(t)} = \sqrt{\frac{\log d}{d} \cdot \frac{\|w_i^{(T_1)}\|_2^2 + \|v_i^{(T_1)}\|_2^2}{2}}$ and $b_i^{(t+1)} = (1 + \frac{\eta}{d})b_i^{(t)}$ until all $\|\|w_i^{(T_2)}\|_2 \geq \Omega(d)\|w_i^{(T_1)}\|_2,$. In this phase, the weights $(w_i, v_i)$ will mostly ignore the features $\mathbf{M}_j$, $\mathbf{H}_j$ if $i \notin S_{j,\text{sure}}$ and the noise features $\mathbf{M}^\perp$, $\mathbf{H}^\perp$, and learn to emphasize the features $\mathbf{M}_j$, $\mathbf{H}_j$ if $i \in S_{j,\text{sure}}$.

For $i \in S_{j,\text{sure}}$, using Lemma B.5, the following holds with high probability $1 - e^{-\Omega(d_1)}$ when $T_1 < t \leq T_2$ :

$$
\left|\langle w_i^{(t)}, \xi\rangle\right|^2 \leq O\left(\frac{\left\|w_i^{(t)}\right\|_2^2}{d^{1+c_0}}\right) < b_i^{(t)}
\tag{158}
$$

Under the assumption that, with high probability, the indicator function satisfies the condition when $t = T_1$:

$$
\mathbf{1}_{\left|\left\langle w_i^{(t)}, x_p\right\rangle\right| \geq b_i^{(t)}} \cdot \mathbf{1}_{\left|\left\langle v_i^{(t)}, y_p\right\rangle\right| \geq b_i^{(t)}} = 1,
\tag{159}
$$

we can ensure that:

$$
\mathbb{E}\left[z_x^j z_y^j \cdot \mathbf{1}_{\left|\left\langle w_i^{(t)}, x_p\right\rangle\right| \geq b_i^{(t)}} \cdot \mathbf{1}_{\left|\left\langle v_i^{(t)}, y_p\right\rangle\right| \geq b_i^{(t)}}\right] = \frac{C_z}{d}.
\tag{160}
$$

The weight dynamics for $|\langle w_i^{(t+1)}, \mathbf{M}_j\rangle|$ can be expressed as:

$$
|\langle w_i^{(t+1)}, \mathbf{M}_j\rangle| = \left(1 + \eta\frac{C_z}{d}\right)\left(\frac{\langle w_i^{(t)}, \mathbf{M}_j\rangle + \langle v_i^{(t)}, \mathbf{H}_j\rangle}{2}\right).
\tag{161}
$$

Given that $\left(1 + \eta\frac{C_z}{d}\right) > \left(1 + \frac{\eta}{d}\right)$, and $\frac{\langle w_i^{(t)}, \mathbf{M}_j\rangle + \langle v_i^{(t)}, \mathbf{H}_j\rangle}{2} > b_i^{(t)}$, it follows that:

$$
|\langle w_i^{(t+1)}, \mathbf{M}_j\rangle| > b_i^{(t+1)}.
\tag{162}
$$

Thus, with high probability, for $t \leq T_2$, we have:

$$
\mathbf{1}_{\left|\left\langle w_i^{(t)}, x_p\right\rangle\right| \geq b_i^{(t)}} \cdot \mathbf{1}_{\left|\left\langle v_i^{(t)}, y_p\right\rangle\right| \geq b_i^{(t)}} = 1.
\tag{163}
$$

so for $T_1 < t \leq T_2$ we have

$$
|\langle w_i^{(t+1)}, \mathbf{M}_j\rangle| = \left(1 + \eta\frac{C_z}{d}\right)^t\left(\frac{\langle w_i^{(T_1)}, \mathbf{M}_j\rangle + \langle v_i^{(T_1)}, \mathbf{H}_j\rangle}{2}\right)
\tag{164}
$$

For $i \notin S_{j,\text{sure}}$, the projection of weights onto a generic feature $\xi$ at iteration $T_1$ satisfies:

$$\Pr\left(\mathbf{1}_{\left|\left\langle w_i^{(t)}, x_p \right\rangle\right| \geq b_i^{(t)}} \cdot \mathbf{1}_{\left|\left\langle v_i^{(t)}, y_p \right\rangle\right| \geq b_i^{(t)}} = 1\right) \leq o\left(\frac{1}{d}\right). \tag{165}$$

We can ensure that:

$$\mathbb{E}\left[z_x^j z_y^j \cdot \mathbf{1}_{\left|\left\langle w_i^{(t)}, x_p \right\rangle\right| \geq b_i^{(t)}} \cdot \mathbf{1}_{\left|\left\langle v_i^{(t)}, y_p \right\rangle\right| \geq b_i^{(t)}}\right] = o\left(\frac{1}{d^2}\right). \tag{166}$$

The weight dynamics for $|\langle w_i^{(t+1)}, \mathbf{M}_j\rangle|$ can now be expressed as:

$$|\langle w_i^{(t+1)}, \mathbf{M}_j\rangle| = \left(1 + o\left(\frac{\eta}{d^2}\right)\right)\left(\frac{\langle w_i^{(t)}, \mathbf{M}_j\rangle + \langle v_i^{(t)}, \mathbf{H}_j\rangle}{2}\right). \tag{167}$$

Given that $\left(1 + o\left(\frac{\eta}{d^2}\right)\right) < \left(1 + \frac{\eta}{d}\right)$, and $\frac{\langle w_i^{(t)}, \mathbf{M}_j\rangle + \langle v_i^{(t)}, \mathbf{H}_j\rangle}{2} < b_i^{(t)}$, it follows that:

$$|\langle w_i^{(t+1)}, \mathbf{M}_j\rangle| < b_i^{(t+1)}. \tag{168}$$

If $|\langle w_i^{(T_1)}, \mathbf{M}_j\rangle| < b_i^{(T_1)}$, then $|\langle w_i^{(t)}, \mathbf{M}_j\rangle| < b_i^{(t)}$ for $t \leq T_2$. Thus, with high probability, for $t \leq T_2$, we have:

$$\mathbf{1}_{\left|\left\langle w_i^{(t)}, x_p \right\rangle\right| \geq b_i^{(t)}} \cdot \mathbf{1}_{\left|\left\langle v_i^{(t)}, y_p \right\rangle\right| \geq b_i^{(t)}} = 0. \tag{169}$$

$$|\langle w_i^{(t+1)}, \mathbf{M}_j\rangle| \leq \left(1 + o\left(\frac{\eta}{d^2}\right)\right)^t \left(\frac{\langle w_i^{(T_1)}, \mathbf{M}_j\rangle + \langle v_i^{(T_1)}, \mathbf{H}_j\rangle}{2}\right) \tag{170}$$

There exists $T_2 = \Theta\left(\frac{d \log d}{\eta}\right)$ such that the following conditions hold:

$$\left(1 + \eta\frac{C_z}{d}\right)^{T_2} = \Theta(d), \tag{171}$$

indicating that $|\langle w_i^{(t+1)}, \mathbf{M}_j\rangle|$ for $i \in S_{j,\text{sure}}$ increase iteratively until:

$$\|w_i^{(T_2)}\|_2 \geq \Omega(d)\|w_i^{(T_1)}\|_2 \tag{172}$$

while, for $i \notin S_{j,\text{sure}}$, the updates diminish, such that:

$$\left(1 + o\left(\frac{\eta}{d^2}\right)\right)^{T_2} \leq 1 + o\left(\frac{1}{d}\right), \tag{173}$$

indicating negligible growth in $|\langle w_i^{(t+1)}, \mathbf{M}_j\rangle|$.

Thus we have

$$\begin{aligned}
|\langle w_i^{(T_2)}, \mathbf{M}_j\rangle|^2 &= \|w_i^{(T_2)}\|_2^2 - \sum_{j \in [d], j \notin \mathcal{N}_i} \langle w_i^{(T_2)}, \mathbf{M}_j\rangle^2 - \sum_{j \in [d_1]\setminus[d]} \langle w_i^{(T_2)}, \mathbf{M}_j^\perp\rangle^2 \\
&\geq \|w_i^{(T_2)}\|_2^2 - (1 + o(1))\|w_i^{(T_1)}\|_2^2 - (1 + o(1))\|w_i^{(0)}\|_2^2 \\
&\geq (1 - o(1))\|w_i^{(T_2)}\|_2^2.
\end{aligned} \tag{174}$$

Finally, for $i \notin S_{j,\text{sure}}$, we have:

$$\|w_i^{(T_2)}, \mathbf{M}_j\|_2 \leq (1 + o(\frac{1}{d})) \cdot O\left(\frac{\|w_i^{(T_1)}\|_2}{\sqrt{d}}\right) \leq O\left(\frac{\|w_i^{(T_2)}\|_2}{\sqrt{d}}\right), \tag{175}$$

and for noise components:

$$|\langle w_i^{(T_2)}, \mathbf{M}_j^{\perp}\rangle|_2 \leq O\left(\frac{\|w_i^{(T_2)}\|_2}{\sqrt{d_1}}\right). \tag{176}$$

We summarize the results when $T_1 < t \leq T_2$ as follows:

1. For $i \in S_{j,\text{sure}}$, the alignment strength satisfies:

$$|\langle w_i^{(T_2)}, \mathbf{M}_j\rangle|^2 > (1 - o(1))\frac{\|w_i^{(T_2)}\|_2^2 + \|v_i^{(T_2)}\|_2^2}{2} \tag{177}$$

without $j'$ that represents the corresponding spurious alignment feature.

2. For $i \notin S_{j,\text{pot}}$, the alignment strength satisfies:

$$|\langle w_i^{(T_1)}, \mathbf{M}_j\rangle|^2 \leq O(\frac{1}{d}) \cdot \frac{\|w_i^{(T_2)}\|_2^2 + \|v_i^{(T_2)}\|_2^2}{2} \tag{178}$$

3. For $\mathbf{M}_j^{\perp}$ where $j \in [d_1] \setminus [d]$, we have:

$$|\langle w_i^{(t+1)}, \mathbf{M}_j^{\perp}\rangle|^2 < O\left(\frac{1}{d_1}\right) \cdot \frac{\|w_i^{(T_2)}\|_2^2 + \|v_i^{(T_2)}\|_2^2}{2}. \tag{179}$$

Similar results also hold for $v_i$.

### H.3. Stage III Convergence of ITCP on Data

Similarly to convergence stage in ITCP on Raw Data, using Eq (31), Eq (34), and Eq (35), the loss function $L$ becomes convex with respect to $w_i$ and $v_i$ independently when $(x_p, y_p)$ and $(x_n, y_n)$ contain the true feature $j$.

We verify that the following inequality holds

$$L_j(w_i^{(t+1)}, v_i^{(t+1)}) \leq L_j(w_i^{(t)}, v_i^{(t)}) + \left\langle \nabla L_j(w_i^{(t)}, v_i^{(t)}), (w_i^{(t+1)} - w_i^{(t)}, v_i^{(t+1)} - v_i^{(t)}) \right\rangle + \frac{l_{i,j}}{2}\left\|(w_i^{(t+1)} - w_i^{(t)}, v_i^{(t+1)} - v_i^{(t)})\right\|^2 \tag{180}$$

Let $L = \max_{i \in m}(l_{i,j}/(2\tau)) = \Theta(1)$ and $\eta = \frac{1}{L}$ to ensure a monotonic decrease, plug Eq (32) and Eq (33) into Eq (180), we have

$$L_j(w_i^{(t+1)}, v_i^{(t+1)}) \leq L_j(w_i^{(t)}, v_i^{(t)}) - \frac{\eta}{2}\|\nabla L_j(w_i^{(t)}, v_i^{(t)})\|^2. \tag{181}$$

Under our data assumptions for $S_w$ and conclusion in Eq (100), we define $w_i^* = \alpha_{i,j}^* \mathbf{M}_j, v_i^* = \alpha_{i,j}^* \mathbf{H}_j$. Thus, $L_j(w_i^*, v_i^*)$ captures both the alignment with the true feature $\mathbf{M}_j, \mathbf{H}_j$ and the spurious feature $\mathbf{M}_{j'}, \mathbf{H}_{j'}$, representing the minimal loss achievable under the influence of both true and spurious features in the optimization process. Using Eq (85), we know $w_i^{(T_2)} = \Theta(d)$, so $L_j(w_i^*, v_i^*) = -\Theta(d)$.

By the property of smoothness, we have

$$\|\nabla L_j(w_i^{(t)}, v_i^{(t)})\|_2^2 \geq \frac{2}{L}\left(L_j(w_i^{(t)}, v_i^{(t)}) - L_j(w_i^*, v_i^*)\right) \tag{182}$$

Take the telescope sum of from $T_2$ to $T_3$, we have

$$\frac{1}{T_3 - T_2}\sum_{t=T_2}^{T_3} L_j(w_i^{(t)}, v_i^{(t)}) \leq L_j(w_i^*, v_i^*) + \frac{L^2 \Delta_0}{T_3 - T_2} \tag{183}$$

$$\overset{\diamond}{\leq} L_j(w_i^*, v_i^*) + \Theta(1)$$

where $\Delta_0 = L_j(w_i^{(T_1)}, v_i^{(T_1)}) - L_j(w_i^*, v_i^*) = \Theta(1)$. In $\diamond$, we use $T_2 = \Theta(d)$, and $L = \Theta(\frac{1}{d})$.

Generalized to every $j \in d$, the same convergence holds for all $i \in S_{j,\text{sure}}$ when $(x_p, y_p)$ and $(x_n, y_n)$ contain feature $j, j'$. For all $(x_p, y_p)$ and $(x_n, y_n)$ in $S_w$, the following inequality holds:

$$\frac{1}{T_3 - T_2} \sum_{t=T_2}^{T_3} L(f^{(T_3)}, h^{(T_3)}) \leq L(f^*, h^*) + \Theta(1). \tag{184}$$

## I. Downstream Task

In the zero-shot downstream task, we consider a multi-class classification task given image data $x$. The task involves $K = \Theta(1)$ text prompts defined as $\{y_k : y_k = \mathbf{H}z_{y_k} + \xi_{y_p}, k \in [K]\}$, where each prompt corresponds to one of $K$ classes. The entries of the $\{z_{y_k}\}$ are binary and $z_{y_k}^j \sim \text{Bernoulli}\left(\Theta\left(\frac{1}{d}\right)\right)$, i.e., each $(z_{y_k})_i \in \{0, 1\}$ and $\|z_{y_k}\|_0 = \Theta(1)$. The goal is to classify $x$ into the class associated with the text prompt $y_k$ that best matches $x$.

For a given image data $x = \mathbf{M}z_x + \xi_x$, where $z_x^j \sim \text{Bernoulli}\left(\left(\frac{C_z'}{d}\right)\right)$, the goal is to predict its label among $K$ classes.

Using Eq. (100) and Eq. (146), let $f(x)$ and $h(y)$ represent the image encoder and text encoder of ITCP on Raw Data, respectively. Given a data sample $x$ containing $\mathbf{M}_j$ and $y$ containing $\mathbf{H}_{j'}$, where $j'$ is the spurious feature corresponding to $j$, it holds with high probability that:

$$\left\langle \frac{f(x)}{\|f(x)\|_2}, \frac{h(y)}{\|h(y)\|_2} \right\rangle = \Theta(1). \tag{185}$$

This result implies that the image and text encoders of ITCP on Raw Data struggle to distinguish between features $j$ and $j'$, leading to misclassification caused by spurious correlations.

However, using Eq. (177) and Eq. (178), $\tilde{f}(x)$ and $\tilde{g}(y_k)$ represent the image encoder and text encoder of ITCP on Data, respectively. Given a data sample $x$ containing $\mathbf{M}_j$ and $y$ containing $\mathbf{H}_{j'}$, where $j'$ is the spurious feature corresponding to $j$, it holds with high probability $1 - \Theta\left(\frac{1}{d}\right)$ that:

$$\left\langle \frac{\tilde{f}(x)}{\|\tilde{f}(x)\|_2}, \frac{\tilde{g}(y)}{\|\tilde{g}(y)\|_2} \right\rangle \leq \Theta\left(\frac{1}{d}\right). \tag{186}$$

This result implies that the image and text encoders of ITCP on synthetic Data are capable of effectively distinguishing the true feature from the spurious feature.

Because $K = \Theta(1)$ and $\|z_{y_k}\|_0 = \Theta(1)$, we only have constant class classification and constant features in images. Thus, we have:

1. For the image encoder $f(x)$ and text encoder $h(y_k)$ of ITCP on Raw Data, it holds that:

$$\Pr\left(\arg\max_k \langle f(x), h(y_k) \rangle = k_x\right) = 1 - \Theta(1) \tag{187}$$

2. For the image encoder $\tilde{f}(x)$ and text encoder $\tilde{g}(y_k)$ of ITCP on synthetic Data, it holds that:

$$\Pr\left(\arg\max_k \langle \tilde{f}(x), \tilde{g}(y_k) \rangle = k_x\right) = 1 - o(1) \tag{188}$$

## J. Experiment

**Appendix: Analysis of Image Feature Representation.** As shown in Figure 7, the alignment properties of the image projection weights differ significantly between BLIP and ALBEF. Specifically, $|\langle \tilde{w}_i, \mathbf{M}_j \rangle|$ in BLIP exhibits a more concentrated distribution, demonstrating improved orthogonality and alignment compared to ALBEF.

Appendix Figure 7 provides histograms comparing the distribution of $|\langle w_i, \mathbf{M}_j \rangle|$ values in the image projection layers of BLIP and ALBEF:

1. **Left Subplot:** Focuses on lower value ranges (0 to 0.4), showing that BLIP reduces spurious correlations, concentrating values around smaller scores.

2. **Right Subplot:** Highlights higher value ranges (0.4 to 0.8), demonstrating that BLIP increases alignment, concentrating values around 0.6 to 0.7.

These observations are consistent with Theorem 4.5, which indicates that training on synthetic data promotes orthogonality and alignment, enabling neurons to specialize in distinct image features and improving feature representation.

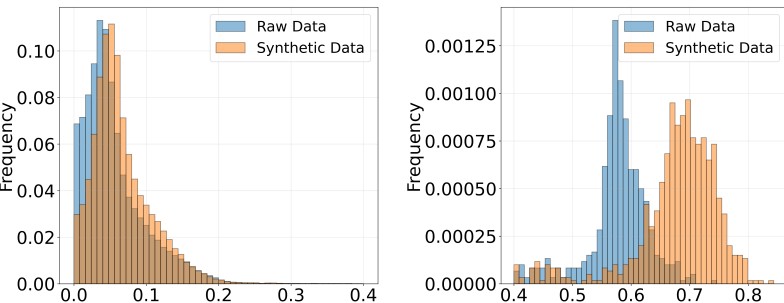

Figure 6: Histogram of $|\langle \bar{w}_i, \mathbf{M}_j \rangle|/|\bar{w}_i|$ for ITCP on raw data and $|\langle \tilde{w}_i, \mathbf{M}_j \rangle|/\tilde{w}_i$ for ITCP on synthetic data (split into two figures to highlight the significant differences in the value distributions).

**Image Feature Representation Learning.** The final image projection layer in BLIP/ALBEF shares functional similarities with $\mathbf{W}$ in our simplified model in (5). Similar to the analysis of text projection layers in Figure 5, Figure 7 presents the histogram of normalized inner products $\langle w_j, w_{j'} \rangle/(\|w_j\|\|w_{j'}\|)$ for all $j, j' \in \{1, 2, \ldots, 256\}$. The comparison shows that the weight vectors in BLIP exhibit a stronger concentration around zero, demonstrating a higher degree of orthogonality than those in ALBEF. This aligns with our theoretical results in Theorem 4.5, which suggest that training on synthetic data promotes orthogonality among weight vectors, enabling improved feature separation and alignment. Such orthogonality ensures that individual neurons specialize in learning distinct image features, thereby enhancing representation learning.

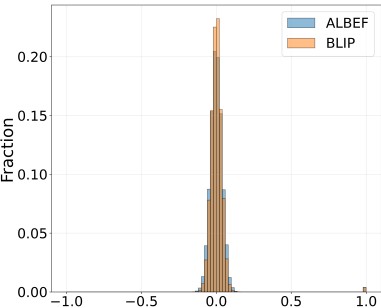

Figure 7: Histogram of $\langle w_j, w_{j'} \rangle/(\|w_j\|\|w_{j'}\|)$ of the image linear projection layer of ALBEF and BLIP.

