# OpenReview forum: "Theoretical Analysis of Contrastive Learning in Vision-Language Model Pretraining: The Role of Synthetic Text Captions for Feature Alignment"
_ICML.cc/2025/Conference — Submitted to ICML 2025_

### Official Review · Reviewer_yyfG · 2025-03-14

**Overall Recommendation:** 4

**Summary:**

The paper considers the theoretical analysis of the contrastive learning (of image-text pairs) in VLM pre-training. In particular, the paper considers the training dynamics (with potentially noisy and low-quality data), nonlinear activation (ReLU in the one-hidden-layer model), zero-shot generation of VLM, and the role and potential enhancement introduced by synthetic text captions. The theoretical findings suggest that carefully generated synthetic caption can potentially help "filter" (or, replace) spurious features in the raw data, and therefore, enhance VLM pre-training. Empirical results are also presented (e.g., visualizations based on BLIP).

---

### After Rebuttal

I confirm that I read authors' rebuttal, and also went through comments from other reviewers as well as the rebuttal (discussion) therein. While I lean towards the positive side, I recognize concerns and issues raised by other reviewers, and therefore do not champion the submission in its current shape.

**Claims And Evidence:**

The claims center around the role of synthetic caption in VLM pre-training. The evidence comes from both theoretical analyses (under certain assumptions, which are reasonably mild) and empirical observations (e.g., t-SNE and cos-similarity histogram visualizations).

**Essential References Not Discussed:**

The paper positions itself in the literature very well. The connection to and difference from very related previous works are carefully presented (e.g., Section 1.1) and concisely summarized (e.g., Table 1) at the same time.

**Experimental Designs Or Analyses:**

The experiments are based on simulation and real-world data/model. The empirical findings are consistent with the theoretical implications.

**Methods And Evaluation Criteria:**

The paper starts from theoretical characterization, followed by empirical experiments (simulation and real-world data/model). The evaluation criteria for the experiments include t-SNE visualization of feature embeddings, histogram of image-text cosine similarities.

**Other Comments Or Suggestions:**

The figure quality can be improved to enhance readability (e.g., legend font size).

**Other Strengths And Weaknesses:**

The strength of the paper comes from the theoretically-grounded understanding of the VLM pre-training, and also the potential benefit of utilizing carefully generated synthetic text captions.

The paper could be further enhanced by considering:

- When visualizing the difference of ITCP on raw/synthetic data, use example text-image pairs, in addition to t-SNE and histogram.

The visualizations (even for t-SNE, since the influence from the choice of hyperparameters) are not that direct, if compared to image-text pairs. It would be very helpful (or, to some extent, necessary) to include raw image-text pairs to help make the illustration more informative.

**Questions For Authors:**

Can image-text pairs be included in addition to visualizations based on t-SNE and histogram?

What is the errorbar of the cosine similarity for BLIP generation ("with a mean similarity of 0.26 compared to 0.24 for raw captions" in Section 5)?

**Relation To Broader Scientific Literature:**

The paper has a relatively broad scope of implications to VLM-related area.

**Theoretical Claims:**

The theoretical claims are (roughly) w.r.t. the relation between the pre-training and the data quality. The assumptions are reasonably mild (in the sense that they are abstracted from understandings of the underlying data generating process, and/or present in previous literature). The theoretical results themselves are clearly presented (e.g., under what assumption, serve which part of the goal).

---

> ### Author Rebuttal · Authors · 2025-04-01
>
> # Reviewer yyfG
> We thank the reviewer for the valuable time in the evaluation.
>
> ## Absence of Image-Text Pairs and Caption-Quality Metrics
> Although one of the contributions of this paper is the theoretical demonstration that a well-designed recaptioning process can yield high-quality data, the goal of our experiments is not to re-justify this insight, as it has already been extensively validated by empirical evidence in prior works [1,2,3]. Instead, we directly leverage their models and datasets to support our theoretical analysis. For example, Figure 4 and Figure 6 in [1] present image-text pairs, Appendix A of [2] provides further examples, and Appendix C of [3] includes numerous captions generated by LLMs.
>
> Moreover, Table 1 and Figure 4 in [2] empirically shows that higher cosine similarity between synthetic captions and image features correlates with improved ImageNet accuracy, thereby serving as a proxy caption-quality metric. Figure 2 in [2] further demonstrates the effectiveness of using a mixture of raw and synthetic captions filtered by cosine similarity. In addition, Section 4.3 of [1] discusses the diversity of synthetic captions and their benefits in pretraining.
>
> ## T-SNE not direct
> We agree that t-SNE visualization (influence from the choice of hyperparameters) is not direct, and does not provide statistical evidence for the separation quality between different methods. **To address this limitation, we adopt the Silhouette Score (SS) with cosine distance to quantitatively and statistically assess feature embedding quality.**
> We did a new experiment to compare vanilla CLIP (trained on original captions) with the LaCLIP model [3], where the model has the same architecture and the same training data with the vanilla CLIP, while the only difference is that a fraction of the captions are replaced with synthetic captions generated by LLM in training LaCLIP.
>
> Tables 1, 2, and 3 show the comparison between vanilla CLIP and LaCLIP on CIFAR-100, CIFAR-10, and Caltech-101 datasets, respectively. The results show that LaCLIP consistently achieve higher Silhouette Scores than their CLIP counterparts. Since we use cosine distance, a higher Silhouette Score indicates that feature embeddings within the same class are more aligned with high cosine similarity, and embeddings from different classes are more orthogonal with low cosine similarity. This provides quantitative evidence for Theorem 4.3 and 4.5, which show that the neurons can learn purified representations better when some captions are replaced with synthetic captions.
>
> Table 1: Comparison of CLIP and LaCLIP on CIFAR-100
> |Pre-training Dataset|Model|Accuracy(%)|SS|
> |---|---|---|---|
> |CC3M|CLIP|21.8|-0.0399±0.001|
> ||LaCLIP|**27.5**|**-0.0328±0.001**|
> |CC12M|CLIP|38.5|0.0051±0.001|
> ||LaCLIP|**43.9**|**0.0288±0.002**|
> |RedCaps|CLIP|39.9|$-0.0015±0.002|
> ||LaCLIP|**40.7**|**0.0114±0.002**|
> |LAION-400M|CLIP|71.7|0.0781±0.002|
> ||LaCLIP|**73.9**|**0.1081±0.002**|
>
> Table 2: Comparison of CLIP and LaCLIP on CIFAR-10
> |Pre-training Dataset|Model|Accuracy(%)|SS|
> |---|---|---|---|
> |CC3M|CLIP|54.9|$0.0194±0.001$|
> ||LaCLIP|**57.1**|**0.0364±0.001**|
> |CC12M|CLIP|64.9|0.1129±0.001|
> ||LaCLIP|**75.1**|**0.1565±0.001**|
> |RedCaps|CLIP|70.4|0.1002±0.001|
> ||LaCLIP|**74.8**|**0.1071±0.001**|
> |LAION-400M(ViT-B/16)|CLIP|93.0|0.1809±0.001|
> ||LaCLIP|**93.5**|**0.2145±0.001**|
>
> Table 3: Comparison of CLIP and LaCLIP on Caltech-101
> |Pre-training Dataset|Model|Accuracy(%)|SS|
> |---|---|---|---|
> |CC3M|CLIP|43.3|0.1295±0.003|
> ||LaCLIP|**52.7**|**0.1620±0.003**|
> |CC12M|CLIP|77.4|0.2252±0.003|
> ||LaCLIP|**83.3**|**0.2756±0.003**|
> |RedCaps|CLIP|72.8|0.2102±0.004|
> ||LaCLIP|**76.4**|**0.2327±0.004**|
> |LAION-400M|CLIP|91.2|0.2584±0.002|
> ||LaCLIP|**92.4**|**0.3063±0.002**|
>
> ## Errorbar of the cosine similarity
> BLIP-generated captions demonstrate higher semantic alignment with a mean cosine similarity of $0.2633$ ($σ=0.0373$) compared to raw captions at $0.2467$ ($σ=0.0472$).
>
> References:
>
> [1] Li et al., BLIP: Bootstrapped Language-Image Pretraining, 2022
>
> [2] Chen et al., Improving Multimodal Datasets with Image Captioning, 2023
>
> [3] Fan et al., Improving CLIP Training with Language Rewrites, 2023

---

### Official Review · Reviewer_C7ye · 2025-03-14

**Overall Recommendation:** 2

**Summary:**

This paper theoretically analyzes the issue of spurious correlations in Vision-Language Models (VLMs) trained via contrastive learning. It mathematically demonstrates that using synthetic text captions can enhance feature alignment and improve zero-shot performance by reducing these correlations. The key contribution lies in the theoretical modeling of contrastive learning dynamics, specifically addressing how high-quality (synthetic) captions facilitate better alignment and generalization. Empirical experiments comparing BLIP (with synthetic captions) and ALBEF (without synthetic captions) provide evidence supporting these claims.

**Claims And Evidence:**

The paper primarily makes three claims:
1. Contrastive learning models inherently learn spurious correlations from noisy captions.
2. High-quality or less noisy (Synthetic and Filtered) captions mitigate spurious correlations and enhance feature alignment.
3. High-quality or less noisy captions can improve zero-shot classification performance.
- While the theoretical analyses supporting these claims are rigorous, empirical evidence from experiments is limited and not entirely convincing. Specifically, the analyses shown in Figures such as t-SNE visualizations and cosine similarity histograms do not provide statistical tests demonstrating that the observed differences are marginal.
- Furthermore, the paper lacks explicit quantitative results regarding zero-shot classification accuracy, making it impossible to determine the practical magnitude or statistical significance of any claimed improvements between BLIP and ALBEF. This absence of clear numerical and statistical analysis undermines the empirical support for the theoretical claims.
- Additionally, the use of "synthetic" captions is quite misleading; both the theoretical analysis and experiments essentially compare filtered high-quality captions with low-quality (noisy) captions, rather than exploring differences related to the synthetic generation process.

**Essential References Not Discussed:**

Two essential theoretical papers addressing generalization differences due to noisy annotations in contrastive learning contexts are not cited or discussed.
- [1] provides a robust theoretical analysis of noisy labels' effects on contrastive learning generalization, offering insights into inductive biases necessary for robustness.
- [2] theoretically investigates how contrastive learning inherently enhances robustness against label noise and explains why learned representations are less sensitive to noisy labels.

The authors should discuss how its findings relate to and improve upon these existing analyses.


[1] Saunshi et al. "Understanding Contrastive Learning Requires Incorporating Inductive Biases". ICML 2022

[2] Xue et al. "Investigating Why Contrastive Learning Benefits Robustness Against Label Noise", ICML 2022

**Experimental Designs Or Analyses:**

The experimental design comparing BLIP and ALBEF models is conceptually sound. However, key limitations exist:
- Experiments do not explicitly quantify or statistically test the significance of performance differences.
- Experiments are restricted to relatively small-scale models (BLIP, ALBEF) and lack validation on widely used large-scale models like CLIP.
- The experiments primarily compare high-quality captions with low-quality (noisy) captions, rather than specifically analyzing the synthetic caption generation process itself. It could mislead readers into attributing improvements explicitly to the synthetic generation method rather than simply the presence of higher-quality captions.

**Methods And Evaluation Criteria:**

- The provided theoretical analysis is well-explained.
- However, evaluation criteria such as cosine similarity, t-SNE visualizations, and zero-shot classification accuracy are insufficient for conclusively determining caption quality or distinguishing between the effects of filtering versus caption quality.
- The clarity of the results is insufficient because of the absence of direct caption-quality metrics (e.g., diversity, human evaluation scores) and explicit quantitative evidence of improvements in feature alignment and zero-shot classification accuracy. Consequently, it remains unclear whether the proposed improvements are practically meaningful, statistically significant, and driven primarily by intrinsic caption quality or the filtering process.

**Other Comments Or Suggestions:**

See the above

**Other Strengths And Weaknesses:**

See the above

**Questions For Authors:**

See the above

**Relation To Broader Scientific Literature:**

- This work extends the theoretical analysis of contrastive learning into multimodal contexts and explicitly addresses multimodal alignment issues arising from noisy data. The paper clearly mentions several baseline theoretical works related to multimodal contrastive learning.
- However, while the paper clearly differentiates its contributions from existing multimodal contrastive learning theories, it does not adequately compare or distinguish its theoretical contributions from existing theoretical analyses specifically addressing generalization issues related to high-quality versus low-quality annotations, such as those discussed in [1] and [2].

[1] Saunshi et al. "Understanding Contrastive Learning Requires Incorporating Inductive Biases". ICML 2022

[2] Xue et al. "Investigating Why Contrastive Learning Benefits Robustness Against Label Noise", ICML 2022

**Theoretical Claims:**

- The theoretical claims, especially Theorems 4.1, 4.3, 4.5, and 4.7, are rigorously developed and seem mathematically sound upon review.
- The proofs provided in the supplementary material are comprehensive and clearly structured.
- However, these theoretical results fundamentally address the generalization differences caused by high-quality versus low-quality (noisy) captions rather than any specific theoretical property related to synthetic caption generation itself.
- This raises a question regarding the appropriateness of the term "synthetic" throughout the theoretical analysis, as the main assumptions essentially relate to caption quality rather than caption origin or generation method.
- This ambiguity makes the paper's theoretical contribution less clear when compared to existing literature that already examines the effects of label quality on generalization, such as [1] and [2]. Without explicit differentiation or comparison to these previous works, the specific novelty and significance of the presented theoretical analysis remain uncertain.

[1] Saunshi et al. "Understanding Contrastive Learning Requires Incorporating Inductive Biases". ICML 2022

[2] Xue et al. "Investigating Why Contrastive Learning Benefits Robustness Against Label Noise", ICML 2022

---

> ### Author Rebuttal · Authors · 2025-04-01
>
> We thank the reviewer for the evaluation.
> ## General Response 3: New experiments
> ### Quantitative Results on Silhouette Score
> We agree that t-SNE visualization does not provide statistical evidence for the separation quality between different methods. **To address this limitation, we adopt the Silhouette Score (SS) with cosine distance to quantitatively and statistically assess feature embedding quality.** A higher score indicates better intra-class alignment and inter-class orthogonality, reflecting more purified representations.
> We first calculate the SS in the simulated experiments of Figure 1 in the main paper. As shown in Table 1, when $C_s$ decreases, SS increases, and both the number of neurons learning purified features and the corresponding classification accuracy increase. This empirically supports our theoretical insight that reducing spurious and misaligned data encourages more neurons to learn purified representations, thereby improving the quality of learned embeddings. Moreover, training with a mixture of synthetic and raw data consistently yields significant improvements compared to using raw data alone.
>
> Table 1: Comparison between Raw and Synthetic Data under varying $C_s$
> |$C_s$|Only Raw Data|||Synthetic and Raw data|||
> |---|---|---|---|---|---|---|
> ||SS|# Purified|Accuracy|SS|# Purified|Accuracy|
> |0.00|0.0984±2e-5|49.9|0.9812|0.1022±4e-6|50.0|0.9789|
> |0.10|0.0890±3e-5|49.2|0.9423|0.0991±5e-6|50.0|0.9520|
> |0.20|0.0822±9e-6|49.2|0.8985|0.0959±5e-6|50.0|0.9310|
> |0.30|0.0682±1e-4|43.6|0.8162|0.0926±7e-6|49.5|0.9048|
> |0.40|0.0477±1e-4|38.0|0.6828|0.0802±1e-4|47.5|0.8291|
> |0.50|0.0285±3e-4|30.6|0.5626|0.0669±3e-4|42.8|0.7170|
> ### Vanilla CLIP vs CLIP with synthetic text
> We appreciate the reviewer's concern regarding architectural alignment between theory and experiments. To address this, we did a new experiment to compare vanilla CLIP (trained on original text) with the LaCLIP model [1], where the model has the same architecture and the same training data with the vanilla CLIP, while the only difference is that a fraction of the text are replaced with synthetic text generated by LLM in training LaCLIP.
>
> Tables 2 show the comparison between vanilla CLIP and LaCLIP on CIFAR-100 (Results for CIFAR-10 and Caltech-101 can be found in our response to Reviewer yyfG.). The results show that LaCLIP consistently achieves higher Silhouette Scores than its CLIP counterparts. Since we use cosine distance, a higher Silhouette Score indicates that feature embeddings within the same class are more aligned with high cosine similarity, and embeddings from different classes are more orthogonal with low cosine similarity. This provides quantitative evidence for Theorem 4.3 and 4.5, which show that the neurons can learn purified representations better when some text is replaced with synthetic text.
>
> Table 2: Comparison of CLIP and LaCLIP on CIFAR-100
> |Pre-training Dataset|Model|Accuracy(%)|SS|
> |---|---|---|---|
> |CC3M|CLIP|21.8|-0.0399±0.001|
> ||LaCLIP|**27.5**|**-0.0328±0.001**|
> |CC12M|CLIP|38.5|0.0051±0.001|
> ||LaCLIP|**43.9**|**0.0288±0.002**|
> |RedCaps|CLIP|39.9|-0.0015±0.002|
> ||LaCLIP|**40.7**|**0.0114±0.002**|
> |LAION-400M|CLIP|71.7|0.0781±0.002|
> ||LaCLIP|**73.9**|**0.1081±0.002**|
> ## No theoretical analysis of synthetic text generation
> - We quantitatively prove that synthetic text with filtering exhibits better feature alignment with images than raw text, and that contrastive learning dynamics on such data lead to improved representation learning.
> - To demonstrate the existence of such synthetic text with better feature alignment, we focus on a simplified text generation model in (10) and theoretically prove in Appendices F and G that this even such a simple text decoder can generate synthetic text that is better aligned with images (see GR2 in Reviewer iZBY).
> ## Absence of direct text-quality metrics
> See Section Absence of Caption-Quality Metrics in Reviewer yyfG.
> ## Essential References Not Discussed:
> We thank the reviewer for pointing out relevant papers. We will cite and discuss them in the revision. However, we would like to clarify that our contributions are fundamentally different.
> - We believe the concern stems from a misunderstanding of our theoretical results. Beyond analyzing how feature misalignment affects contrastive learning, we rigorously prove that retexted data contains fewer spurious features and more task-aligned features, which in turn improve representation learning (see GR2 Reviewer iZBY). This theoretical guarantee is not provided in the referenced works.
> - Regarding the role of data quality in generalization, our work offers a detailed analysis of the training dynamics of multimodal encoders $f$ and $h$ with nonlinear activations, which is absent from the mentioned papers. Both Xue et al. and Saunshi et al. directly assume access to a pretrained optimal encoder without analyzing the nonconvex training problem.
>
> References:
> [1] Fan et al., Improving CLIP Training with Language Rewrites

---

> > ### Comment · Reviewer_C7ye · 2025-04-08
> >
> > Thanks for the authors' rebuttal that has resolved some of my concerns. However, I maintain my initial score. I acknowledge that this work consistently demonstrates that high-quality text outperforms low-quality text, but it is difficult to agree that this finding is sufficiently novel. Additionally, while the authors mention in GR2 (response to Reviewer iZBY) that filtered synthetic texts are indeed of high quality, showing that a model trained on high-quality data $S_h$ can generate better-quality texts (after filtering) than the original low-quality data $S_w$ is not significant enough to change my overall evaluation.

---

> > > ### Author Response · Authors · 2025-04-09
> > >
> > > Thank you for the acknowledgment. We are glad to see that some of your concerns have been addressed through our clarifications and additional experiments. However, we respectively disagree with the statement that our contribution is not significant enough. Our work provides the first theoretical understanding of how recaptioning leads to richer and less spurious features, and why high-quality captions further enhance performance by shaping the training dynamics. This is an important and previously unexplored contribution to the development of contrastive learning frameworks.
> > >
> > > First, this paper presents **the first theoretical characterization of how the recaptioning process enhances caption quality**, whereas all prior justifications have been purely empirical. Specifically, this paper is the first work to rigorously prove that synthetic captions (after filtering) reduce spurious feature activation and recover more relevant features, with provable bounds on their probabilities. Such an analysis is particularly challenging, as it requires explicitly characterizing the training dynamics involved in learning from raw data and the recaptioning process. Prior works [1, 2] circumvent this difficulty by directly assuming the existence of a favorable convergence point.
> > >
> > > Second, this paper presents **the first theoretical characterization of the training dynamics in vision-language contrastive learning with nonlinear models**. In contrast, the state-of-the-art analysis in Nakada et al. (2023) is restricted to linear models for both text and image encoders. For such linear models, training dynamics can be studied using singular value decomposition, as shown in Nakada et al. (2023). However, this approach is not applicable to nonlinear models. In our work, both the text and image encoders are nonlinear, requiring us to analyze the behavior of nonlinear activations across three distinct training stages (as shown in Appendix C,D,E in the paper), as well as the non-convex interactions between modalities. Both challenges that do not arise in the linear setting.
> > >
> > > Third, this paper offers **novel theoretical insights that are absent in prior works**. Specifically, we demonstrate that the performance of contrastive learning depends critically on the model’s ability to learn purified features. We also show that captioning and filtering improve the cosine similarity between image-text pairs, which in turn suppresses spurious features and facilitates the learning of purified ones. This provides a new perspective on how contrastive learning performance can be systematically enhanced through data-centric interventions, an aspect that has not been theoretically established in previous studies.
> > >
> > > [1] Saunshi et al. ”Understanding Contrastive Learning Requires Incorporating Inductive Biases”. ICML 2022
> > >
> > > [2] Xue et al. ”Investigating Why Contrastive Learning Benefits Robustness Against Label Noise”, ICML 2022

---

### Official Review · Reviewer_iZBY · 2025-03-18

**Overall Recommendation:** 2

**Summary:**

This paper provides a comprehensive theoretical overview of VLM training dynamics, establishing theoretically why training VLMs with synthetically generated text captions might bring improved downstream performance on zero-shot classification tasks. The paper conducts its analysis with one-hidden-layer neural networks with ReLU activation functions as the backbone for both image and
text encoders. The paper describes that training on such synthetic data reduces the likelihood of spurious correlations between image and text features, and hence improves generalization. Finally, the paper presents some real-world results using BLIP and ALBEF.

**Claims And Evidence:**

Yes, most of the claims made in the paper are supported.

**Essential References Not Discussed:**

None that I know of. The ones that I do know, I have mentioned in the strengths and weaknesses section.

**Experimental Designs Or Analyses:**

I think some of the experiments done in the paper lack real-world grounding, which I describe below in the strengths and weaknesses section.

**Methods And Evaluation Criteria:**

As a stand-alone theoretical work, the paper might be using the right methods for analysis. However, I think there are some key assumptions that I am concerned do not hold true in the real-world, and hence some of the methods used in the paper might not be practically relevant for actual training of VLMs. I lay out these concerns in the strengths and weaknesses section.

**Other Comments Or Suggestions:**

NA

**Other Strengths And Weaknesses:**

Strengths:
- The paper is tackling a novel problem by trying to theoretically establish the connections between VLM pretraining and synthetic captioning. To the best of my knowledge, no prior work has done this.
- The paper seems to be extremely comprehensive in detailing the various assumptions made for the theoretical results.

Weaknesses:
- In my opinion, some of the assumptions made in the paper are too restrictive, and at times, definitely not true in the real world. For example, in Assumption 3.4, the paper assumes perfect alignment to mean z_y_p = z_x_p. This is not possible at all in the real world if I understood this correctly. Further, prior results like the modality gap [Liang et al, Mind the Gap: Understanding the Modality Gap in Multi-modal Contrastive Representation Learning] seem to be in direct opposition of this assumption by showing that in the real-world, this assumption is unlikely to hold true.
- Similarly, in Assumption 3.5, the paper suggests that there can be only one spurious correlation between image and text features. This again seems extremely unlikely to occur in practice.
- The assumption that the image-grounded decoder will be trained on high-quality image-text pairs alone again seems highly unrealistic. In most of the recent VLM pretraining works, the image-grounded decoders themselves are trained on a mix of low-quality and high-quality image-text pairs. In most cases, the vision encoders utilized in these captioners are in-fact trained purely on the same alt-text datasets that are used to train the VLMs themselves (see [Fan et al, Improving CLIP Training with Language Rewrites; Li et al, What If We Recaption Billions of Web Images with LLaMA-3?])
- It is generally understood in the CLIP literature that training purely on synthetic captions leads to worse zero-shot classification performance compared to training with noisy alt-text pairs, see [Li et al, What If We Recaption Billions of Web Images with LLaMA-3?, Zhang et al,  Long-clip: Unlocking the long-text capability of clip]. However, this result seems to be at odds with the theoretical results claimed in the paper, suggesting again that some assumptions made in the paper might not be realistic.

**Questions For Authors:**

In a few cases, VLMs are trained with fusing alt-text pairs with synthetic captions, see [Yu et al, CapsFusion: Rethinking Image-Text Data at Scale; Lai et al, VeCLIP: Improving CLIP Training via Visual-enriched Captions]. Could the paper's current results explain the success of these methods in some way?

**Relation To Broader Scientific Literature:**

The paper adds a key contribution to the VLM pretraining literature from the theoretical side, since there is almost no theoretical work describing the relationship between VLM pretraining and the need for synthetic recaptioning.

**Theoretical Claims:**

I briefly skimmed them, but to be honest I did not verify them too closely for correctness.

---

> ### Author Rebuttal · Authors · 2025-04-01
>
> We thank the reviewer for the valuable time in the evaluation.
> ## General Response
> ### GR1: Clarification of modality misalignment
> - **Our feature misalignment model includes both spurious correlation and less informativeness in the raw text.** The latter means that synthetic text adds relevant features that raw text misses, but the former is not only missing from the text but also wrongly described as unrelated text. **Therefore, less informativeness is merely a simple and special case within our misalignment analysis.** We apologize that Assumption 3.5 only reflects spurious correlations. A more complete form of Assumption 3.5 appears in (67)–(68), where $P_1$ is the probability of spurious features and $1 - P_2$ is the probability of missing relevant features. The improvement lies in reducing both $P_1$ and $1-P_2$ through the generation of synthetic text. **This improvement is rigorously proved, not assumed.** (152) and (151) show that synthetic texts reduce spurious features and retain relevant ones, explaining their benefit in producing more informative content.
> - **Assuming a single spurious feature is a simplification for presentation that was made for ease of presentation in the proof and can be extended to a more general setting without altering the underlying insights.** If each feature $j$ has $K{-}1$ spurious correlates, (38) becomes a $2K{\times}2K$ matrix, and $N_i=\{j,j'\}$ in the last sentence of Theorem 4.3 contains $j$ and other $K-1$ features. Our analysis relies on the total spurious feature probability (bounded by $C_s$), not the number of correlated features, so **as long as the sum of all spurious feature probabilities is upper bounded by $C_s$, the core mechanism and insights of the theorem remain unchanged.**
> ### GR2: Clarification of synthetic text generation
> Regarding the concern of implicitly assuming spurious-free synthetic text, we clarify that our work is not a simple comparison of text quality. Theorem 4.5 does not assume that synthetic text is better.. **Instead, we formally analyze the synthetic text generation process, proving that the generated text is of high quality.** As shown in Appendices F and G, we prove that after synthetic captioning and filtering, the resulting text contains fewer spurious features and more relevant features than raw text. In particular, the probability of spurious features can be reduced from a constant to $\frac{1}{d}$, while the probability of retaining all relevant features increases from $\frac{1}{2}$ to $1-\frac{1}{d}$ as shown in (151) and (152). We apologize for not including these results more prominently in the main text due to space constraints.
> ## Weaknesses1:
> - We think there may be some confusion of $f(x)=g(y)$ and $z_x=z_y$. Although we assume the latter, we do not mean the former holds. In contrast, our analysis shows that $f(x)$ captures information about $z_x$, and $g(y)$ about $z_y$. Although $z_x=z_y$, this does not imply $f(x)=z_x$ or $g(y)=z_y$.
> - $z_x=z_y$ does not imply perfect alignment between image and text due to the presence of noise $\xi$, which can be order-wise larger than the signal itself in Assumption 3.3(d).
> - This modeling assumption $z_x=z_y$ is standard in contrastive learning analysis, such as in [2].
> ## Weaknesses2:
> Please see GR1.
> ## Weaknesses3:
> We believe this is a misunderstanding from our imprecise statement in section 2.1, where we say "We use the high-quality data pairs in $S_h$ to train an image-grounded text decoder." Here, what we should have written is that the image-grounded text decoder is FINE-TUNED on $S_h$. We consider a simplified image-grounded text decoder, which is initialized from the weights $\overline{\mathbf{W}}$ and $\overline{\mathbf{V}}$ learned in stage 1 using a mixture of low-quality and high-quality image-text pairs. The decoder is then fine-tuned on high-quality pairs. Our setup is consistent with many real-world systems such as BLIP, LLaVA, and GIT, where text decoders are initially trained on noisy datasets like LAION and subsequently fine-tuned on curated data such as COCO. We also add the comparison of vanilla CLIP and LaCLIP [Fan et al, Improving CLIP Training with Language Rewrites], further validating our theoretical framework (see GR3 in Reviewer C7ye).
> ## Weaknesses4:
> We do not consider training on synthetic text only. In Section 2.1, we adopt a partial replacement strategy, where only detected noisy text is replaced with synthetic text, while the rest of the original dataset remains unchanged. In stage 4, the model is retrained using a mixture of raw and synthetic text.
> ## Questions For Authors:
> Following the previous comment, our paper exactly focuses on how fusing alt-text pairs with synthetic text affects the representation learning.
>
> References:
>
> [1] Liang et al., Mind the Gap: Understanding the Modality Gap in Multi-modal Contrastive Representation Learning
>
> [2] Chen et al., Understanding Transferable Representation Learning and Zero-Shot Transfer in CLIP

---

### Official Review · Reviewer_feb2 · 2025-03-24

**Overall Recommendation:** 2

**Summary:**

This paper presents the first theoretical analysis of the training dynamics of vision-language models (VLMs) with nonlinear activation functions and provides a theoretical justification for the effectiveness of synthetic text captions in improving pre-training performance. Specifically, the authors analyze the impact of misaligned image-text pairs using a one-hidden-layer neural network model, showing that neurons trained on noisy data tend to learn a mixture of true and spurious features. The paper further attempts to validate the theoretical and simulation results through experiments using BLIP and ALBEF.

**Claims And Evidence:**

While the theoretical analysis presented in this work is novel and technically sound within its scope, I have concerns about the strong gap between the theoretical assumptions (Assumptions 3.3–3.5) and the realistic settings used in real models (ALBEF and BLIP).


1. **Oversimplified model architecture and loss function**: While the use of a one-hidden-layer neural network with a spectral contrastive loss provides analytical tractability, it remains unclear whether such simplifications can meaningfully approximate the training dynamics of real-world VLMs such as ALBEF or BLIP. These models adopt significantly more complex transformer-based architectures with 12 layers of self-attention and cross-attention, and employ a multi-modal fusion design where the text encoder (e.g., BERT) receives cross-attention from the image encoder. Furthermore, their training objectives include not only contrastive loss, but also image-text matching (ITM) loss and language modeling losses (masked or autoregressive). As such, the theoretical modeling based on Equations (1) and (3) in the paper more closely aligns with CLIP-style architectures and should be interpreted with this constraint.

2. **Continued architectural and methodological mismatches between theory and experiments**: While the paper attempts to validate its theoretical claims through comparisons between ALBEF and BLIP, it is important to note that these two models differ significantly in both architecture and training objectives. Specifically, ALBEF uses a masked language modeling (MLM) loss, whereas BLIP adopts an autoregressive language modeling (AR) loss. These differing objectives necessitate different model architectures, as documented in their respective original papers. Consequently, attributing the differences in Figure 3–5 solely to the use of synthetic captions may be misleading—especially if authors simply utilize pre-trained weights from the public, where multiple factors (architecture, loss functions, data) are entangled. Again, it seems that the theory proposed in this paper would be more aligned with CLIP. I think authors should compare vanilla CLIP vs CLIP (with synthetic captions with the same amount of data). Furthermore, it is highly unclear whether the simple visualization-based comparisons (e.g., t-SNE plots and cosine similarity histograms in Figures 4 and 5) offer validation of the theoretical claims. I believe more concrete explanations and experiments would be required. The main claim regarding the presence and influence of spurious features is not directly verifiable through the current experimental setup. Moreover, the difficulty of demonstrating this phenomenon using existing models such as ALBEF and BLIP highlights a substantial gap between the proposed theoretical framework and the real VLMs.

3. **Modeling over oversimplified assumptions**:
- Concerns on assumption 3.5: The assumption 3.5 makes a claim that every image feature in low-quality data is spuriously correlated with exactly one text feature with a constant probability C. However, in practice (as also noted in Section 5.1), spurious correlations in large-scale web data often involve multiple spurious features. Even in their simulation setting (Section 5.1), the authors adopt a more general experimental setup where each image feature can be spuriously correlated with any text feature. This discrepancy highlights a gap between the core assumption used for theoretical analysis and the real scenarios.

- Implicit assumption on spurious-free synthetic captions: I'm not sure that my understanding is right (Please correct me if my understanding is wrong), but it seems that several theorems (particularly Theorem 4.5) implicitly assume that the synthetic captions generated by the image-grounded decoder G are free from spurious features.  Although the decoder is trained on high-quality data, there is no guarantee that its outputs are fully purified, especially given that the encoder used to train G was itself pre-trained on noisy image-text pairs during the initial stage of the real training pipeline. This concern is also shown in simulation results (Figure 1), where the performance of synthetic data slightly degrades as the level of spurious correlation increases. I am therefore concerned that the theoretical claims may overstate the idealized nature of synthetic captions.

- Assumption limited to spurious features: The authors primarily attribute the effectiveness of synthetic captions to their ability to mitigate spurious features. However, the analysis does not account for other important factors that may contribute to improved performance. For example, synthetic captions often provide more detailed and descriptive content—for example, "a black dog sitting on the couch" instead of a caption like "a dog".  While the paper assumes perfect alignment in high-quality data, it does not explicitly account for this aspect of informativeness or semantic detail.

4. **Comparison with prior theoretical work**: While the paper highlights its contributions in Table 1 by comparing with prior theoretical studies, it lacks a concrete and quantitative discussion of how its analytical framework differs from existing approaches or advances beyond them.  Moreover, the paper lacks experimental validation to substantiate whether its theoretical insights yield stronger or more generalizable results compared to prior work. I believe the authors should include more concrete explanations regarding methodological differences, and ideally, provide supporting empirical results.

5. **Difficulty in finding theorem proofs in the appendix**: It is quite difficult to find the corresponding proof for each theorem in the appendix, as the paper does not provide clear references or section labels linking main theorems to their proofs.


Given the current concerns, I believe a revision would be needed to adequately address these issues. However, I would still like to read the authors’ rebuttal and the comments from other reviewers.

**Essential References Not Discussed:**

I believe the essential references are included.

**Experimental Designs Or Analyses:**

Wrote them above.

**Methods And Evaluation Criteria:**

Wrote them above

**Other Comments Or Suggestions:**

Wrote them above

**Other Strengths And Weaknesses:**

Wrote them above

**Questions For Authors:**

Wrote them above

**Relation To Broader Scientific Literature:**

I believe that the attempt to provide theoretical justifications for VLMs is academically significant, as such analyses have not been thoroughly developed in prior work.

**Theoretical Claims:**

I tried to check them

---

> ### Author Rebuttal · Authors · 2025-04-01
>
> ### **We STRONGLY recommend the reviewer to first read the General Responses 1 and 2 provided in Reviewer iZBY's rebuttal because of the space limit.**
>  We thank the reviewer for the valuable time in the evaluation.
> ## Oversimplified model architecture and loss function
> - The training dynamics analysis of one-hidden-layer neural networks with non-linear activation function is the SOTA for contrastive learning. As shown in Table 1 in the main paper, the model we consider is already the most advanced for theoretical analysis among existing theoretical works regarding modeling fidelity. However, other works are limited to linear models or focus solely on training a single encoder in the VLM setting.
> - Our paper aims to provide theoretical explanations for the advantage of synthetic captions within the CLIP-style learning framework. We initially chose BLIP and ALBEF as testbeds to empirically verify the theoretical benefits of synthetic captions and took necessary steps to minimize the differences between the two models (see details in the next response). To further strengthen our findings, we have added new experiments using vanilla CLIP and LaCLIP (trained with partially synthetic captions) (see GR3 in Reviewer C7ye).
> ## Continued architectural and methodological mismatches between theory and experiments
> - We fully agree with the reviewer that comparing vanilla CLIP with CLIP pre-trained on the same dataset with captioner and filter would be valuable. Thus, We have added such a new experiment (see GR3 in Reviewer C7ye).
> - We initially considered BLIP and ALBEF because, at that time, publicly available vanilla CLIP models and CLIP models trained with a captioner and filter were not known to us. Consequently, we selected BLIP and ALBEF, as both use the same 14M pre-training dataset—with and without the captioner and filter, respectively (as reported in BLIP Section 4.1). We acknowledge the differences of model architecture and loss functions as pointed out by the reviewer. As an attempt to reduce the impact of these differences, in the paper, we focus exclusively on the Image-Text Contrastive (ITC) pathway in these two models, avoiding cross-attention, fusion, or decoding modules, as the ITC pathways of these two models share the same architecture.
> ## Modeling over oversimplified assumptions:
> We thank the reviewer for raising these concerns. However, we believe they arise from misunderstandings due to unclear presentation in our submission rather than fundamental weaknesses in our work.
> To clarify, we do not assume that synthetic captions are free of spurious correlations. Instead, we formally prove that synthetic captions help reduce such correlations (see GR2 in Reviewer iZBY). Furthermore, our model of image-text feature misalignment considers not only spurious correlations but also missing features, where raw captions often lack detailed and descriptive content, while synthetic captions provide richer information (see GR1 in Reviewer iZBY).
>
> **Regarding Figure 1:** Figure 1 in the main paper is consistent with our theoretical results. Theorems 4.3 and 4.5 require $C_s$ less than 1/2, as indicated in Assumption 3.5, meaning that the probability of misalignment cannot be too large. This is consistent with the resulting in Figure 1 that the performance degrades as $C_s$ increases.
> ## Comparison with Prior Theoretical Work
> - We rigorously prove that recaptioned data contains fewer spurious features and more task-aligned features, which in turn improve representation learning. Specifically, the probability of spurious feature activation is reduced from a constant to $\frac{1}{d}$, while the probability of retaining all relevant features increases from $\frac{1}{2}$ to $1-\frac{1}{d}$. These theoretical guarantees are not provided in prior works.
> - Our work provides a detailed analysis of the training dynamics of multimodal encoders $f$ and $h$ with nonlinear activations. Unlike most previous studies, we consider the realistic case where both encoders are jointly trained under a non-convex objective with multi-weight interactions. For example, [1] only considers a single encoder, while [2] only analyzes a linear model.
> - Both [3] and [4] assume access to a pretrained optimal encoder, without analyzing the optimization dynamics of contrastive learning.
> ## Difficulty in Finding Theorem Proofs in the Appendix
> We will provide brief proof sketches in the main paper to guide the reader toward the detailed derivations in the appendix.
>
> References:
>
> [1] Wen et al., Toward Understanding the Feature Learning Process of Self-Supervised Contrastive Learning
>
> [2] Li et al., Understanding Multimodal Contrastive Learning and Incorporating Unpaired Data
>
> [3] Saunshi et al., Understanding Contrastive Learning Requires Incorporating Inductive Biases
>
> [4] Xue et al., Investigating Why Contrastive Learning Benefits Robustness Against Label Noise

---

> > ### Comment · Reviewer_feb2 · 2025-04-09
> >
> > Thank you for the thoughtful rebuttal. Some of my concerns—particularly regarding the assumptions—have been addressed. However, I still believe that the current experiments, which rely primarily on simple statistics or visualizations, are insufficient to convincingly link the theoretical results to the behavior of real models (merely observing the strong performance of models that use synthetic clean captions does not seem sufficient). To address this, I believe a new version of the paper with more concrete experiments is needed to bridge the gap, including resolving the mismatch in the choice of target models. Therefore, I maintain my score.

---

> > > ### Author Response · Authors · 2025-04-09
> > >
> > > Dear Reviewer feb2,
> > >
> > > Thank you very much for your updated comments.
> > >
> > > We have already made our best effort to enhance the experiments by including new experiments on CLIP models. In terms of evaluation, we have quantitatively assessed both the separation between different classes and the accuracy on downstream tasks.  Please see General Response 3 to Reviewer C7ye. We are unsure about the specific types of experiments you are looking for. Could you please clarify what additional experiments you would like to see?
> > >
> > > Best regards,
> > > Authors

---

### Decision · Program_Chairs · 2025-05-01

**Decision:**

Reject

**Comment:**

This paper provides the theoretical analysis of VLM training with nonlinear activations and theoretically shows that synthetic text captions enhance pre-training performance (by showing that synthetic captions reduce spurious correlations in noisy image-text data). This improves feature alignment and generalization, enhancing zero-shot performance. Findings are supported by experiments on models like BLIP and ALBEF.

This paper has mixed opinions, mostly negative ones (3 weak reject and 1 accept). After the reviewer-AC discussion, Reviewer yyfG also agreed that the current shape is not strong enough to be accepted. The concerns raised by the reviewers are summarized below:

1. The theoretical assumptions are too strong to be extended to the real models (ALBEF and BLIP), hence, the current theoretical results are unrealistic. (feb2, iZBY)
2. Weak experimental validation to support their claim -- only visualization and Silhouette Scores (SS) were provided; the models are relatively smaller than widely used models (feb2, C7ye, yyfG)
3. Lack of comparisons (or contradictory results) with prior works (feb2, iZBY, C7ye)
4. The terminology "synthetic" could be misleading or somewhat unrealistic (iZBY, C7ye)

Although this is a theoretical work, I also agree with the reviewers' arguments. The current theoretical results are too narrow to be extended to the real-world models, as well as, the current empirical validation is not sufficiently strong to support their theoretical claims.

I don't think that all theories should be well-aligned with real-world models or that all experimental results are strong to show a new "state-of-the-art". However, because the position of this paper is to explain the realistic scenario in theory, I think that the assumption would be more general and the experimental results would be more at scale.

I believe that the authors' effort to show the theoretical result for the relationship between synthetic captions and spurious correlations could be beneficial, but due to the above reasons, I think the current paper has a large room for improvement. I strongly recommend the authors (1) improve their theoretical results to be more realistic **_or_** (2) add more empirical validations to be sufficiently sound to practitioners to make their submission stronger. Also, I understand that it is very difficult to design an experiment to directly support their own theoretical claim, but as the reviewers already pointed out, it would be great if the authors can add more experiments to directly support their claims (I don't think SS is sufficient).

Overall, I recommend reject for this paper.